# BIT-AWARE RANDOMIZED RESPONSE FOR LOCAL DIFFERENTIAL PRIVACY IN FEDERATED LEARNING

## ABSTRACT

In this paper, we develop **BitRand**, a bit-aware randomized response algorithm, to preserve local differential privacy (LDP) in federated learning (FL). We encode embedded features extracted from clients' local data into binary encoding bits, in which different bits have different impacts on the embedded features. Based upon that, we randomize all the bits to preserve LDP with three key advantages: **(1)** Bit-aware: Bits with a more substantial influence on the model utility have smaller randomization probabilities, and vice-versa, under the same privacy protection; **(2)** Dimension-elastic: Increasing the dimensions of embedded features, gradients, model outcomes, and training rounds marginally affect the randomization probabilities of binary encoding bits under the same privacy protection; and **(3)** LDP protection is achieved for both embedded features and labels with tight privacy loss and expected error bounds ensuring high model utility. Extensive theoretical and experimental results show that our BitRand significantly outperforms various baseline approaches in text and image classification.

## 1 INTRODUCTION

Recent data privacy and security regulations (GDPR, 2018; Regulation, 2018; Cybersecurity Law, 2016) pose major challenges in collecting and using personally sensitive data in different places for machine learning (ML) applications. Federated Learning (FL) (McMahan et al., 2017; Kairouz et al., 2019) is a promising way to address these challenges, enabling clients to jointly train ML models by sharing and aggregating gradients computed from clients' local data through a coordinating server for model updates. However, recent attacks (Zhu et al., 2019; Y. et al., 2021; Zhao et al., 2020a) have shown that clients' training samples, each of which includes an input $x$ and a ground-truth label $y_x$, can be extracted from the shared gradients. These attacks underscore the implicit privacy risk in FL.

Our main goal is to provide a strong guarantee that the shared gradients protect the privacy of clients' local data without undue sacrifice in model utility. Local differential privacy (LDP) has emerged as a crucial component in various FL applications (Yang et al., 2019; Kairouz et al., 2019). To achieve our goal, we focus on preserving LDP in cross-device FL, i.e., in which clients jointly train an FL model (Kairouz et al., 2019).

In cross-device FL, existing LDP-preserving approaches can be categorized into three lines: Clients **(1)** add noise into local gradients derived from their local training samples, e.g., using Gaussian mechanism (Abadi et al., 2016), to protect membership information at the training sample level with DP guarantees (Zheng et al., 2021; Dong et al., 2019; Malekzadeh et al., 2021; Geyer et al., 2017; Huang et al., 2020), **(2)** add noise to local gradients using Randomized Response (RR) mechanisms to protect the values of the local gradients with LDP guarantees (Sun et al., 2021; Liu et al., 2020; Zhao et al., 2020b; Wang et al., 2019a), and **(3)** add noise into each training sample, e.g., embedded features and labels, using RR mechanisms to protect the value of each training sample with LDP guarantees (Arachchige et al., 2019; Lyu et al., 2020a), then the clients use LDP-preserved training samples to derive local gradients. For all three approaches, in each training round, clients send DP or LDP-preserved local gradients to the coordinating server for model updates, which will be sent back to the clients for the next training round.

In this paper, we focus on protecting clients' training data at the value level with LDP guarantees. Existing RR mechanisms to preserve LDP in FL suffer from the *curse of privacy composition*, in which excessive privacy budgets are consumed proportionally to the large dimensions of input or embedded features (Arachchige et al., 2019), gradients (Zhao et al., 2020b; Wang et al., 2019a), and training rounds (Zhao et al., 2020b; Wang et al., 2019a), causing loose privacy protection or inferior model accuracy (Wagh et al., 2021).

Addressing the curse of privacy composition is non-trivial. Existing approaches, such as anonymizers (assumed to be trusted), i.e., shuffler (Erlingsson et al., 2019; Sun et al., 2021; L. et al., 2020; Wang et al., 2019b; Cheu et al., 2019; Balle et al., 2019) or anonymity approaches (e.g., faking source IP, VPN, Proxy, etc. (Sun et al., 2021; Cormode et al., 2018)), and dimension reduction (Liu et al., 2020; Zhao et al., 2020b; Shin et al., 2018; Xu et al., 2019), mitigate the problem but also have limitations. In the real world, it is possible that the anonymizers can either be compromised or collude with the coordinating server to extract sensitive information from observing LDP-preserved local gradients (Erlingsson et al., 2019). Meanwhile, applying RR mechanisms on reduced sets of embedded features or gradients using dimension reduction techniques can work well with lightweight models, such as logistic regression and SVM (Liu et al., 2020; Zhao et al., 2020b; Wang et al., 2019a). However, it is challenging for these techniques to achieve good model utility under tight LDP guarantees given complex models and tasks, such as DNNs, since the dimensions of reduced embedded features, gradients, and training rounds still need to be sufficiently large (Zhao et al., 2020b; Liu et al., 2020).

Hence, the curse of privacy composition in preserving LDP by applying RR mechanisms in FL remains a largely open problem. Orthogonal to this, preserving LDP to protect ground-truth labels $y_x$ in FL has not been well-studied. Two known approaches for centralized training are 1) injecting Laplace noise into the labels (Phan et al., 2020) and 2) applying RR mechanisms on the labels to achieve DP at the label level (Ghazi et al., 2021). However, centralized training in Ghazi et al. (2021) has not been designed for FL with LDP guarantees since they require centralized and trusted databases. The model utility in Phan et al. (2020) is notably affected by the number of model outcomes.

**Key Contributions.** To mitigate the curse of privacy composition and optimize the trade-off between privacy and model utility, our paper is structured around the following contributions:
**1)** We propose **BitRand**, which is a combination of a novel *bit-aware $f$-**RR** mechanism and *label*-**RR** mechanism, to preserve LDP at both levels of embedded features and labels in FL. In $f$-**RR**, we encode embedded features (extracted from $x$) into a binary-bit string, which will be adaptively randomized such that bits with *a more substantial impact* on model utility will have *smaller randomization probabilities* and vice-versa under the same privacy budget. To preserve LDP on $y_x$, we develop a generalized randomization, in which the probability of randomizing label $y_x$ from one to another class is a function of the number of model outcomes $C$. By doing that, we can optimize the trade-off between model utility and privacy loss with significantly tighter expected error bounds.
**2)** By incorporating sensitivities of binary encoding bits into a generalized privacy loss bound, we show that increasing the dimensions of embedded features $r$, encoding bits $l$, and model outcomes $C$ marginally affect the randomization probabilities in BitRand under the same privacy budget. This *dimension-elastic* property is crucial to evade the curse of privacy composition by retaining a high value of data transmitted correctly through our randomization given large dimensions of $r$, $l$, and $C$.
**3)** These bit-aware and dimension-elastic properties allow us to work with complex models and tasks with formal LDP guarantees for training samples $(x, y_x)$ while retaining high model utility. Extensive theoretical analysis and experimental results conducted on fundamental FL tasks, i.e., text and image classification, using benchmark datasets and our collected Security and Exchange Commission financial contract dataset show that our BitRand significantly outperforms a variety of baseline approaches in terms of model utility under the same privacy budget.

## 2 BACKGROUND

LDP-preserving mechanisms (Erlingsson et al., 2014; Duchi et al., 2018; Wang et al., 2017; Acharya et al., 2019; Bassily & Smith, 2015) generally build on the ideas of randomized response (Warner, 1965), which was initially introduced to allow survey respondents to provide their correct inputs while maintaining their confidentiality. $\epsilon$-LDP is presented as follows:

**Definition 1.** *$\epsilon$-LDP. A randomized algorithm $\mathcal{M}$ fulfills $\epsilon$-LDP, if for any two inputs $x$ and $x'$, and for all possible outputs $\mathcal{O} \in Range(\mathcal{M})$, we have: $Pr[\mathcal{M}(x) = \mathcal{O}] \leq e^\epsilon Pr[\mathcal{M}(x') = \mathcal{O}]$, where $\epsilon$ is a privacy budget and $Range(\mathcal{M})$ denotes every possible output of the algorithm $\mathcal{M}$.*

The privacy budget $\epsilon$ controls the amount by which the distributions induced by inputs $x$ and $x'$ may differ. A smaller $\epsilon$ enforces a stronger privacy guarantee. We revisit RR mechanisms for LDP preservation in **Appendix A**. Our approach is a binary encoding-based approach, similar to (Arachchige et al., 2019; Lyu et al., 2020a), since it has the potential to overcome the curse of privacy composition. In binary encoding, $x$ is converted into an $l$-bit vector $v$ consisting of 1 sign bit, $m$ bits for the integer part, and $l-m-1$ bits for the fraction part (Figure 1), as follows:

$$\forall i \in [0, l-1]: v_i = \lfloor 2^{i-m}|x| \rfloor \mod 2 \tag{1}$$

Each bit in $v$ is randomized by applying a RR mechanism, e.g., (Erlingsson et al., 2014; Bassily & Smith, 2015; Wang et al., 2017), to generate a perturbed $l$-bit vector $v'$, which preserves LDP. However, in our theoretical reassessment (**Appendices I** and **J**), directly applying RR mechanisms on binary encoded vectors as in existing mechanisms, i.e.,

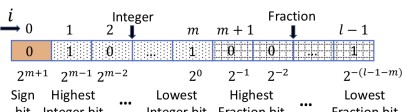

Figure 1: Binary encoding.

LATENT (Arachchige et al., 2019) and OME (Lyu et al., 2020a), consumes huge privacy budgets since each binary encoding bit cannot be treated as a bit in a hash. Each binary encoding bit $i$ has a different sensitivity, i.e., $\Delta_i = 2^{m-i}$ for the integer and fraction parts or $\Delta_i = 2^{m+1}$ for the sign bit (**Lemma 1**), compared with a bit $B_i$ in a hash, i.e., $\forall B_i : \Delta_{B_i} = 1$. Our mechanism does not suffer from this problem, thanks to our bit-aware randomization in binary encoding (**Theorem 2**).

## 3 BITRAND ALGORITHM

Let us now present our FL setting, threat model, and BitRand algorithm (Figure 2 and Alg. 1, Appendix B), and privacy guarantees. Then, we will show that our algorithm is dimension-elastic and the ability to optimize the randomization probabilities with expected error bounds in our theoretical analysis.

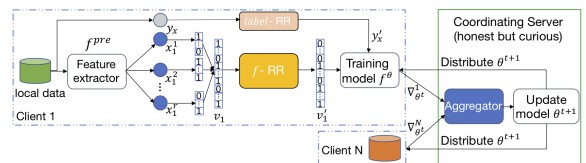

Figure 2: Federated Learning with BitRand Algorithm.

**Federated Learning** Given $N$ clients, each client $u \in [1, N]$ has a set of training samples $D_u = \{(x, y_x)\}_{n_u}$ where $x \in \mathbb{R}^d$ is the input features, and its associated ground-truth label $y_x \in \mathbb{Z}^C$ is one-hot encoded with $C$ categorical model outcomes $\{y_{x,1}, \ldots, y_{x,C}\}$, and $n_u$ is the number of training samples. In a pre-processing step, each client $u$ extracts $r$ numerical embedded features from $x$, denoted $e_x$, by using a pre-trained model $f^{pre}$. In practice, $f^{pre}$ could be trained on large-scale and publicly available datasets without introducing any extra privacy cost to the clients' local data (He et al., 2016; Devlin et al., 2018). $N$ clients jointly train the model $f^\theta : \mathbb{R}^r \to \mathbb{R}^C$ by minimizing a loss function $\mathcal{L}\big(f^\theta(e_x), y_x\big)$ on their local training samples in $D_u$ that penalizes mismatching between the prediction $f^\theta(e_x)$ and the ground-truth label $y_x$, given the model's parameters $\theta$. In each training round $t$, each client $u$ receives the most updated model parameters $\theta^t$ from the coordinating server and then computes local gradients $\nabla^u_{\theta_t} = \sum_{(v_x, y_x) \in D_u} \nabla_{\theta_t} \mathcal{L}(f(e_x), y_x)/n_u$, which are sent to the coordinating server for aggregating and model updating: $\theta_{t+1} = \theta_t - \eta_t \sum_{u \in [1,N]} \nabla^u_{\theta_t}/N$.

**Threat Model.** The coordinating server strictly follows the training procedure but curious about the training data $D_u$. This is a practical threat model in the real world since service providers always aim at providing the best services to the clients (Haeberlen et al., 2011; Truex et al., 2019; Lyu et al., 2020b). Given the observed gradients $\nabla^u_{\theta_t}$, the coordinating server can extract the clients' data $\{e_x, y_x\}_{n_u}$ by using recently developed attacks (Carlini et al., 2020; Fredrikson et al., 2015). In a defense-free environment, $\{e_x, y_x\}_{n_u}$ can be used to infer the sensitive training data $D_u = \{(x, y_x)\}_{n_u}$ using the pre-trained model $f^{pre}$, since $f^{pre}(x) = e_x$ (Song & Raghunathan, 2020). This poses a severe privacy risk to the sensitive data $D_u$.

**BitRand Algorithm.** To protect the sensitive training data $D_u$ against the threat model, in our algorithm, we preserve LDP on both embedded features $e_x$ and labels $y_x$.

Each of the $r$ embedded features in $e_x$ is encoded into $l$ binary bits following Eq. 1. Binary encoded features are concatenated together into a vector $v_x$ consisting of $rl$ binary bits to represent the embedded features $e_x$ (Alg. 1, line 16). Each bit $i \in [0, rl - 1]$ in $v_x$ is randomized by our $f$-**RR** mechanism (Alg. 1, line 17) with a *bit-aware term* $\frac{i\%l}{l}$ optimizing randomization probabilities:

$$(f\text{-RR mechanism}) \quad \forall i \in [0, rl-1] : P(v'_x(i) = 1) = \begin{cases} p_X = \dfrac{1}{1 + \alpha \exp(\frac{i\%l}{l}\epsilon_X)}, & \text{if } v_x(i) = 1 \\ q_X = \dfrac{\alpha \exp(\frac{i\%l}{l}\epsilon_X)}{1 + \alpha \exp(\frac{i\%l}{l}\epsilon_X)}, & \text{if } v_x(i) = 0 \end{cases} \tag{2}$$

where $v_x(i) \in \{0, 1\}$ is the value of $v_x$ at the bit $i$, $v'_x$ is the perturbed vector created by randomizing all the bits in $v_x$, $\epsilon_X$ is a privacy budget, and $\alpha$ is a parameter bounded in Theorem 2. From Eq. 2, we also have that $P(v'_x(i) = 0) = 1 - p_X$ if $v_x(i) = 1$, and $P(v'_x(i) = 0) = 1 - q_X$ if $v_x(i) = 0$. We use the bit-aware term $\frac{i\%l}{l}$ to indicate the location of bit $i$, which is associated with the sensitivity of the bit at that location, in its $l$-bit binary encoded vector among $rl$ concatenated binary bits.

One of the key differences between our mechanism and existing works (Sun et al., 2021; Zhao et al., 2020b; Wang et al., 2019a; Liu et al., 2020; Arachchige et al., 2019; Lyu et al., 2020a) is the bit-aware randomization probabilities. By introducing the bit-aware term $\frac{i\%l}{l}$, we are able to: 1) Derive significantly tighter privacy loss and expected error bounds compared with existing approaches (**Sections 4 and 5**); and 2) Adaptively control the randomization probabilities across bits, such that bits with a *stronger influence* on the model utility, e.g., sign bits and integer bits, have *smaller randomization probabilities* $q_X$, and vice-versa (**Section 5**). These advantages are crucial to evade the curse of privacy composition enabling us to work with complex tasks and models (**Section 6**).

In addition, inspired by the LabelDP (Ghazi et al., 2021), we randomize $y_x$ using the following *label*-**RR** mechanism (Alg. 1, line 18):

$$(\textit{label-}\text{RR mechanism}) \quad P(y'_x = \bar{y}_x) = \begin{cases} p_Y = \dfrac{\exp(\beta)}{1 + \exp(\beta)}, & \text{if } \bar{y}_x = y_x \\ q_Y = \dfrac{1}{(1 + \exp(\beta))(C - 1)}, & \text{if } \bar{y}_x \neq y_x, \bar{y}_x \in \mathbb{Z}^C \end{cases} \tag{3}$$

where $\bar{y}_x$ is one-hot encoded, and $\beta$ is a parameter bounded in **Theorem 3** under a privacy budget $\epsilon_Y$.

Randomizing the label $y_x$ provides a complete LDP protection to each local training sample $(e_x, y_x)$. All the perturbed training samples $(e'_x, y'_x)$ are included in a local dataset $D'_u$, which will be used to train the model $f^\theta : \mathbb{R}^r \to \mathbb{R}^C$, i.e., $e'_x = \mathcal{E}(v'_x)$ where $\mathcal{E}(\cdot)$ is a decoding function, (Alg. 1, lines 20-22). The training will never access the original data $D_u$. All other operations remain the same with our aforementioned FL setting.

## 4 PRIVACY GUARANTEES OF BITRAND

In this section, we focus on bounding formal privacy loss of BitRand for input and label protection. To achieve our goal, we need to bound $\alpha$ and $\beta$ in Eqs. 2 and 3 such that our algorithm preserves LDP for each training sample $(x, y_x) \in D_u$ given $(v'_x, y'_x) \in D'_u$. Note that the decoding $e'_x = \mathcal{E}(v'_x)$ does not incur any extra privacy risk following the post-processing property (Dwork & Roth, 2014). Given $v_x$ and $\widetilde{v}_x$ can be different at any bit, for the LDP condition to hold, the ratio of two probabilities $\frac{P(f\text{-RR}(v_x)=v_z)}{P(f\text{-RR}(\widetilde{v}_x)=v_z)}$ needs to be bounded by $\exp(\epsilon_X)$ with $v_z \in Range(f\text{-RR})$.

Given a feature $a$, i.e., one of the $r$ features, let us first consider a bit $i$ in the encoding vector $v_a$. Intuitively, when we apply our $f$-RR only on the bit $i$ in $v_a$ (i.e., all the other bits remain the same), denoted as $f$-RR$(v_a, i)$, the $l_1$-sensitivity of the bit $i$ is 1. This is because, given *two neighboring vectors $v_a$ and $v_{a|i}$ that differs only at the bit $i$*, we have that $\forall i \in v_a : \arg\max_{v_a} \|f\text{-RR}(v_a, i) - f\text{-RR}(v_{a|i}, i)\|_1 = 1$. However, the coordinating server can infer the decoded features $\mathcal{E}(f\text{-RR}(v_a))$ instead of just the intermediate result $f$-RR$(v_a)$. Thus, we need to quantify the $l_1$-sensitivity of a single encoding bit $i \in v_a$ to determine just how accurately we can return the decoded features $\mathcal{E}(f\text{-RR}(v_a))$ through our randomization $f$-RR applied on $v_a$. Following (Dwork & Roth, 2014), the sensitivity $\Delta_i$ of a bit $i$ can be quatified as follows:

**Definition 2.** *Bit $l_1$-sensitivity. Given two neighboring vectors $v_a$ and $v_{a|i}$ that differs only at a bit $i$, the sensitivity $\Delta_i$ captures the magnitude by which the bit $i$ can change the decoding function $\mathcal{E}(f\text{-RR}(\cdot))$ in the worst case, as follows:*

$$\forall i \in [0, rl-1] : \Delta_i = \max_{v_a} \|\mathcal{E}(f\text{-RR}(v_a)) - \mathcal{E}(f\text{-RR}(v_{a|i}))\|_1 \tag{4}$$

Based on Eq. 4, sensitivities $\Delta_i$ of all the bits in $v_x$ are bounded in the following lemma.

**Lemma 1.** *The $l_1$-sensitivity of a single binary encoding bit is bounded as follows:*

$$\forall i \in [0, rl-1] : \Delta_i = \begin{cases} 2^{m+1}, & \text{if } i \text{ is a sign bit} \\ 2^{m-i\%l}, & \text{if } i \text{ is an integer/fraction bit} \end{cases} \tag{5}$$

**All the proofs are in Appendix.** Unlike existing RR mechanisms, we incorporate the $l_1$-sensitivity $\Delta_i$ into the privacy loss bound of $f$-RR by ensuring that the privacy loss in randomizing a bit $i$ is bounded by the privacy loss in the embedded feature space. By doing so, we can derive tighter privacy loss bounds as discussed next.

Given $\Delta_i$, there always exists a Laplace noise injected into the feature $a$, i.e., $\mathcal{M}(v_a, i) = \mathcal{E}(v_a) + Lap(\Delta_i/\epsilon_i)$, to achieve $\epsilon_i$-LDP in the embedded feature space (Dwork & Roth, 2014). In other words, $\frac{P(\mathcal{M}(v_a, i)=z)}{P(\mathcal{M}(v_{a|i}, i)=z)} \leq \exp\left(\frac{\epsilon_i |\mathcal{E}(v_a) - \mathcal{E}(v_{a|i})|}{\Delta_i}\right)$ and $z \in Range(\mathcal{M})$. To ensure that the privacy loss

in randomizing the bit $i$ is bounded by the privacy loss in the embedded feature space is to find $\alpha$ in Eq. 2 such that: $\frac{P(f\text{-RR}(v_a,i)=v_z)}{P(f\text{-RR}(v_{a|i},i)=v_z)} \leq \exp(\frac{\epsilon_i|\mathcal{E}(v_a)-\mathcal{E}(v_{a|i})|}{\Delta_i})$. However, randomizing a binary encoding bit $i$ given $v_a$ and $v_{a|i}$ results in a smaller $l_1$-distance $|\mathcal{E}(f\text{-RR}(v_a,i))-\mathcal{E}(f\text{-RR}(v_{a|i},i))|$ compared with $|\mathcal{E}(v_a)-\mathcal{E}(v_{a|i})|$; since $|\mathcal{E}(f\text{-RR}(v_a,i))-\mathcal{E}(f\text{-RR}(v_{a|i},i))| \leq |\mathcal{E}(v_a)-\mathcal{E}(v_{a|i})|$. Thus, we can derive a tighter bound by replacing $|\mathcal{E}(v_a)-\mathcal{E}(v_{a|i})|$ with $|\mathcal{E}(f\text{-RR}(v_a,i))-\mathcal{E}(f\text{-RR}(v_{a|i},i))|$. Given $v_a(i)$ is the bit $i$ in $v_a$, we have

$$\frac{P(f\text{-RR}(v_a,i)=v_z)}{P(f\text{-RR}(v_{a|i},i)=v_z)} = \frac{P(f\text{-RR}(v_a(i))=v_z(i))}{P(f\text{-RR}(v_{a|i}(i))=v_z(i))} \times \prod_{j\neq i, j\in[0,l-1]} \frac{P(v_a(j)=v_z(j))}{P(v_{a|i}(j)=v_z(j))}$$

$$= \frac{P(f\text{-RR}(v_a(i))=v_z(i))}{P(f\text{-RR}(v_{a|i}(i))=v_z(i))} \leq \exp(\frac{\epsilon_i|\mathcal{E}(f\text{-RR}(v_a,i))-\mathcal{E}(f\text{-RR}(v_{a|i},i))|}{\Delta_i}) \qquad (6)$$

However, finding a closed-form solution of $\alpha$ for the tight privacy bound in Eq. 6 is non-trivial, since $\epsilon_i$ is intractable. To address this, we consider two cases: (1) $\frac{P(f\text{-RR}(v_a(i))=v_z(i))}{P(f\text{-RR}(v_{a|i}(i))=v_z(i))} \geq 1$ below, and (2) $0 < \frac{P(f\text{-RR}(v_a(i))=v_z(i))}{P(f\text{-RR}(v_{a|i}(i))=v_z(i))} < 1$ in Appendix D. Since $\Delta_i$ captures the magnitude by which the bit $i$ can change the decoding function $\mathcal{E}(\cdot)$ in the worst case, we have $\frac{\Delta_i}{|\mathcal{E}(f\text{-RR}(v_a,i))-\mathcal{E}(f\text{-RR}(v_{a|i},i))|} \geq 1$. As a result, we have

$$\frac{P(f\text{-RR}(v_a(i))=v_z(i))}{P(f\text{-RR}(v_{a|i}(i))=v_z(i))} \leq \left(\frac{P(f\text{-RR}(v_a(i))=v_z(i))}{P(f\text{-RR}(v_{a|i}(i))=v_z(i))}\right)^{\frac{\Delta_i}{|\mathcal{E}(f\text{-RR}(v_a,i))-\mathcal{E}(f\text{-RR}(v_{a|i},i))|}} \leq \exp(\epsilon_i) \qquad (7)$$

Eq. 7 enables us to quantify a *generalized privacy loss bound* of a RR mechanism, in which different bits have different sensitivities by randomizing all the bits in $v_x$ independently in Theorem 1.

**Theorem 1.** *Generalized privacy loss bound. The privacy loss for randomizing a binary encoding vector $v_x$ is bounded as follows:*

$$\frac{P(f\text{-}RR(v_x)=v_z)}{P(f\text{-}RR(\widetilde{v}_x)=v_z)} \leq \prod_{i=0}^{rl-1} \left(\frac{P(f\text{-}RR(v_x(i))=v_z(i))}{P(f\text{-}RR(v_{x|i}(i))=v_z(i))}\right)^{\frac{\Delta_i}{|\mathcal{E}(f\text{-}RR(v_x,i))-\mathcal{E}(f\text{-}RR(v_{x|i},i))|}} \leq \exp(\sum_{i=0}^{rl-1}\epsilon_i) \qquad (8)$$

Now, we enforce the condition $\sum_{i=0}^{rl-1}\epsilon_i = \epsilon_X$ to ensure that the total privacy budget will be bounded by $\epsilon_X$. Based upon that, we derive a closed-form solution showing that there always exists an upper bound of $\alpha$ so that $v'_x$ preserves $\epsilon_X$-LDP given $v_x$ in **Theorem 2** (Alg. 1, line 17).

**Theorem 2.** $\forall\alpha : 0 < \alpha \leq \sqrt{\frac{\epsilon_X+rl}{2r\sum_{i=0}^{l-1}\exp(2\frac{\epsilon_X}{l}i\%l)}}$, *the $f$-RR mechanism satisfies $\epsilon_X$-LDP:* $\frac{P(f\text{-}RR(v_x)=z)}{P(f\text{-}RR(\widetilde{v}_x)=z)} < \epsilon_X$, *where $v_x$ and $\widetilde{v}_x$ can be different at any bit, and $z \in Range(f\text{-}RR)$.*

Regarding to the ground-truth label $y_x$, given a privacy budget $\epsilon_Y$, we show that there always exists an upper bound of $\beta$ so that $y'_x$ preserves $\epsilon_Y$-LDP given $y_x$.

**Theorem 3.** $\forall\beta : \beta \leq \epsilon_Y - \ln(C-1)$, *the label-RR mechanism (Alg. 1, line 18) satisfies $\epsilon_Y$-LDP:* $\frac{P(label\text{-}RR(y_x)=z|y_x)}{P(label\text{-}RR(\widetilde{y}_x)=z|\widetilde{y}_x)} \leq \exp(\epsilon_Y)$, *given any distinct labels $y_x$ and $\widetilde{y}_x$, and $z \in Range(label\text{-}RR)$.*

From Theorems 2 and 3, our BitRand preserves $\epsilon_X$-LDP for the vector $v_x$ and $\epsilon_Y$-LDP for label $y_x$. The gradients $\nabla_{\theta_t}^u$ preserve $(\epsilon_X, \epsilon_Y)$-LDP in any training rounds $t$ for any training samples $(x, y_x) \in D_u$ and for all clients $u \in [1, N]$; since, $\forall u, t: \nabla_{\theta_t}^u$ are computed from the randomized training samples $\{(v'_x, y'_x)\}_{n_u}$ without accessing any further information from $\{(x, y_x)\}_{n_u}$.

## 5 PRIVACY AND UTILITY TRADE-OFF

As shown in Theorem 1, we can derive a tighter privacy loss bound. Therefore, we focus on understanding how BitRand can address the privacy-utility trade-off in comparison with existing approaches by theoretically studying: **(1)** the utility of $f$-RR regarding the expected error bound (**Theorem 4**), and **(2)** the trade-off between privacy budget and randomization probabilities. *All statistical tests are 2-tail t-tests.*

**Expected Error Bounds.** We analyze the data utility through *expected error*, denoted as $\xi_a$, measuring the

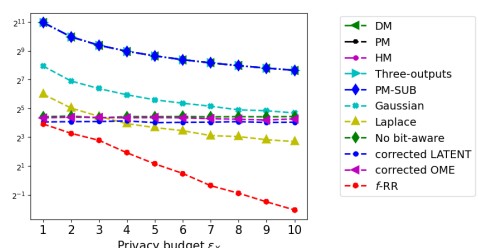

Figure 3: Expected error bound comparison for an embedded feature $a$ with $r = 1,000$, $l = 10$, and $m = 5$.

expected change of each embedded feature $a$ after applying $f$-RR: $\xi_a = \mathbb{E}|\mathcal{E}(f\text{-}RR(v_a)) - \mathcal{E}(v_a)|$. The *smaller expected error* is, the *better data utility* the randomization mechanism achieves. The expected error $\xi_a$ is bounded by $\sum_{i \in [0, l-1]} q_{Xi} \times \Delta_i$.

**Theorem 4.** *The $f$-RR expected error bound is quantified by $\xi_a = \mathbb{E}|\mathcal{E}(f\text{-}RR(v_a)) - \mathcal{E}(v_a)| = \sum_{i \in [0, l-1]} q_{Xi} \times \Delta_i$.*

Theorem 4 can be directly applied to quantify the expected error bounds of $f$-RR without the bit-aware term $i\%l/l$, corrected LATENT (Appendix I), and corrected OME (Appendix J). Regarding existing mechanisms applied on the embedded feature $a$, including Duchi mechanism (DM) (Duchi et al., 2013), Piecewise mechanism (PM) (Wang et al., 2019a), Hybrid mechanism (HM) (Wang et al., 2019a), Suboptimal mechanism (PM-SUB) (Zhao et al., 2020b), Gaussian and Laplace (Dwork & Roth, 2014), we derive a general form of expected error bounds $\xi_a$ for these mechanisms as: $\xi_a = \mathbb{E}|\mathcal{M}(a) - a| \approx 1/r \sum_{a \in [1,r]} |\mathcal{M}(a) - a|$ where $\mathcal{M}$ is an $\epsilon_X$-LDP preserving mechanism, since $\lim_{r \to \infty} \mathbb{E}|\mathcal{M}(a) - a| = 1/r \sum_{a \in [1,r]} |\mathcal{M}(a) - a|$.

Figure 3 illustrates the expected error bound of each algorithm as a function of $\epsilon_X$. It is obvious that $f$-RR has significantly tighter expected error bounds compared with baseline approaches under a wide range of privacy budgets given reasonable large numbers of embedded features and encoding bits ($p = 0.02$); thus indicating that our mechanism achieves better data utility compared with the baselines. The improvement of our $f$-RR over existing mechanisms is larger when $\epsilon_X$ increases.

To profoundly study the data utility, we take a depth look into bit-level analysis by relaxing Theorem 4 into **(1)** an expected error bound $\xi_i = q_{Xi} \times \Delta_i$ for each bit $i \in [0, l-1]$, and **(2)** an average top-$k$ expected error bound $\xi_{top-k} = 1/k \sum_{i=0}^{k} \xi_i, \forall k \in [0, l-1]$. Figure 11 (Appendix K) show that, at the bit-level, $f$-RR achieves smaller values of $\xi_i$ for most important bits, especially the sign bit and integer bits, and comparable $\xi_i$ for least important bits, i.e., fraction bits. The gap between $f$-RR and the baselines are significantly larger given the $\xi_{top-k}$. We obtain smaller values of $\xi_{top-k}$ for all $k \in [0, l-1]$, under a tight privacy budget $\epsilon_X = 0.1$. Importantly, when increasing privacy budget $\epsilon_X$ (Figures 11b,c), the gap between $f$-RR and the baselines is larger. In fact, the expected error bounds in $f$-RR is reduced while the expected error bounds in the baselines are remained the same.

**Privacy budget and randomization probabilities trade-off.** Compared with existing RR mechanisms (Lyu et al., 2020a; Arachchige et al., 2019; Sun et al., 2021; Zhao et al., 2020b; Duchi & Rogers, 2019; Wang et al., 2019a; Liu et al., 2020), in our algorithm, bits with a *stronger influence* on the model utility, e.g., sign bits and integer bits, have *smaller randomization probabilities* $q_X$, and vice-versa. This is because we consider the influence of each bit $i$ through the term $\frac{i\%l}{l}$ in modeling the randomization probabilities $q_X$ (and $p_X$). This unique property of our algorithm enables us to better optimize the trade-off between privacy loss and model utility.

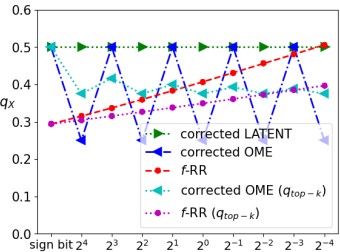

Figure 4: Randomization probability $q_X$ and $q_{top-k}$, given $l = 10$, $r = 1,000$, and $\epsilon_X = 1$.

To demonstrate this, we examine the behavior of $q_X$ across binary encoding bits under a wide range of $\epsilon_X \in \{0.1, 1, 2\}$, given reasonable values of $r$ and $l$, i.e., $r = 1,000$ and $l = 10$ (Figures 4 and 12, Appendix K). Our mechanism achieves a smaller randomization probability $q_X$ than the corrected LATENT in all cases ($p = 3.7e - 9$), especially more significant bits, such as sign bits and integer bits. The gap is more prominent when $\epsilon_X$ increases. This observation is less obvious when comparing our mechanism with the corrected OME, given an uneven randomization probability $q_X$ across bits in the corrected OME. To better show the comparison, we draw an average top-$k$ measure curve: $\forall k \in [0, l-1] : q_{top-k} = 1/k \sum_{i=0}^{k} q_i$, where $q_i$ is the bit $i$'s $q_X$, to evaluate the average $q_X$ across bits. The smaller $q_{top-k}$ is, the better the randomization probability $q_X$ is. Given a tight privacy budget $\epsilon_X = 0.1$, our mechanism and the corrected OME have a similar $q_{top-k}$. However, when $\epsilon_X$ increases, i.e., $\epsilon_X \in \{1, 2\}$, our mechanism achieves significantly smaller values of $q_{top-k}$ than the corrected OME ($p = 5.1e - 3$).

## 6 DIMENSION-ELASTIC ANALYSIS

In existing RR mechanisms, the curse of privacy composition is rooted in the privacy composition across bits $l$ and features $r$ (Duchi et al., 2013; Lyu et al., 2020a; Arachchige et al., 2019), gradients $\nabla_{\theta_t}^u$ (Zhao et al., 2020b; Wang et al., 2019a), and training rounds $T$ (Zhao et al., 2020b; Wang et al.,

2019a). Thus, increasing the dimensions of $l$, $r$, $\bigtriangledown_{\theta_t}^u$, and $T$ either significantly increases the privacy budget (i.e., resulting in loose privacy protection) (Wang et al., 2019a; 2017) or notably affects the randomization probabilities (i.e., the value transmitted correctly through the randomization becomes smaller resulting in poor utility (Wang et al., 2016a)) (Lyu et al., 2020a; Arachchige et al., 2019).

In Theorem 1, the privacy composition across bits and features is unavoidable. However, the values of embedded features transmitted correctly through our randomization mechanism is minimally affected when increasing the dimensions of $r$, $l$, $\bigtriangledown_{\theta_t}^u$, $T$, and $C$ under the same $\epsilon_X$ and $\epsilon_Y$. That enable us to evade the curse of privacy composition when working with complex models and tasks under rigorous LDP protection. To shed light into this property, we conduct a theoretical analysis to examine: **(1)** How the dimensions of $r$, $l$, gradients $\bigtriangledown_{\theta_t}^u$, and training rounds $T$ impact the privacy budget $\epsilon_X$ and the randomization probabilities $q_X$ and $p_X$ ($= 1 - q_X$); and **(2)** How the number of model outcomes $C$ impacts the privacy budget $\epsilon_Y$ and the randomization probabilities $q_Y$ and $p_Y$. We select the upper bounds of $\alpha = \sqrt{(\epsilon_X + rl)/(2r\sum_{i=0}^{l-1}\exp(2\epsilon_X \frac{i}{l}))}$ and $\beta = \epsilon_Y - \ln(C-1)$ (Theorems 2, 3) in our analysis.

**Dimensions of gradients $\bigtriangledown_{\theta_t}^u$ and training rounds $T$.** In Bit-Rand, the clients only use the perturbed samples $(v_x', y_x')$ to train their local models without accessing any further information from $(x, y_x)$. Thus, the privacy budgets $\epsilon_X$ and $\epsilon_Y$ are independent of the dimension of gradients $\bigtriangledown_{\theta_t}^u$ and the training rounds $T$, i.e., following the post-processing property in DP (Dwork & Roth, 2014). Also, the size of $\bigtriangledown_{\theta_t}^u$ and $T$ do not affect the randomization probabilities, since $\bigtriangledown_{\theta_t}^u$ and $T$ are not used to model $q_X$, $p_X$, $q_Y$, and $p_Y$ as in Eqs. 2 and 3.

**Dimensions of embedded features $r$ and encoding bits $l$.** Varying the dimensions of $r$ and $l$ does not affect the privacy budget $\epsilon_X \in \mathbb{R}^+$, since there always exists an $\alpha$ for Theorem 2 to hold. However, it is necessary to understand the influence of varying $r$ and $l$ on the privacy-utility trade-off. We theoretically analyze the impacts of $r$ and $l$ on the randomization probabilities $q_X$ and $p_X$, given fixed values of $\epsilon_X$. A model is expected to achieve higher model utility given smaller values of $q_X$ (higher values of $p_X$) under the same privacy budget. We examine $q_X$ and $p_X$ in the following experiments: **(i)** Fixing $l$, then varying $r$; and **(ii)** Fixing $r$, then varying $l$.

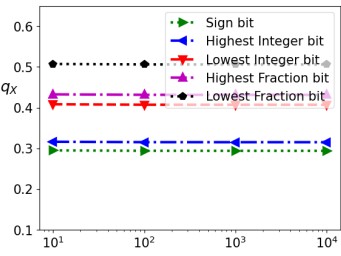

Figure 5: Randomization probability $q_X$ as a function of $r$ with fixed $l = 10$ and $\epsilon_X = 1$.

Figures 5 and 13 (Appendix K) illustrate $q_X$ as a function of $r$, under $l \in \{5, 20, 100, 1{,}000\}$ and $\epsilon_X \in \{0.1, 1, 2\}$. Varying $r$ does not affect the randomization probability $q_X$ (and $p_X$) for all the bits in $v_x$. To explain this, we take a deeper look into the $\alpha$'s bound, which is the only factor affecting $q_X$ and $p_X$. Given fixed $\epsilon_X$ and $l$, $\sum_{i=0}^{l-1}\exp(2\epsilon_X \frac{i}{l})$ is a constant, denoted $\mathcal{C}$. So, we have $\alpha = \sqrt{(\epsilon_X + rl)/(2r\mathcal{C})} = \sqrt{(\frac{\epsilon_X}{r} + l)/(2\mathcal{C})} \cong \sqrt{l/(2\mathcal{C})}$, since $\frac{\epsilon_X}{r} \cong 0.0$ in practice. Thus, with fixed $\epsilon_X$ and $l$, $q_X$ and $p_X$ are $r$-elastic, given $\alpha$ approximately is a constant $\sqrt{l/(2\mathcal{C})}$.

Now, we fix $r = 1{,}000$, which is a decent number of embedded features, and show $q_X$ as a function of $l$ under a wide range of $r \in [10, 10{,}000]$ and $\epsilon_X \in \{0.1, 1, 2\}$ (Figures 6 and 14, Appendix K). Given a tight privacy budget $\epsilon_X = 0.1$, varying $l$ does not affect the randomization probability $q_X$, i.e., $m = \lfloor l/2 \rfloor$ in our analysis covering most values of embedded features in practice. However, with higher values of $\epsilon_X \in \{1.0, 2.0\}$, using more encoding bits $l$ lowers $q_X$ for most important bits, i.e., sign and integer bits; while increasing $q_X$ for least important bits, i.e., fraction bits. When $l$ is large enough ($l \geq 20$), the randomization probability $q_X$ is $l$-elastic since the impact of $l$ becomes marginal to all the bits. This is also true for $p_X$.

Figure 6: Randomization probability $q_X$ as a function of $l$ with fixed $r = 1{,}000$ and $\epsilon_X = 1$.

**Number of model outcomes $C$.** Similar to $\epsilon_X$, from Eq. 3 and Theorem 3, the privacy budget $\epsilon_Y$ is not directly affected by the number of model outcomes $C$, since $\forall C, \epsilon_X$: $\exists \beta$ for Theorem 3 to hold. However, $C$ may impact the trade-off between privacy and model utility by affecting the randomization probabilities $q_Y$ and $p_Y$. Figure 7 shows that, when $C$ is sufficiently large, i.e., $C \geq 100$, the randomization probability $q_Y$ is $C$-elastic since the impact of $C$ on $q_Y$ is marginal given a wide range of the privacy budget $\epsilon_Y$. This is also true for $p_Y$. When $\epsilon_Y$ increases, the randomization probability $q_Y$ becomes smaller ($p_Y$ becomes larger). This is a reasonable observation.

Thanks to the bit-aware and dimension-elastic properties, our BitRand can achieve high data utility, especially under tight privacy budgets and expected error bounds.

## 7 EXPERIMENTAL RESULTS

We have conducted an extensive experiment on benchmark datasets under two fundamental FL tasks, text and image classification, to shed light on understanding 1) the interplay among privacy budget and model utility in BitRand, 2) the effectiveness of the dimension-elastic and bit-aware properties on model utility, and 3) different settings of applying RR to preserve LDP.

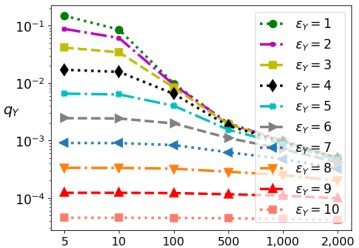

Figure 7: Randomization probability $q_Y$ with varying $\epsilon_Y$ and $C$.

**Baseline Approaches.** We consider a variety of LDP-preserving mechanisms as baseline approaches: **(1)** Binary encoding approaches, including the **corrected LATENT** (Arachchige et al., 2019) and the **corrected OME** (Lyu et al., 2020a) (Appendices I and J); **(2) LDP-FL** (Sun et al., 2021); **(3)** Duchi mechanism (**DM**) (Duchi et al., 2013); **(4)** Piecewise mechanism (**PM**) (Wang et al., 2019a); **(5)** Hybrid mechanism (**HM**) (Wang et al., 2019a); **(6) Three-Outputs** mechanism (Zhao et al., 2020b); **(7)** Suboptimal mechanism (**PM-SUB**) (Zhao et al., 2020b); and **(8) Label-Laplace** (Phan et al., 2020). Each baseline approach is applied to randomize (when applicable): **(i)** Embedded features $e_x$; **(ii)** Gradients $\bigtriangledown_{\theta_t}^u$; and **(iii)** Gradients $\bigtriangledown_{\theta_t}^u$ with a recent anonymizer (Sun et al., 2021) to reduce the privacy budget consumption. In addition, Label-Laplace is used as a baseline to protect the ground-truth labels $y_x$. Note that, in our experiment, we use the upper bound of $\beta$ resulting in the same randomizing probabilities as in LabelDP. Therefore, we do not include LabelDP in comparison. More details about the difference of $label$-RR and LabelDP are in Appendix G. These settings are widely accepted to preserve LDP in FL; thus, offering a comprehensive view of preserving LDP in FL. Note that LATENT, OME, and BitRand can only be applied on embedded features $e_x$; while LDP-FL can only be applied on gradients $\bigtriangledown_{\theta_t}^u$ with and without the anonymizer (Sun et al., 2021). We include the **Noiseless** FL model trained on the original data $D_u$ to show upper-bounds and a **Random** guess model to understanding model utility better.

**Datasets, Metrics, and Model Configuration.** The complete details of the datasets, metrics, and model configuration are in Appendix L. We carried out our experiments on two textual datasets and two image datasets, including the AG dataset (Gulli et al., 2012), our collected Security and Exchange Commission (SEC) financial contract dataset, the large-scale celebFaces attributes (CelebA) dataset (Liu et al., 2015), and the Federated Extended MNIST (FEMNIST) dataset (Caldas et al., 2018). We use the test accuracy and the test area under the curve (AUC) as evaluation metrics. Models with higher values of test accuracy and AUC are better. We use the BERT-Base (Uncased) pre-trained model (ber; Devlin et al., 2018) to extract embedded features in the AG and SEC datasets. In the CelebA and FEMNIST datasets, we use the ResNet-18 pre-trained model (img; He et al., 2016). For text and image classification tasks, we use two fully connected layers on top of embedded features, each of which consists of $1,500$ hidden neurons and uses a ReLU activation function.

**Evaluation Results.** Comprehensive results show that BitRand offers stronger privacy protection with higher model utility, compared with all baseline approaches, as discussed next.

**LDP-preserving approaches applied on the embedded features $e_x$.** Baseline approaches do not work well when they are applied on embedded features $e_x$ (Figures 15 and 19, Table 3, Appendix L). In SEC, AG, and FEMNIST datasets, BitRand achieves the highest model utility compared with the best baseline approach, which is the corrected OME, under a tight privacy budget $\epsilon_X = 1$. In terms of accuracy and AUC values, BitRand ($\epsilon_Y = \infty$) achieves an improvement of $46.03\%$ and $38.51\%$ in the AG dataset ($p = 2.7e - 22$), $13.69\%$ and $13.79\%$ in the SEC dataset ($p = 4.1e - 12$), and $21.62\%$ and $13.42\%$ in the FEMNIST dataset ($p = 5.6e - 11$) respectively. In the CelebA dataset (Table 3, Appendix L), BitRand outperforms the best baseline approach, i.e., PM-SUB, with an average improvement of $1.66\%$ across all 40 attributes in terms of AUC measure ($p = 1.2e - 2$). Since the CelebA dataset is highly imbalanced, we use the AUC measure instead of the model accuracy. The gaps between BitRand and the baseline approaches are significantly wider when the privacy budget $\epsilon_X$ is larger. In addition to $\epsilon_X = 1$, with a tight privacy budget for the class labels $\epsilon_Y \in \{1, 2.5\}$, BitRand still outperforms baseline approaches in most of cases, offering stronger privacy protection with better model utility, i.e., LDP at both embedded feature and label levels instead of only LDP on the embedded features as in baseline approaches.

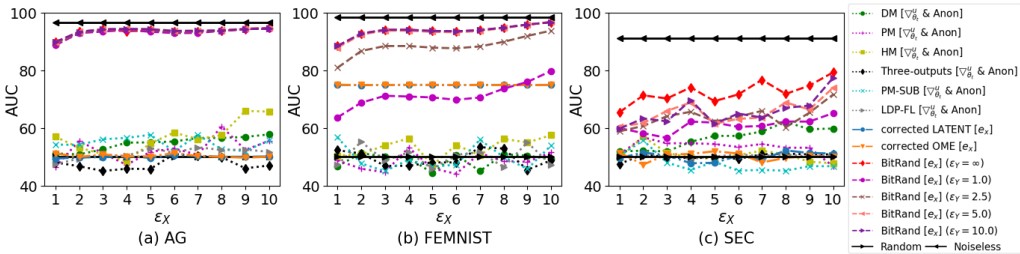

Figure 8: AUC values of each algorithm applied on the gradients $\triangledown^u_{\theta_t}$ with the anonymizer.

The key reason is that, in the baseline approaches, the model utility is significantly affected by the size of the embedded features. Thanks to the dimension-elastic and bit-aware properties, BitRand can achieve high model accuracy and AUC values, especially under tight privacy budgets. In addition, BitRand achieves the highest improvement in the AG dataset, since it is a balanced dataset compared with the highly imbalanced CelebA dataset, in which BitRand achieves the least improvement. Addressing imbalanced data in FL under DP (Huang et al., 2020) is out-of-scope of our study.

**LDP-preserving approaches applied on on gradients $\triangledown^u_{\theta_t}$ without and with the anonymizer (Sun et al., 2021).** We observe the same phenomenon when baseline approaches are applied on gradients $\triangledown^u_{\theta_t}$ without and with the anonymizer (Sun et al., 2021), even though the gaps between BitRand and baseline approaches are (marginally) smaller (Figures 8, 16-17, 20, and Tables 4, 5, Appendix L). Without using the anonymizer in the baseline approaches, in terms of accuracy and AUC, compared with the best baseline approach PM-SUB, BitRand ($\epsilon_X = 1$, $\epsilon_Y = 1$) achieves an improvement of $44.95\%$ and $37.52\%$ in the AG dataset ($p = 3.9e - 20$), $12.82\%$ and $12.92\%$ in the SEC dataset ($p = 1.3e - 11$), $24.17\%$ and $13.40\%$ in the FEMNIST dataset ($p = 4.1e - 11$). When the anonymizer is applied in the baseline approaches, in terms of accuracy and AUC, compared with the best baseline approache HM, our BitRand achieves an improvement of $39.38\%$ and $32.75\%$ in the AG dataset ($p = 1.2e - 16$), $13.59\%$ and $13.69\%$ in the SEC dataset ($p = 2.1e - 10$), and $23.12\%$ and $37.02\%$ in the FEMNIST dataset ($p = 2.8e - 11$). Regarding the CelebA dataset, BitRand outperforms the best baseline approaches, which are Three-outputs and HM, with improvements of $1.28\%$ and $0.3\%$, in the cases of with and without the anonymizer, across all $40$ attributes in terms of AUC ($p = 3.1e-2$). The model utility in baselines is affected by the size of gradients and the training rounds (when the anonymizer is not applied) and their finite numbers of randomization outputs of the gradients (Zhao et al., 2020b). Thanks to the bit-aware and dimension-agnostic properties, BitRand can achieve high model utility under rigorous LDP protection.

**LDP-preserving labels.** Figures 18, 21, and Table 3 (Appendix L) present the model utility of BitRand as a function of the privacy budgets $\epsilon_X$ and $\epsilon_Y$, in which $label$-RR is replaced by the Label-Laplace, denoted as $f$-RR & Label-Laplace. $label$-RR outperforms the Label-Laplace in all values of $\epsilon_X$, $\epsilon_Y$, and datasets. Under rigorous LDP protection $\epsilon_Y = 1$ given $\epsilon_X \in [1, 10]$, there is an average improvement of $9.65\%$ accuracy and $7.92\%$ AUC in the AG dataset ($7.3e - 6$), $7.84\%$ accuracy and $7.35\%$ AUC in the SEC dataset ($1.2e - 5$), and $9.91\%$ accuracy and $10.55\%$ AUC in the FEMNIST dataset ($p = 9.8e - 5$), and $2.04\%$ AUC in the CelebA dataset ($p = 1.6e - 2$). The gaps are delicately smaller in the AG and SEC datasets, and substantially larger in the FEMNIST and CelebA datasets when $\epsilon_Y$ is increased. Given $\epsilon_Y = 2.5$, there is an average improvement of $1.71\%$ accuracy and $1.37\%$ AUC in the AG dataset ($8.5e - 2$), $4.65\%$ accuracy and $4.73\%$ AUC in the SEC dataset ($6.9e - 3$), and $14.43\%$ accuracy and $12.02\%$ AUC in the FEMNIST dataset ($p = 1.9e - 4$), and $4.24\%$ AUC in the CelebA dataset ($p = 2.3e - 2$). The reason is that Label-Laplace injects Laplace noise across $C$ classes in a label $y_x$ causing more noisy model outcomes compared with $label$-RR, in which only one of the $C$ model outcomes is selected as the result of the randomization.

## 8 CONCLUSION

In this paper, we introduced a bit-aware algorithm, called BitRand, providing rigorous LDP protection to both embedded features and labels in FL via binary encoding. In BitRand, the trade-off between the privacy budget consumption and randomization probabilities is dimension-elastic to the numbers of embedded features, encoding bits, gradients, model outcomes, and training rounds, enabling us to work with complex models and FL tasks. We further optimize the randomization probabilities by having smaller randomization probabilities assigned to more critical bits and vice-versa under the same privacy budgets. Theoretical analysis and extensive experiments showed that our BitRand outperforms baseline approaches in text and image classification.

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

APPENDIX

A    REVISITING RANDOMIZED RESPONSE MECHANISMS FOR LDP

To preserve LDP given the client's input $x$, we can apply existing RR mechanisms (Wang et al., 2016b; Fanti et al., 2015; Bassily et al., 2017; Kim et al., 2018; Ren et al., 2018; Zheng et al., 2017; Wang & Xu, 2019; Zhao et al., 2019; Gursoy et al., 2019; Alvim et al., 2018; Xiong et al., 2019), such as unary encoding-based approaches (Wang et al., 2017; Erlingsson et al., 2014), hash-based approaches (Wang et al., 2017; Bassily & Smith, 2015; Acharya et al., 2019; Wang et al., 2019c), binary encoding-based approaches (Arachchige et al., 2019; Lyu et al., 2020a), etc. For instance, hash-based approaches such as those of Google RAPPOR (Erlingsson et al., 2014) and OLH (Wang et al., 2017) hash the client's input $x$ onto a bloom filter $B$ of size $k$ using $h$ hash functions. Then, for each client's input $x$ and a bit $i \in B$, RAPPOR creates a perturbed binary value $B'_i$ from $B_i$ with the following randomization probability:

$$B'_i = \begin{cases} 1, & \text{with probability } p/2 \\ 0, & \text{with probability } p/2 \\ B_i, & \text{with probability } 1-p \end{cases} \tag{9}$$

where $p$ is a hyper-parameter. This $B'$ is reused as the basis for all future analysis, learning, and reports on this distinct input $x$. This approach achieves $\epsilon_X$-LDP, where $\epsilon_X = 2h \ln((1-\frac{p}{2})/\frac{p}{2})$, given that the sensitivity of every bit $B_i$ is $\Delta_{B_i} = 1$ (Erlingsson et al., 2014).

To deal with numerical inputs, e.g., embedded features, generalized RR mechanisms such as Duchi (Duchi & Rogers, 2019; Bhowmick et al., 2018), Piece-wise (Wang et al., 2019a), Hybrid (Wang et al., 2019a), Three-outputs (Zhao et al., 2020b), Suboptimal (Zhao et al., 2020b), LDP-FL (Sun et al., 2021), LATENT (Arachchige et al., 2019), and OME (Lyu et al., 2020a) can be applied.

Asymmetric version of RAPPOR (e.g., (Wang et al., 2017)) designs different randomization probabilities for different inputs. The technique is well-applied in the context of frequency estimation and successfully reduce the communication cost from $O(d)$ to $O(\log n)$ ($d$ is data dimension and $n$ is the number of samples). However, simply applying the mechanism (Wang et al., 2017) does not optimize the model utility and the privacy-utility trade-off when working with machine learning or deep learning models.

Another line of work in LDP is Mironov (2012), which addresses the floating-point arithmetic in implementation of DP applications. The inconsistency between mathematical abstraction of Laplace mechanism with sampling "uniform" floating-point numbers can be exploited to carry out privacy attacks. Floating-point arithmetic is a leaky abstraction, which is ubiquitous in computer systems and is difficult to argue about formally and hard to get right in applications, including all the RR mechanisms.

However, different from the asymmetric version of RAPPOR and the floating-point arithmetic, our proposed $f$-RR mechanism focuses on mitigating the privacy-utility trade-off. To achieve that, besides the asymmetric nature of the randomization probabilities, our designed $f$-RR consists of two key components: 1) The bit-aware term $i\%l/l$, which indicates the location of the bit i in each embedded feature associated with the sensitivity of the bit at that location; and 2) The adjustable but bounded $\alpha$, which takes into account the correlation between privacy loss and the sensitivity of embedded features to mitigate the privacy-utility trade-off and the curse of privacy composition.

The bit-aware property refers to the bits with a more substantial influence on the model utility have smaller randomization probabilities, and vice-versa, under the same privacy protection. By incorporating sensitivities of binary encoding bits into a generalized privacy loss bound, we show that increasing the dimensions of embedded features $r$, encoding bits $l$, and model outcomes $C$ marginally affect the randomization probabilities in BitRand under the same privacy budget. This dimension-elastic property is crucial to mitigate the curse of privacy composition by retaining a high value of data transmitted correctly through our randomization given large dimensions of $r$, $l$, and $C$.

Besides the $f$-RR for protecting the data, we also include the label-RR for protecting the label in our proposed BitRand mechanism, that provides a complete protection for every data sample.

## B BitRand Algorithm Pseudo-Code

1: **Input**: Privacy budget $\epsilon_X$ and $\epsilon_Y$, number of training iterations $T$, learning rate $\eta_t$, binary encoding parameters ($l$ and $m$)
2: *At server side:*
3: Initialize model $\theta^0$
4: Send the pre-trained model $f^{pre}$ to clients
5: **for** $t \in T$ **do**
6:     Distribute model parameter $\theta^t$ to each client
7:     **for** each client $u$ **do**
8:         $\triangledown^u_{\theta_t} \leftarrow$ **Client-Update**$(\theta^t)$
9:     **end for**
10:     $\theta_{t+1} = \theta_t - \eta_t \sum_{u \in [1,N]} \triangledown^u_{\theta_t} / N$
11: **end for**
12: **Output:** $(\epsilon_X, \epsilon_Y)$-LDP $\theta$

13: *At client side* $u \in [1, N]$*:*
14: **for** each data sample $(x, y_x) \in D_u$ **do**
15:     **Extracting** embedded features: $e_x \leftarrow f^{pre}(x)$
16:     $v_x \leftarrow BinaryEncoding(e_x)$ # using Eq. 1
17:     **Randomizing** $v_x$: $v'_x \leftarrow f$-RR$(v_x)$ with
$$\alpha = \sqrt{\frac{\epsilon_X + rl}{2r \sum_{i=0}^{l-1} \exp(2\epsilon_X \frac{i}{l})}}$$ # using Eq. 2
18:     **Randomizing** the label $y_x$: $y'_x \leftarrow label$-RR$(y_x)$ with $\beta = \epsilon_Y - \ln(C-1)$ # using Eq. 3
19: **end for**
20: **Client-Update**$(\theta^t)$:
21:     $\triangledown^u_{\theta_t} = \sum_{(v'_x, y'_x) \in D'_u} \triangledown_{\theta_t} \mathcal{L}(f(e'_x), y'_x) / n_u$
22:     return $\triangledown^u_{\theta_t}$

**Algorithm 1:** BitRand Algorithm in Federated Learning

## C Proof of Lemma 1

*Proof.* Without loss of generality, let us study the $l_1$-sensitivity of binary encoding bits of a feature's value $a$ in $e_x$. It is obvious that $f$-RR$(v_a, i)$ and $f$-RR$(v_{a|i}, i)$ differs at the bit $i$ in the worst case. The decoded feature of $f$-RR$(v_a, i)$ is $a' = \mathcal{E}(f\text{-RR}(v_a, i))$.

Let us denote $b_0 b_1 \ldots b_{l-1}$ as the binary representation $f$-RR$(v_a, i)$ of $a'$, where $b_0$ is the value of a sign bit (i.e., $b_0 = 1$ if $a' >= 0$ and $b_0 = 0$ if $a' < 0$), and $\{b_i\}_{i=1}^{l-1}$ is the value of integer bits and fraction bits. We have a decoding function: $\mathcal{E}(f\text{-RR}(v_a, i)) = (2b_0 - 1) \sum_{i=1}^{l-1} b_i \times 2^{m-i}$. This decoding function is also applicable to $\mathcal{E}(f\text{-RR}(v_{a|i}, i))$. Let us denote $\{b'_i\}_{i=0}^{l-1}$ as the value of the bit $i$ in $f$-RR$(v_{a|i}, i)$. Following Eq. 4, the $l_1$-sensitivity of the bit $b_i$ is computed as follows:

- If $i$ is an integer or a fraction bit ($i \in [1, l-1]$), we have: $\Delta_i = \max_{v_a} \| \mathcal{E}(f\text{-RR}(v_a, i)) - \mathcal{E}(f\text{-RR}(v_{a|i}, i)) \|_1 = \max_{v_a} \|(2b_0 - 1)\left(\sum_{j=1, j \neq i}^{l-1} b_j 2^{m-j} + b_i 2^{m-i}\right) - (2b_0 - 1)\left(\sum_{j=1, j \neq i}^{l-1} b_j 2^{m-j} + b'_i 2^{m-i}\right)\|_1 = \max_{v_a} \|(2b_0 - 1)(b_i - b'_i)2^{m-i}\|_1 \leq \max_{v_a}\left(|(2b_0 - 1)|\|b_i - b'_i\|_1 2^{m-i}\right)$. The $\Delta_i$ is maximized when $b_i$ and $b'_i$ are different, and $|2b_0 - 1| = 1$. Therefore, $\Delta_i = 2^{m-i}$.

- If $i$ is the sign bit ($i = 0$), we have: $\Delta_i = \max_{v_a} \| \mathcal{E}(f\text{-RR}(v_a, 0)) - \mathcal{E}(f\text{-RR}(v_{a|i}, 0)) \|_1 = \max_{v_a} \|(2b_0 - 1)\sum_{i=1}^{l-1} b_i 2^{m-i} - (2b'_0 - 1)\sum_{i=1}^{l-1} b_i 2^{m-i}\|_1 = \max_{v_a} \|(2b_0 - 2b'_0)\sum_{i=1}^{l-1} b_i 2^{m-i}\|_1 \leq \max_{v_a}\left(|2b_0 - 2b'_0|\|\sum_{i=1}^{l-1} b_i 2^{m-i}\|_1\right)$. The $\Delta_i$ is maximized when $b_0$ and $b'_0$ are different and all $\{b_i\}_{i=1}^{l-1} = 1$. Then, $\Delta_i \leq \max_{v_{a|0}} 2 \sum_{i=1}^{l-1} 2^{m-i}$. Since $2^1 + \ldots + 2^{l-2} = 2^{l-1} - 2$ (mat), we have: $2\sum_{i=1}^{l-1} 2^{m-i} = 2^{m+1-(l-1)}(2^{l-1} - 1) < 2^{m+1}$. As a result, $\Delta_i = 2^{m+1}$.

From the aforementioned $\Delta_i$ of a feature's value in $e_x$, it is easily expanded to the entire $e_x$. Since $e_x$ is the concatenation of all $l$ bits of $r$ values in $e_x$, all bits with the same value of $i\%l$ have the same $l_1$-sensitivity across $rl$ bits. Therefore, the $\Delta_i$ of the sign bit and the integer/fraction bits are $2^{m+1}$ and $2^{m-i\%l}$, respectively. Consequently, Lemma 1 hold. $\square$

## D Proof of Theorem 1

Let us consider a bit $i$ belonging to a feature $a$, i.e., one of the $r$ features. We denote $f$-RR$(v_a, i)$ as vector $v_a$ with only the bit $i$ randomized by our $f$-RR, i.e., all the bits different from $i$ are kept the same in $v_a$; that is, we only protect the bit $i$ when using the notation $f$-RR$(v_a, i)$. Given the $l_1$-sensitivity $\Delta_i$, there always exists a Laplace noise injected into the embedded feature $a$, i.e.,

$\mathcal{M}(v_a, i) = \mathcal{E}(v_a) + Lap(\Delta_i/\epsilon_i)$, to achieve $\epsilon_i$-LDP in the embedded feature space (Dwork & Roth, 2014). In other words, $\frac{P(\mathcal{M}(v_a,i)=z)}{P(\mathcal{M}(v_{a|i},i)=z)} \leq \exp(\frac{\epsilon_i|\mathcal{E}(v_a)-\mathcal{E}(v_{a|i})|}{\Delta_i})$ where $v_{a|i}$ is the vector that differs from $v_a$ only at the bit $i$ and $z \in Range(\mathcal{M})$.

To ensure that the privacy loss in randomizing the bit $i$, we need to bound $\alpha$ in Eq. 2 such that $\frac{P(f\text{-RR}(v_a,i)=v_z)}{P(f\text{-RR}(v_{a|i},i)=v_z)} \leq \exp(\frac{\epsilon_i|\mathcal{E}(v_a)-\mathcal{E}(v_{a|i})|}{\Delta_i})$. However, randomizing a binary encoding bit $i$ given $v_a$ and $v_{a|i}$ results in a smaller $l_1$-distance $|\mathcal{E}(f\text{-RR}(v_a,i)) - \mathcal{E}(f\text{-RR}(v_{a|i},i))|$ compared with $|\mathcal{E}(v_a) - \mathcal{E}(v_{a|i})|$; since $|\mathcal{E}(f\text{-RR}(v_a,i)) - \mathcal{E}(f\text{-RR}(v_{a|i},i))| \leq |\mathcal{E}(v_a) - \mathcal{E}(v_{a|i})|$. Thus, we can derive a tighter bound by replacing $|\mathcal{E}(v_a) - \mathcal{E}(v_{a|i})|$ with $|\mathcal{E}(f\text{-RR}(v_a,i)) - \mathcal{E}(f\text{-RR}(v_{a|i},i))|$. Given $v_a(i)$ is the bit $i$ in $v_a$, we have

$$\frac{P(f\text{-RR}(v_a,i)=v_z)}{P(f\text{-RR}(v_{a|i},i)=v_z)} = \frac{P(f\text{-RR}(v_a(i))=v_z(i))}{P(f\text{-RR}(v_{a|i}(i))=v_z(i))} \times \prod_{j \neq i, j \in [0,l-1]} \frac{P(v_a(j)=v_z(j))}{P(v_{a|i}(j)=v_z(j))}$$

$$= \frac{P(f\text{-RR}(v_a(i))=v_z(i))}{P(f\text{-RR}(v_{a|i}(i))=v_z(i))} \leq \exp(\frac{\epsilon_i|\mathcal{E}(f\text{-RR}(v_a,i)) - \mathcal{E}(f\text{-RR}(v_{a|i},i))|}{\Delta_i}) \quad (10)$$

However, finding a closed-form solution of $\alpha$ for the tight privacy bound in Eq. 10 is non-trivial, since $\epsilon_i$ is intractable. To address this problem, we consider two cases: (1) $\frac{P(f\text{-RR}(v_a(i))=v_z(i))}{P(f\text{-RR}(v_{a|i}(i))=v_z(i))} \geq 1$ and (2) $0 < \frac{P(f\text{-RR}(v_a(i))=v_z(i))}{P(f\text{-RR}(v_{a|i}(i))=v_z(i))} < 1$. Also, since $\Delta_i$ captures the magnitude by which a bit $i$ can change the decoding function $\mathcal{E}()$ in the worst case, we have $\frac{\Delta_i}{|\mathcal{E}(f\text{-RR}(v_a,i))-\mathcal{E}(f\text{-RR}(v_{a|i},i))|} \geq 1$.

In the first case, the privacy loss for a bit $i$ can be bounded as follows:

$$\frac{P(f\text{-RR}(v_a(i))=v_z(i))}{P(f\text{-RR}(v_{a|i}(i))=v_z(i))} \leq \left(\frac{P(f\text{-RR}(v_a(i))=v_z(i))}{P(f\text{-RR}(v_{a|i}(i))=v_z(i))}\right)^{\frac{\Delta_i}{|\mathcal{E}(f\text{-RR}(v_a,i))-\mathcal{E}(f\text{-RR}(v_{a|i},i))|}} \leq \exp(\epsilon_i) \quad (11)$$

Since the RR mechanism is independently applied on each bit $i$ in $v_a$ and on every $v_a$ of the $r$ features, Eq. 11 enables us to quantify a *generalized privacy loss bound* of a RR mechanism, in which different bits have different sensitivities to the randomized outcome as follows.

$$\frac{P(f\text{-RR}(v_x)=v_z)}{P(f\text{-RR}(\widetilde{v}_x)=v_z)} = \prod_{i=0}^{rl-1} \frac{P(f\text{-RR}(v_x(i))=v_z(i))}{P(f\text{-RR}(v_{x|i}(i))=v_z(i))}$$

$$\leq \prod_{i=0}^{rl-1} \left(\frac{P(f\text{-RR}(v_x(i))=v_z(i))}{P(f\text{-RR}(v_{x|i}(i))=v_z(i))}\right)^{\frac{\Delta_i}{|\mathcal{E}(f\text{-RR}(v_x,i))-\mathcal{E}(f\text{-RR}(v_{x|i},i))|}} \leq \prod_{i=0}^{rl-1} \exp(\epsilon_i) = \exp(\sum_{i=0}^{rl-1} \epsilon_i)$$

$$(12)$$

In the second case, the privacy loss for a bit $i$ can be bounded as follows:

$$\frac{P(f\text{-RR}(v_a(i))=v_z(i))}{P(f\text{-RR}(v_{a|i}(i))=v_z(i))} \leq \frac{P(f\text{-RR}(v_{a|i}(i))=v_z(i))}{P(f\text{-RR}(v_a(i))=v_z(i))}$$

$$\leq \left(\frac{P(f\text{-RR}(v_{a|i}(i))=v_z(i))}{P(f\text{-RR}(v_a(i))=v_z(i))}\right)^{\frac{\Delta_i}{|\mathcal{E}(f\text{-RR}(v_a,i))-\mathcal{E}(f\text{-RR}(v_{a|i},i))|}} \leq \exp(\epsilon_i)$$

$$(13)$$

Similarly, we obtain the same result with Eq. 12 in the second case:

$$\frac{P(f\text{-RR}(v_x)=v_z)}{P(f\text{-RR}(\widetilde{v}_x)=v_z)} = \prod_{i=0}^{rl-1} \frac{P(f\text{-RR}(v_x(i))=v_z(i))}{P(f\text{-RR}(v_{x|i}(i))=v_z(i))} \leq \prod_{i=0}^{rl-1} \frac{P(f\text{-RR}(v_{x|i}(i))=v_z(i))}{P(f\text{-RR}(v_x(i))=v_z(i))}$$

$$\leq \prod_{i=0}^{rl-1} \left(\frac{P(f\text{-RR}(v_{x|i}(i))=v_z(i))}{P(f\text{-RR}(v_x(i))=v_z(i))}\right)^{\frac{\Delta_i}{|\mathcal{E}(f\text{-RR}(v_x,i))-\mathcal{E}(f\text{-RR}(v_{x|i},i))|}} \leq \prod_{i=0}^{rl-1} \exp(\epsilon_i) = \exp(\sum_{i=0}^{rl-1} \epsilon_i)$$

$$(14)$$

Consequently, Theorem 1 holds.

## E  PROOF OF THEOREM 2

To ensure that the privacy loss in randomizing the bit $i$ is bounded by the privacy loss in the embedded space and to take into account different sensitivities of different bits (Theorem 1), we need to find $\alpha$ in Eq. 2 to solve Eq. 12. However, solving Eq. 12 is not straightforward since the privacy budget $\epsilon_i$ and the actual $l_1$-distance $|\mathcal{E}(f\text{-RR}(v_x, i)) - \mathcal{E}(f\text{-RR}(v_{x|i}, i))|$ are intractable. To address this problem, from Eq. 12, we first consider the bit $i$ in all the $r$ embedded features, as follows: $\prod_{a \in [1,r]} \frac{P(f\text{-RR}(v_a(i)) = v_z(i))}{P(f\text{-RR}(v_{a|i}(i)) = v_z(i))} \leq \exp(\sum_{a \in [1,r]} \frac{\epsilon_i |\mathcal{E}(f\text{-RR}(v_a, i)) - \mathcal{E}(f\text{-RR}(v_{a|i}, i))|}{\Delta_i}) = \exp(\frac{\epsilon_i}{\Delta_i} \sum_{a \in [1,r]} |\mathcal{E}(f\text{-RR}(v_a, i)) - \mathcal{E}(f\text{-RR}(v_{a|i}, i))|)$. Note that all the bits $i$, e.g., sign bits, in all the features $a \in [1,r]$ consume the same privacy budget $\epsilon_i$ with the same sensitivity $\Delta_i$. The term $\sum_{a \in [1,r]} |\mathcal{E}(f\text{-RR}(v_a, i)) - \mathcal{E}(f\text{-RR}(v_{a|i}, i))|$ can be unbiasedly replaced with $r \times \mathbb{E}|\mathcal{E}(f\text{-RR}(v_a, i)) - \mathcal{E}(f\text{-RR}(v_{a|i}, i))|$ where $\mathbb{E}$ is the expectation of $|\mathcal{E}(f\text{-RR}(v_a, i)) - \mathcal{E}(f\text{-RR}(v_{a|i}, i))|$, since $\lim_{r \to \infty} \mathbb{E}|\mathcal{E}(f\text{-RR}(v_a, i)) - \mathcal{E}(f\text{-RR}(v_{a|i}, i))| = \frac{\sum_{a \in [1,r]} |\mathcal{E}(f\text{-RR}(v_a, i)) - \mathcal{E}(f\text{-RR}(v_{a|i}, i))|}{r}$. Hence, we have that

$$\prod_{a \in [1,r]} \frac{P(f\text{-RR}(v_a(i)) = v_z(i))}{P(f\text{-RR}(v_{a|i}(i)) = v_z(i))} \leq \exp(\frac{r\epsilon_i \times \mathbb{E}|\mathcal{E}(f\text{-RR}(v_a, i)) - \mathcal{E}(f\text{-RR}(v_{a|i}, i))|}{\Delta_i})$$

$$\Leftrightarrow \prod_{a \in [1,r]} \left(\frac{P(f\text{-RR}(v_a(i)) = v_z(i))}{P(f\text{-RR}(v_{a|i}(i)) = v_z(i))}\right)^{\frac{\Delta_i}{\mathbb{E}|\mathcal{E}(f\text{-RR}(v_a, i)) - \mathcal{E}(f\text{-RR}(v_{a|i}, i))|}} \leq \exp(r\epsilon_i) \tag{15}$$

Note that, in our work, we consider the worst case is the case that all the bits of two neighboring vectors can be different. The expectation $\mathbb{E}|\mathcal{E}(f\text{-RR}(v_a, i)) - \mathcal{E}(f\text{-RR}(v_{a|i}, i))|$ in Eq. 15 is used to quantify the difference of every two extreme vectors at bit $i$. Then for the whole vector, it is the sum over the expectation $\mathbb{E}(\cdot)$ of all bits $i$, as follows: $\sum_{a \in [1,r]} |\mathcal{E}(f\text{-RR}(v_a, i)) - \mathcal{E}(f\text{-RR}(v_{a|i}, i))| = r \times \mathbb{E}|\mathcal{E}(f\text{-RR}(v_a, i)) - \mathcal{E}(f\text{-RR}(v_{a|i}, i))|$. Therefore, the expectation in Eq. 15 does not imply the average-case scenario.

The sum over the expectation $\mathbb{E}(\cdot)$ of all bits $i$, i.e., $\sum_{a \in [1,r]} \sum_{i \in [0,l-1]} |\mathcal{E}(f\text{-RR}(v_a, i)) - \mathcal{E}(f\text{-RR}(v_{a|i}, i))| = r \times \sum_{i \in [0,l-1]} \mathbb{E}|\mathcal{E}(f\text{-RR}(v_a, i)) - \mathcal{E}(f\text{-RR}(v_{a|i}, i))|$, is used to bound the privacy loss as follows:

$$\frac{P(f\text{-RR}(v_x) = v_z)}{P(f\text{-RR}(\widetilde{v}_x) = v_z)} \leq \prod_{a \in [1,r]} \prod_{i \in [0,l-1]} \left(\frac{P(f\text{-RR}(v_a(i)) = v_z(i))}{P(f\text{-RR}(v_{a|i}(i)) = v_z(i))}\right)^{\frac{\Delta_i}{\mathbb{E}|\mathcal{E}(f\text{-RR}(v_a, i)) - \mathcal{E}(f\text{-RR}(v_{a|i}, i))|}}$$

$$\leq \exp\left(\sum_{a \in [1,r]} \sum_{i \in [0,l-1]} \frac{\epsilon_i |\mathcal{E}(f\text{-RR}(v_a, i)) - \mathcal{E}(f\text{-RR}(v_{a|i}, i))|}{\Delta_i}\right)$$

$$= \exp\left(\sum_{i \in [0,l-1]} \frac{\epsilon_i (r \times \mathbb{E}|\mathcal{E}(f\text{-RR}(v_a, i)) - \mathcal{E}(f\text{-RR}(v_{a|i}, i))|)}{\Delta_i}\right)$$

$$\leq \exp\left(r \sum_{i \in [0,l-1]} \epsilon_i\right)$$

$$\Leftrightarrow \prod_{i=0}^{rl-1} \left(\frac{P(f\text{-RR}(v_x(i)) = v_z(i))}{P(f\text{-RR}(v_{x|i}(i)) = v_z(i))}\right)^{\frac{\Delta_i}{\mathbb{E}|\mathcal{E}(f\text{-RR}(v_x, i)) - \mathcal{E}(f\text{-RR}(v_{x|i}, i))|}} \leq \exp\left(\sum_{i=0}^{rl-1} \epsilon_i\right) \tag{16}$$

Now, we need to bound the generalized privacy loss by discovering closed-form solutions of $\alpha$ given the privacy budgets $\epsilon_X$ (Theorem 2). In other words, we need to solve Eq. 16 for discovering the closed-form solution of $\alpha$. To solve it, first we need to calculate $\mathbb{E}|\mathcal{E}(f\text{-RR}(v_x, i)) - \mathcal{E}(f\text{-RR}(v_{x|i}, i))|$. To be more precise, let us denote $p_{Xi}$ and $q_{Xi}$ as $p_X$ and $q_X$ in Eq. 2 for a particular bit $i$, respectively. Given the worse case of $v_x$ and $v_{x|i}$, there are four possible cases of $|\mathcal{E}(f\text{-RR}(v_x, i)) - \mathcal{E}(f\text{-RR}(v_{x|i}, i))|$:

- If $f\text{-RR}(v_x(i)) = 1$ and $f\text{-RR}(v_{x|i}(i)) = 1$, then $|\mathcal{E}(f\text{-RR}(v_x, i)) - \mathcal{E}(f\text{-RR}(v_{x|i}, i))| = 0$.

- If $f\text{-RR}(v_x(i)) = 0$ and $f\text{-RR}(v_{x|i}(i)) = 0$, then $|\mathcal{E}(f\text{-RR}(v_x, i)) - \mathcal{E}(f\text{-RR}(v_{x|i}, i))| = 0$.

- If $f\text{-RR}(v_x(i)) = 1$ and $f\text{-RR}(v_{x|i}(i)) = 0$, then $|\mathcal{E}(f\text{-RR}(v_x, i)) - \mathcal{E}(f\text{-RR}(v_{x|i}, i))| = \Delta_i$. This happens with the probability $P(f\text{-RR}(v_x(i)) = 1, f\text{-RR}(v_{x|i}(i)) = 0)$. To compute this probability, we use marginal probability and Bayes' theorem, as follows:

$$
\begin{aligned}
&P(f\text{-RR}(v_x(i)) = 1, f\text{-RR}(v_{x|i}(i)) = 0) \\
&= P\Big(f\text{-RR}(v_x(i)) = 1, f\text{-RR}(v_{x|i}(i)) = 0, v_x(i) = 1, v_{x|i}(i) = 0\Big) \\
&\quad + P\Big(f\text{-RR}(v_x(i)) = 1, f\text{-RR}(v_{x|i}(i)) = 0, v_x(i) = 0, v_{x|i}(i) = 1\Big) \\
&= P\Big(f\text{-RR}(v_x(i)) = 1 | f\text{-RR}(v_{x|i}(i)) = 0, v_x(i) = 1, v_{x|i}(i) = 0\Big) \\
&\quad \times P(f\text{-RR}(v_{x|i}(i)) = 0 | v_x(i) = 1, v_{x|i}(i) = 0) \\
&\quad \times P(v_x(i) = 1 | v_{x|i}(i) = 0) \times P(v_{x|i}(i) = 0) \\
&\quad + P\Big(f\text{-RR}(v_x(i)) = 1 | f\text{-RR}(v_{x|i}(i)) = 0, v_x(i) = 0, v_{x|i}(i) = 1\Big) \\
&\quad \times P(f\text{-RR}(v_{x|i}(i)) = 0 | v_x(i) = 0, v_{x|i}(i) = 1) \\
&\quad \times P(v_x(i) = 0 | v_{x|i}(i) = 1) \times P(v_{x|i}(i) = 1) \\
&= P\Big(f\text{-RR}(v_x(i)) = 1 | v_x(i) = 1\Big) \\
&\quad \times P(f\text{-RR}(v_{x|i}(i)) = 0 | v_{x|i}(i) = 0) \times P(v_{x|i}(i) = 0) \\
&\quad + P\Big(f\text{-RR}(v_x(i)) = 1 | v_x(i) = 0\Big) \\
&\quad \times P(f\text{-RR}(v_{x|i}(i)) = 0 | v_{x|i}(i) = 1) \times P(v_{x|i}(i) = 1) \\
&= p_{Xi}^2 P(v_{x|i}(i) = 0) + q_{Xi}^2 P(v_{x|i}(i) = 1)
\end{aligned}
\tag{17}
$$

- If $f\text{-RR}(v_x(i)) = 0$ and $f\text{-RR}(v_{x|i}(i)) = 1$, then $|\mathcal{E}(f\text{-RR}(v_x(i))) - \mathcal{E}(f\text{-RR}(v_{x|i}(i)))| = \Delta_i$. This happens with the probability $P(f\text{-RR}(v_x(i)) = 0, f\text{-RR}(v_{x|i}(i)) = 1)$. To compute this probability, we use marginal probability and Bayes' theorem, as follows:

$$
\begin{aligned}
&P(f\text{-RR}(v_x(i)) = 0, f\text{-RR}(v_{x|i}(i)) = 1) \\
&= P\Big(f\text{-RR}(v_x(i)) = 0, f\text{-RR}(v_{x|i}(i)) = 1, v_x(i) = 1, v_{x|i}(i) = 0\Big) \\
&\quad + P\Big(f\text{-RR}(v_x(i)) = 0, f\text{-RR}(v_{x|i}(i)) = 1, v_x(i) = 0, v_{x|i}(i) = 1\Big) \\
&= q_{Xi}^2 P(v_{x|i}(i) = 0) + p_{Xi}^2 P(v_{x|i}(i) = 1)
\end{aligned}
\tag{18}
$$

Consequently, the expectation $\mathbb{E}|\mathcal{E}(f\text{-RR}(v_x, i)) - \mathcal{E}(f\text{-RR}(v_{x|i}, i))|$ is computed as follows:

$$
\begin{aligned}
&\mathbb{E}|\mathcal{E}(f\text{-RR}(v_x, i)) - \mathcal{E}(f\text{-RR}(v_{x|i}, i))| \\
&= \Big(p_{Xi}^2 P(v_{x|i}(i) = 0) + q_{Xi}^2 P(v_{x|i}(i) = 1)\Big)\Delta_i \\
&\quad + \Big(q_{Xi}^2 P(v_{x|i}(i) = 0) + p_{Xi}^2 P(v_{x|i}(i) = 1)\Big)\Delta_i \\
&= (p_{Xi}^2 + q_{Xi}^2)\Delta_i
\end{aligned}
\tag{19}
$$

From Eqs. 12, 16, and 19, we have that

$$\frac{P(f\text{-RR}(v_x) = v_z)}{P(f\text{-RR}(\widetilde{v}_x) = v_z)}$$

$$\leq \prod_{i=0}^{rl-1} \Big( \frac{P(f\text{-RR}(v_x(i)) = v_z(i))}{P(f\text{-RR}(v_{x|i}(i)) = v_z(i))} \Big)^{\frac{\Delta_i}{\mathbb{E}|\mathcal{E}(f\text{-RR}(v_x,i)) - \mathcal{E}(f\text{-RR}(v_{x|i},i))|}}$$

$$= \prod_{i=0}^{rl-1} \Big( \frac{P(f\text{-RR}(v_x(i)) = 0|v_x(i) = 1)P(f\text{-RR}(v_x(i)) = 1|v_x(i) = 0)}{P(f\text{-RR}(v_x(i)) = 1|v_x(i) = 1)P(f\text{-RR}(v_x(i)) = 0|v_x(i) = 0)} \Big)^{\frac{1}{p_{Xi}^2 + q_{Xi}^2}}$$

$$= \prod_{i=0}^{l-1} \Big( \alpha^2 \exp(2\epsilon_X \frac{i}{l}) \Big)^{\frac{r}{p_{Xi}^2 + (1-p_{Xi})^2}} \tag{20}$$

Taking the natural logarithm of two sides of Eq. 20:

$$\ln \frac{P(f\text{-RR}(v_x) = v_z)}{P(f\text{-RR}(\widetilde{v}_x) = v_z)} \leq \sum_{i=0}^{l-1} \ln \Big( \alpha^2 \exp(2\epsilon_X \frac{i}{l}) \Big)^{\frac{r}{p_{Xi}^2 + (1-p_{Xi})^2}}$$

$$= \sum_{i=0}^{l-1} \Big( \frac{r}{p_{Xi}^2 + (1-p_{Xi})^2} \ln \big( \alpha^2 \exp(2\epsilon_X \frac{i}{l}) \big) \Big) \tag{21}$$

Let us bound the summation in Eq. 21 using the following inequality:

$$\ln(a) \leq a - 1 \text{ for } a > 0 \tag{22}$$

The proof of Eq. 22 is as follows. Let $a > 0$, we define $h(a) = \ln(a) - a + 1$. We have: $h'(a) = \frac{1}{a} - 1 = 0 \Leftrightarrow a = 1$, and since $h''(a) = -\frac{1}{a^2} < 0, \forall a > 0$, we get the maximal point at $a = 1$. We also have: $\lim_{a \to 0+} h(a) = -\infty = \lim_{a \to \infty} h(a)$. Therefore, $a = 1$ is the global maximal point and than $\forall a > 0, h(a) \leq h(a = 1) = 0$, so $\ln(a) - a + 1 \leq 0$. Therefore, Eq. 22 does hold.

Note that, to simultaneously satisfy the randomization probabilities $p_X = \frac{1}{1+\alpha \exp(\frac{i\%l}{l}\epsilon_X)} \geq 0$ and $q_X = \frac{\alpha \exp(\frac{i\%l}{l}\epsilon_X)}{1+\alpha \exp(\frac{i\%l}{l}\epsilon_X)} \geq 0$ in Eq. 2, we need to have: *(i)* $1 + \alpha \exp(\frac{i\%l}{l}\epsilon_X) \geq 0$ and *(ii)* $\alpha \exp(\frac{i\%l}{l}\epsilon_X) \geq 0$. Since $\exp(\frac{i\%l}{l}\epsilon_X) \geq 0$ is always true, from *(i)*, $\alpha \geq -\exp(-\frac{i\%l}{l}\epsilon_X)$, and from *(ii)*, $\alpha \geq 0$. Therefore, $\alpha \geq 0$ is necessary to satisfy the condition $p_X \geq 0$ and $q_X \geq 0$. To apply Eq. 22 into Eq. 21, we need to have $a = \alpha^2 \exp(2\epsilon_X \frac{i}{l}) > 0 \Rightarrow \alpha \neq 0$. As a result, we have that

$$\alpha > 0 \tag{23}$$

Applying Eq. 22 into Eq. 21 where $a = \alpha^2 \exp(2\epsilon_X \frac{i}{l})$:

$$\ln \frac{P(f\text{-RR}(v_x) = v_z)}{P(f\text{-RR}(\widetilde{v}_x) = v_z)} \leq \sum_{i=0}^{l-1} \Big( \frac{r}{p_{Xi}^2 + (1-p_{Xi})^2} \ln \big( \alpha^2 \exp(2\epsilon_X \frac{i}{l}) \big) \Big)$$

$$\leq \sum_{i=0}^{l-1} \frac{r\alpha^2 \exp(2\epsilon_X \frac{i}{l})}{p_{Xi}^2 + (1-p_{Xi})^2} - \sum_{i=0}^{l-1} \frac{r}{p_{Xi}^2 + (1-p_{Xi})^2} \tag{24}$$

To bound the logarithm in Eq. 24, we use: $p_{Xi}^2 + q_{Xi}^2 = p_{Xi}^2 + (1-p_{Xi})^2 = \Big( \frac{1}{1+\alpha \exp(\frac{i\%l}{l}\epsilon_X)} \Big)^2 +$

$\Big( \frac{\alpha \exp(\frac{i\%l}{l}\epsilon_X)}{1+\alpha \exp(\frac{i\%l}{l}\epsilon_X)} \Big)^2 = \frac{1 + \Big( \alpha \exp(\frac{i\%l}{l}\epsilon_X) \Big)^2}{\Big( 1+\alpha \exp(\frac{i\%l}{l}\epsilon_X) \Big)^2} \leq 1$, and $p_{Xi}^2 + (1-p_{Xi})^2 \geq \frac{(p_{Xi}+1-p_{Xi})^2}{2} = \frac{1}{2}$ (In fact,

$\forall a, b : a^2 + b^2 \geq \frac{(a+b)^2}{2} \Leftrightarrow (a-b)^2 \geq 0$, which is true). Note that, from Eq. 19, we have that $\frac{\Delta_i}{\mathbb{E}|\mathcal{E}(f\text{-RR}(v_a,i)) - \mathcal{E}(f\text{-RR}(v_{a|i},i))|} = \frac{1}{p_{Xi}^2 + q_{Xi}^2} \geq 1$.

Applying these inequalities in Eq. 24, we obtain:

$$
\begin{aligned}
\ln \frac{P(f\text{-RR}(v_x) = v_z)}{P(f\text{-RR}(\widetilde{v}_x) = v_z)} &\leq \sum_{i=0}^{l-1} \frac{r\alpha^2 \exp(2\epsilon_X \frac{i}{l})}{p_{Xi}^2 + (1 - p_{Xi})^2} - \sum_{i=0}^{l-1} \frac{r}{p_{Xi}^2 + (1 - p_{Xi})^2} \\
&< \sum_{i=0}^{l-1} 2r\alpha^2 \exp(2\epsilon_X \frac{i}{l}) - \sum_{i=0}^{l-1} r = \sum_{i=0}^{l-1} 2r\alpha^2 \exp(2\epsilon_X \frac{i}{l}) - rl \leq \epsilon_X
\end{aligned}
\tag{25}
$$

By solving Eq. 25, we have that

$$
\alpha^2 \leq \frac{\epsilon_X + rl}{2r \sum_{i=0}^{l-1} \exp(2\epsilon_X \frac{i}{l})} \Leftrightarrow |\alpha| \leq \sqrt{\frac{\epsilon_X + rl}{2r \sum_{i=0}^{l-1} \exp(2\epsilon_X \frac{i}{l})}}
\tag{26}
$$

Therefore, from Eqs. 23 and 26, we have that $\forall \alpha : 0 < \alpha \leq \sqrt{\frac{\epsilon_X + rl}{2r \sum_{i=0}^{l-1} \exp(2\epsilon_X \frac{i}{l})}}$, the $f$-RR mechanism satisfies $\epsilon_X$-LDP. Consequently, Theorem 2 holds.

## F  PROOF OF THEOREM 3

*Proof.* We have: $\frac{P(label\text{-RR}(y_x)=z|y_x)}{P(label\text{-RR}(\widetilde{y}_x)=z|\widetilde{y}_x)} \leq \frac{\max P(label\text{-RR}(y_x)=z|y_x)}{\min P(label\text{-RR}(\widetilde{y}_x)=z|\widetilde{y}_x)} = \frac{\frac{\exp(\beta)}{1+\exp(\beta)}}{\frac{1}{(1+\exp(\beta))(C-1)}} = \exp(\beta + \ln(C-1)) \leq \exp(\epsilon_Y) \Leftrightarrow \beta \leq \epsilon_Y - \ln(C-1)$. Consequently, Theorem 3 does hold. $\qquad \square$

## G  *label*-RR AND LABELDP COMPARISON

Although our label-RR is inspired by the randomizing probability (Eq. 1, (Ghazi et al., 2021)) in the LabelDP showcased by (Ghazi et al., 2021), there are two major differences between our label-RR and LabelDP discussed next.

**(1)** We combine $f$-RR and label-RR to completely protect a data sample. As pointed out in (Busa-Fekete et al., 2021), only protecting the label as in the LabelDP offers a weaker privacy protection than it appears, as the features are sufficiently predictive of the label, obscuring the label is not enough, as a classifier can still be trained on such noisy data; hence, a user experiences privacy loss due to both the public release of the features and the private release of the label.

**(2)** The RRWithPrior algorithm that is used to guarantee LabelDP requiring publicly available priors and the multi-stage training (LP-MST) illustrating RRWithPrior algorithm cannot be straightforwardly applied to federated learning. In LP-MST algorithm, the dataset is partitioned into subsets, then based on the prior probability to randomize the label and add the data with that randomized label to the training data. These steps are presently applied on centralized training and it has not been show how to be effectively applied in federated learning.

Using the upper bound of $\beta$ (Theorem 3) results in the same randomizing probabilities between label-RR and LabelDP. For instance, in Eq. 3, $p_Y = \frac{\exp(\beta)}{1+\exp(\beta)} = \frac{\exp(\epsilon_Y - \ln(C-1))}{1+\exp(\epsilon_Y - \ln(C-1))} = \frac{\exp(\epsilon_Y)}{C-1+\exp(\epsilon_Y)}$ and $q_Y = \frac{1}{(1+\exp(\beta))(C-1)} = \frac{1}{C-1+\exp(\epsilon_Y)}$, which is equivalent to Eq. 1 (Ghazi et al., 2021). Therefore, we did not include LabelDP (Ghazi et al., 2021) in comparison.

## H  PROOF OF THEOREM 4

*Proof.* We have $\xi_a = \mathbb{E}|\mathcal{E}(f\text{-RR}(v_a)) - \mathcal{E}(v_a)| = \sum_{i \in [0,l-1]} (p_{Xi} \times 0 + q_{Xi} \times \Delta_i) = \sum_{i \in [0,l-1]} q_{Xi} \times \Delta_i$. Therefore, Theorem 4 hold. $\qquad \square$

## I  CORRECTED PRIVACY BUDGET BOUNDS IN LATENT (ARACHCHIGE ET AL., 2019)

In this section, we aim at providing corrected privacy budget bounds for LATENT (Arachchige et al., 2019). LATENT first encodes embedded features $e_x$ into an $rl$-bit binary vector $v_x$. Then, each bit $i \in [0, rl - 1]$ is randomized by a RR mechanism (i.e., the MOUE algorithm for high sensitivities in Theorem 3.3 (Arachchige et al., 2019)), denoted $f$-LT, as follows:

$$\forall i \in [0, rl - 1] : P(v'_x(i) = 1) = \begin{cases} p_X = \dfrac{1}{1 + \alpha}, & \text{if } v_x(i) = 1 \\ q_X = \dfrac{1}{1 + \alpha \exp(\frac{\epsilon_X}{rl})}, & \text{if } v_x(i) = 0 \end{cases} \tag{27}$$

From Eq. 27, we also have that $P(v'_x(i) = 0) = 1 - p_X = \frac{\alpha}{1+\alpha}$ if $v_x(i) = 1$, and $P(v'_x(i) = 0) = 1 - q_X = \frac{\alpha \exp(\frac{\epsilon_X}{rl})}{1 + \alpha \exp(\frac{\epsilon_X}{rl})}$ if $v_x(i) = 0$.

**Theorem 5.** *LATENT with the randomization probabilities as in Eq. 27 preserves $\epsilon_{corrected}$-LDP, where $\epsilon_{corrected} = \frac{(1+\alpha)(1+\alpha \exp(\frac{\epsilon_X}{rl}))}{\alpha(1+\exp(\frac{\epsilon_X}{rl}))}\epsilon_X$.*

*Proof.* Similar to the analysis in **Appendix E**, we obtain the following inequality:

$$\frac{P(f\text{-LT}(v_x) = v_z)}{P(f\text{-LT}(\widetilde{v}_x) = v_z)} \leq \prod_{i=0}^{rl-1} \left( \frac{P(f\text{-LT}(v_x(i)) = v_z(i))}{P(f\text{-LT}(v_{x|i}(i)) = v_z(i))} \right)^{\frac{\Delta_i}{\mathbb{E}|\mathcal{E}(f\text{-LT}(v_x,i)) - \mathcal{E}(f\text{-LT}(v_{x|i},i))|}} \leq \exp(\epsilon_X) \tag{28}$$

and the expectation $\mathbb{E}|\mathcal{E}(f\text{-LT}(v_x, i)) - \mathcal{E}(f\text{-LT}(v_{x|i}, i))|$ is computed as follows:

$$\begin{aligned}
&\mathbb{E}|\mathcal{E}(f\text{-LT}(v_x, i)) - \mathcal{E}(f\text{-LT}(v_{x|i}, i))| \\
&= \Big( P(f\text{-LT}(v_x, i) = 1 | v_x(i) = 1) \\
&\quad \times P(f\text{-LT}(v_{x|i}, i) = 0 | v_{x|i}(i) = 0) \times P(v_{x|i}(i) = 0) \\
&\quad + P(f\text{-LT}(v_x, i) = 1 | v_x(i) = 0) \\
&\quad \times P(f\text{-LT}(v_{x|i}, i) = 0 | v_{x|i}(i) = 1) \times P(v_{x|i}(i) = 1) \Big) \Delta_i \\
&\quad + \Big( P(f\text{-LT}(v_x, i) = 0 | v_x(i) = 1) \\
&\quad \times P(f\text{-LT}(v_{x|i}, i) = 1 | v_{x|i}(i) = 0) \times P(v_{x|i}(i) = 0) \\
&\quad + P(f\text{-LT}(v_x, i) = 0 | v_x(i) = 0) \\
&\quad \times P(f\text{-LT}(v_{x|i}, i) = 1 | v_{x|i}(i) = 1) \times P(v_{x|i}(i) = 1) \Big) \Delta_i \\
&= \Big( p_{Xi}(1 - q_{Xi}) P(v_{x|i}(i) = 0) + q_{Xi}(1 - p_{Xi}) P(v_{x|i}(i) = 1) \\
&\quad + (1 - p_{Xi}) q_{Xi} P(v_{x|i}(i) = 0) + (1 - q_{Xi}) p_{Xi} P(v_{x|i}(i) = 1) \Big) \Delta_i \\
&= \Big( p_{Xi}(1 - q_{Xi}) + q_{Xi}(1 - p_{Xi}) \Big) \Delta_i \tag{29}
\end{aligned}$$

Furthermore, we have:

$$p_{Xi}(1 - q_{Xi}) + q_{Xi}(1 - p_{Xi}) = \frac{\alpha(1 + \exp(\frac{\epsilon_X}{rl}))}{(1 + \alpha)(1 + \alpha \exp(\frac{\epsilon_X}{rl}))} \tag{30}$$

From Eqs. 28-30, we have that

$$
\frac{P(f\text{-LT}(v_x) = v_z)}{P(f\text{-LT}(\widetilde{v}_x) = v_z)}
$$

$$
\leq \prod_{i=0}^{rl-1} \left( \frac{P(f\text{-LT}(v_x(i)) = v_z(i))}{P(f\text{-LT}(v_{x|i}(i)) = v_z(i))} \right)^{\frac{\Delta_i}{\mathbb{E}|\mathcal{E}(f\text{-LT}(v_x,i)) - \mathcal{E}(f\text{-LT}(v_{x|i},i))|}}
$$

$$
= \prod_{i=0}^{rl-1} \left( \frac{P(f\text{-LT}(v_x(i)) = 1|v_x(i) = 1)}{P(f\text{-LT}(v_x(i)) = 0|v_x(i) = 1)} \right)^{\frac{\Delta_i}{(p_{Xi}(1-q_{Xi}) + q_{Xi}(1-p_{Xi}))\Delta_i}}
$$

$$
\times \prod_{i=0}^{rl-1} \left( \frac{P(f\text{-LT}(v_x(i)) = 0|v_x(i) = 0)}{P(f\text{-LT}(v_x(i)) = 1|v_x(i) = 0)} \right)^{\frac{\Delta_i}{(p_{Xi}(1-q_{Xi}) + q_{Xi}(1-p_{Xi}))\Delta_i}}
$$

$$
= \prod_{i=0}^{rl-1} \left( \exp(\frac{\epsilon_X}{rl}) \right)^{\frac{1}{p_{Xi}(1-q_{Xi}) + q_{Xi}(1-p_{Xi})}} \tag{31}
$$

Then, from Eq. 31, we have:

$$
\epsilon_{corrected} = \ln \left( \Pi_{i=0}^{rl-1} \left( \exp(\frac{\epsilon_X}{rl}) \right)^{\frac{1}{p_{Xi}(1-q_{Xi}) + q_{Xi}(1-p_{Xi})}} \right) = \frac{(1+\alpha)(1+\alpha\exp(\frac{\epsilon_X}{rl}))}{\alpha(1+\exp(\frac{\epsilon_X}{rl}))} \epsilon_X \tag{32}
$$

Consequently, Theorem 5 holds. $\qquad\square$

From Theorem 5, we show the proportion $\epsilon_{corrected}/\epsilon_X$ as a function of $r$ in Figure 9a and as a function of $l$ in Figure 9b. Following the experiment settings in LATENT (Arachchige et al., 2019), with the commonly used $\alpha = 7$, when changing $r \in \{10, 100, 1,000, 10,000\}$ with a fixed $l = 10$ (Figure 9a), or when changing $l \in \{5, 10, 20, 100, 1,000\}$ with a fixed $r = 1,000$ (Figure 9b) and under a tight privacy budget $\epsilon_X = 0.1$, the proportion $\epsilon_{corrected}/\epsilon_X$ moderately changes among $[4.57, 4.75]$. In other words, the $\epsilon_{corrected}$ is remarkably larger than $\epsilon_X$, for most $r$ and $l$ values in practice. Unlike LATENT, our mechanism does not suffer from this problem, i.e., in BitRand, $\epsilon_{corrected}/\epsilon_X = 1$, thanks to our bit-aware randomization probabilities for LDP in binary encoding (**Theorem 2**).

## J  CORRECTED PRIVACY BUDGET BOUNDS IN OME (LYU ET AL., 2020A)

In this section, we aim at providing corrected privacy budget bounds for OME. OME first encodes embedded features $e_x$ into an $rl$-bit binary vector $v_x$. Then, each bit $i \in [0, rl - 1]$ is randomized by the following $f$-OME mechanism:

$$
\forall i \in [0, rl - 1] : P(v'_x(i) = 1) = \begin{cases} p_{1X} = \dfrac{\alpha}{1+\alpha}, & \text{if } i \in 2j, v_x(i) = 1 \\[2mm] p_{2X} = \dfrac{1}{1+\alpha^3}, & \text{if } i \in 2j+1, v_x(i) = 1 \\[2mm] q_X = \dfrac{1}{1+\alpha\exp(\frac{\epsilon_X}{rl})}, & \text{if } v_x(i) = 0 \end{cases} \tag{33}
$$

From Eq. 33, we also have that $P(v'_x(i) = 0) = 1 - p_{1X} = \frac{1}{1+\alpha}$ if $v_x(i) = 1$ and $i \in 2j$, $P(v'_x(i) = 0) = 1 - p_{2X} = \frac{\alpha^3}{1+\alpha^3}$ if $v_x(i) = 1$ and $i \in 2j+1$, and $P(v'_x(i) = 0) = 1 - q_X = \frac{\alpha\exp(\frac{\epsilon_X}{rl})}{1+\alpha\exp(\frac{\epsilon_X}{rl})}$ if $v_x(i) = 0$.

**Theorem 6.** *OME with the randomization probabilities as in Eq. 33 preserves $\epsilon_{corrected}$-LDP, where* $\epsilon_{corrected} = \left( \frac{rl}{Q_1} - \frac{rl}{Q_2} \right)\ln(\alpha) + \frac{\epsilon_X}{2Q_1} + \frac{\epsilon_X}{2Q_2}$ *in which* $Q_1 = \frac{\alpha}{1+\alpha}\frac{\alpha\exp(\frac{\epsilon_X}{rl})}{1+\alpha\exp(\frac{\epsilon_X}{rl})} + \frac{1}{1+\alpha\exp(\frac{\epsilon_X}{rl})}\frac{1}{1+\alpha}$ *and* $Q_2 = \frac{1}{1+\alpha^3}\frac{\alpha\exp(\frac{\epsilon_X}{rl})}{1+\alpha\exp(\frac{\epsilon_X}{rl})} + \frac{1}{1+\alpha\exp(\frac{\epsilon_X}{rl})}\frac{\alpha^3}{1+\alpha^3}$.

*Proof.* Similar to the analysis in **Appendix E** and **Appendix I**, we obtain:

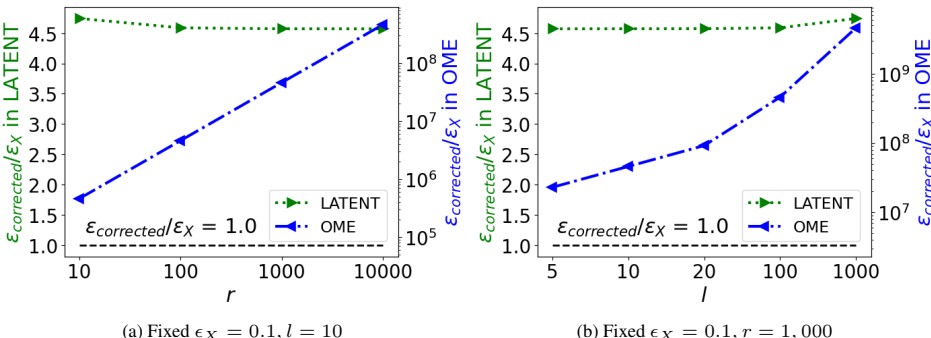

Figure 9: Impacts of $r$ and $l$ on $\epsilon_{Corrected}/\epsilon_X$ in LATENT (Arachchige et al., 2019) and OME (Lyu et al., 2020a).

$$\frac{P(f\text{-OME}(v_x) = v_z)}{P(f\text{-OME}(\widetilde{v}_x) = v_z)} \leq \prod_{i=0}^{rl-1} \left( \frac{P(f\text{-OME}(v_x(i)) = v_z(i))}{P(f\text{-OME}(v_{x|i}(i)) = v_z(i))} \right)^{\frac{\Delta_i}{\mathbb{E}|\mathcal{E}(f\text{-OME}(v_x,i)) - \mathcal{E}(f\text{-OME}(v_{x|i},i))|}} \leq \exp(\epsilon_X)$$
$$(34)$$

and the expectation $\mathbb{E}|\mathcal{E}(f\text{-OME}(v_x, i)) - \mathcal{E}(f\text{-OME}(v_{x|i}, i))|$ is computed as follows:

$$\mathbb{E}|\mathcal{E}(f\text{-OME}(v_x, i)) - \mathcal{E}(f\text{-OME}(v_{x|i}, i))|$$
$$= \begin{cases} \big(p_{1Xi}(1 - q_{Xi}) + q_{Xi}(1 - p_{1Xi})\big)\Delta_i = Q_1\Delta_i, \text{if } i \in 2j \\ \big(p_{2Xi}(1 - q_{Xi}) + q_{Xi}(1 - p_{2Xi})\big)\Delta_i = Q_2\Delta_i, \text{if } i \in 2j+1 \end{cases} \quad (35)$$

where $Q_1 = p_{1Xi}(1 - q_{Xi}) + q_{Xi}(1 - p_{1Xi}) = \frac{\alpha}{1+\alpha} \frac{\alpha \exp(\frac{\epsilon_X}{rl})}{1+\alpha \exp(\frac{\epsilon_X}{rl})} + \frac{1}{1+\alpha \exp(\frac{\epsilon_X}{rl})} \frac{1}{1+\alpha}$,

and $Q_2 = p_{2Xi}(1 - q_{Xi}) + q_{Xi}(1 - p_{2Xi}) = \frac{1}{1+\alpha^3} \frac{\alpha \exp(\frac{\epsilon_X}{rl})}{1+\alpha \exp(\frac{\epsilon_X}{rl})} + \frac{1}{1+\alpha \exp(\frac{\epsilon_X}{rl})} \frac{\alpha^3}{1+\alpha^3}$.

From Eqs. 34 and 35, we have:

$$\frac{P(f\text{-OME}(v_x) = v_z)}{P(f\text{-OME}(\widetilde{v}_x) = v_z)}$$
$$\leq \prod_{i=0}^{rl-1} \left( \frac{P(f\text{-OME}(v_x(i)) = v_z(i))}{P(f\text{-OME}(v_{x|i}(i)) = v_z(i))} \right)^{\frac{\Delta_i}{\mathbb{E}|\mathcal{E}(f\text{-OME}(v_x,i)) - \mathcal{E}(f\text{-OME}(v_{x|i},i))|}}$$
$$= \prod_{i \in 2j} \left( \frac{P(f\text{-OME}(v_x(i)) = 1|v_x(i) = 1)P(f\text{-OME}(v_x(i)) = 0|v_x(i) = 0)}{P(f\text{-OME}(v_x(i)) = 1|v_x(i) = 0)P(f\text{-OME}(v_x(i)) = 0|v_x(i) = 1)} \right)^{\frac{\Delta_i}{Q_1\Delta_i}}$$
$$\times \prod_{i \in 2j+1} \left( \frac{P(f\text{-OME}(v_x(i)) = 1|v_x(i) = 1)P(f\text{-OME}(v_x(i)) = 0|v_x(i) = 0)}{P(f\text{-OME}(v_x(i)) = 1|v_x(i) = 0)P(f\text{-OME}(v_x(i)) = 0|v_x(i) = 1)} \right)^{\frac{\Delta_i}{Q_2\Delta_i}}$$
$$= \alpha^{\frac{rl}{Q_1} - \frac{rl}{Q_2}} \exp\left(\frac{\epsilon_X}{2Q_1} + \frac{\epsilon_X}{2Q_2}\right) \quad (36)$$

Then, from Eq. 36, we have:

$$\epsilon_{corrected} = \ln\left(\alpha^{\frac{rl}{Q_1} - \frac{rl}{Q_2}} \exp\left(\frac{\epsilon_X}{2Q_1} + \frac{\epsilon_X}{2Q_2}\right)\right) = \left(\frac{rl}{Q_1} - \frac{rl}{Q_2}\right)\ln(\alpha) + \frac{\epsilon_X}{2Q_1} + \frac{\epsilon_X}{2Q_2} \quad (37)$$

Consequently, Theorem 6 does hold. $\qquad \square$

From Theorem 6, we show the proportion $\epsilon_{corrected}/\epsilon_X$ as a function of $r$ in Figure 9a and as a function of $l$ in Figure 9b. Following the experiment settings in OME (Lyu et al., 2020a), with the commonly used $\alpha = 100$, when changing $r \in \{10, 100, 1,000, 10,000\}$ with a fixed $l = 10$ (Figure 9a), or when changing $l \in \{5, 10, 20, 100, 1,000\}$ with a fixed $r = 1,000$ (Figure 9b) and under a tight privacy budget $\epsilon_X = 0.1$, the proportion $\epsilon_{corrected}/\epsilon_X$ significantly changes among $[4.6e + 6, 4.6e + 9]$. In other words, the $\epsilon_{corrected}$ is extremely larger than $\epsilon_X$, for most $r$ and $l$ values in practice. Since $\alpha = 100$ causes the extreme privacy exaggeration, in our experiment, to compare with OME, we use $\alpha = 1$. This value is used in OME (Lyu et al., 2020a) and generates $\epsilon_{corrected}/\epsilon_X \approx 2$, which offers a reasonable range to apply OME in practice. Unlike OME, our mechanism does not suffer from this problem, i.e., in BitRand, $\epsilon_{corrected}/\epsilon_X = 1$, thanks to our bit-aware randomization probabilities for LDP in binary encoding (**Theorem 2**).

## K   SUPPLEMENTARY THEORETICAL RESULTS

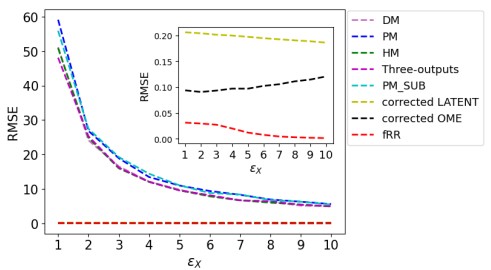

Figure 10: RMSE error comparison as a function of $\epsilon_X$.

**Setting for Gaussian and Laplace mechanisms.** The Gaussian and Laplace mechanisms naturally apply an addition operation, which add noise into the data or embedded features. Therefore, in our analysis of expected error bound comparison for an embedded feature (Figure 3), we add noise into the embedded feature following the Gaussian and Laplace mechanisms. The sensitivity captures the magnitude by which an embedded feature can change in the worst case. In our experiment and analysis, we use $l = 10$ bits in which 1 sign bit, 5 bits for the integer, and 4 bits for the fraction part. Therefore, the maximum the embedded feature can be change, i.e., the sensitivity, is $2\sum_{i=-4}^{4} 2^i$. Note that, we multiply $\sum_{i=-4}^{4} 2^i$ by 2 since when we flip the sign bit, it significantly changes the value of the embedded feature from $-a$ to $a$ in which $a = \sum_{i=-4}^{4} 2^i$.

**RMSE error comparison in mean estimation.** To investigate how our proposed approach $f$-RR works with statistical query, we study our $f$-RR and other baselines with a mean estimation. We created a synthetic data that consists of $N = 1,000$ data samples $\{x_i\}_{i=1}^{N}$, each of them has $d = 768$ dimensions. The mean estimation is calculated over each dimension as $f_j(D) = \frac{1}{N}\sum_{i=1}^{N} x_{ij}$ for $j \in [1, d]$. Root mean square error (RMSE) is used to evaluate the error between the original vector and the randomized/estimated vector. The binary-encoding-based approaches (i.e., $f$-RR, corrected LATENT, and corrected OME) achieve a significantly small error compared with others. As can be seen in Figure 10, $f$-RR obtains the smallest error, which further shows the effectiveness of our proposed mechanism.

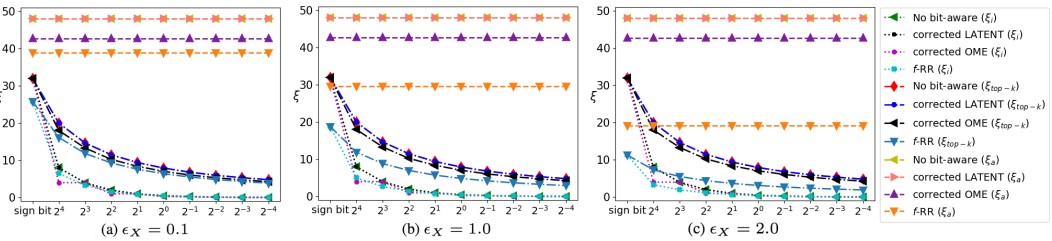

Figure 11: Expected error bound as a function of $\epsilon_X$ with fixed $r$ and $l$.

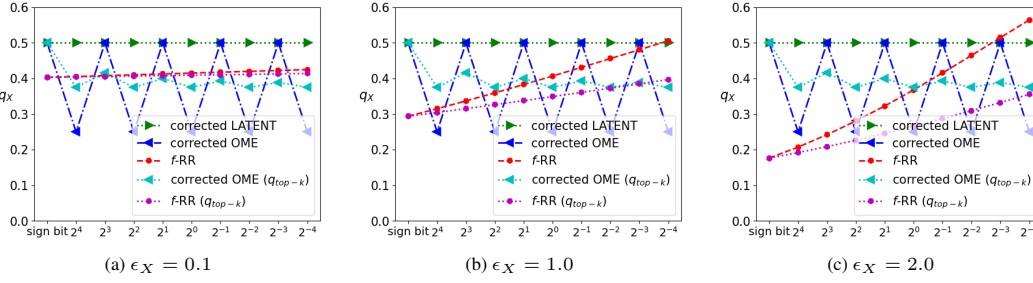

Figure 12: Randomization probability $q_X$ and $q_{top\text{-}k}$, given $l = 10$ and $r = 1,000$.

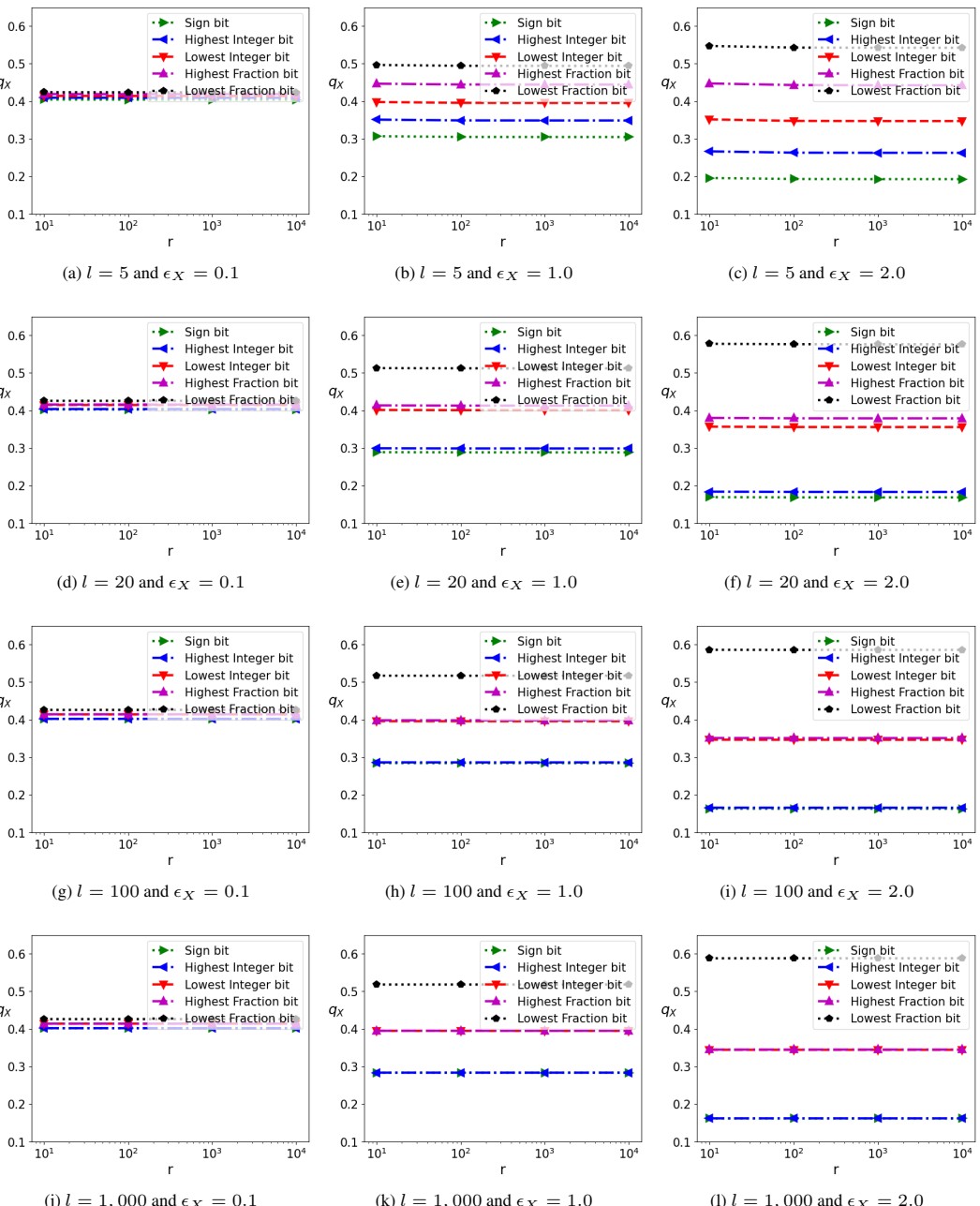

Figure 13: Randomization probability $q$ ($p = 1 - q$) as a function of $r$ with fixed $l$ and $\epsilon$.

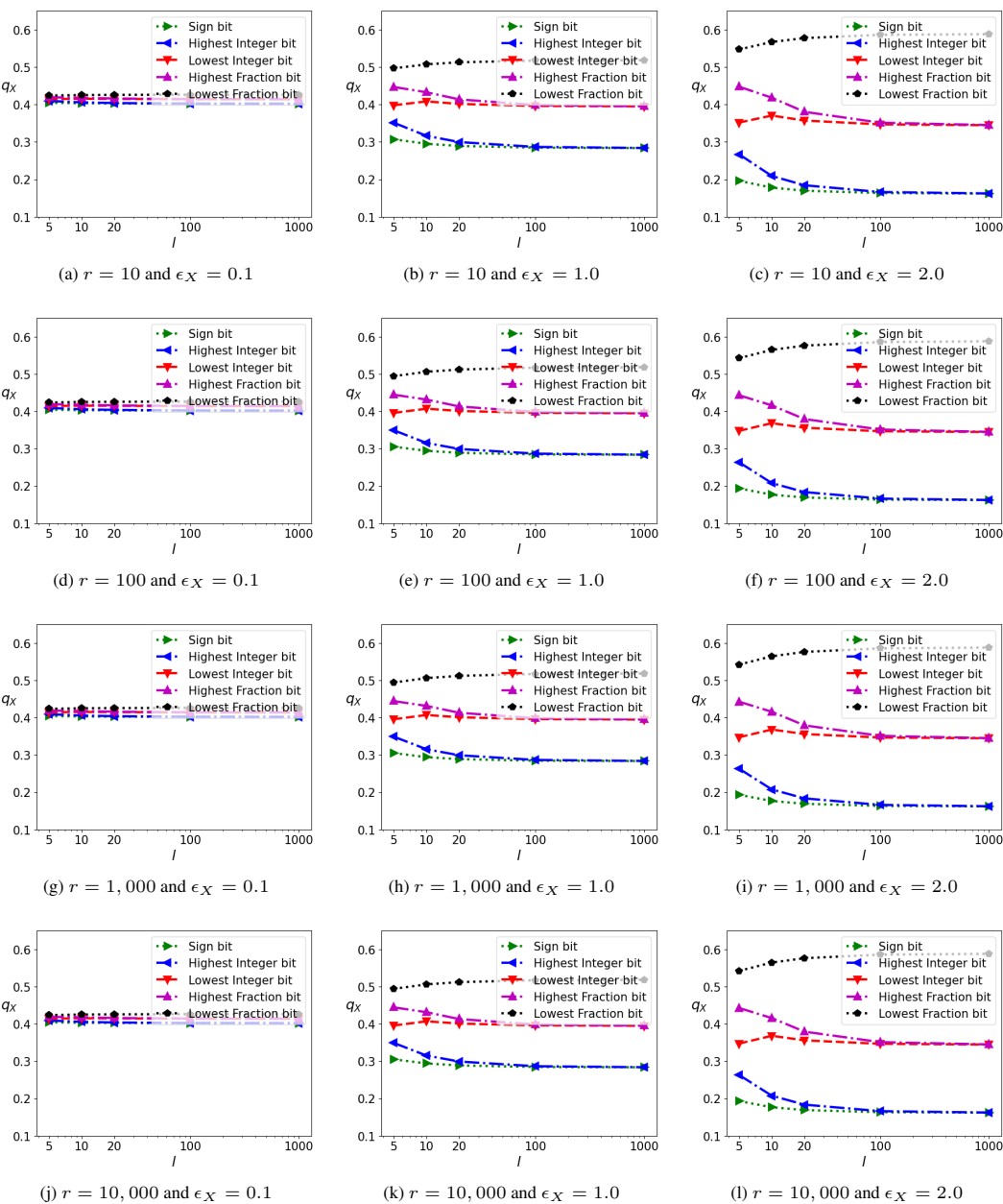

Figure 14: Randomization probability $q_X$ ($p_X = 1 - q_X$) as a function of $l$ with fixed $r$ and $\epsilon$.

## L SUPPLEMENTARY EXPERIMENTAL RESULTS

**Datasets and Data Processing.** We carried out our experiments on two textual datasets and two image datasets, including the AG dataset (Gulli et al., 2012), our collected Security and Exchange Commission (SEC) financial contract dataset, the large-scale celebFaces attributes (CelebA) dataset (Liu et al., 2015), and the Federated Extended MNIST (FEMNIST) dataset (Caldas et al., 2018). The AG dataset is a collection of news articles gathered from more than $2,000$ news sources by (Com). It is categorized into four classes: world, sport, business, and science/technology classes. Our SEC dataset consists of over $1,000$ contract clauses collected from contracts submitted in SEC filings[1]. The CelebA dataset consists of more than $200,000$ celebrity images, each with $40$ attributes, e.g., attractive face, big lips, big noses, black hair, etc., which are used as binary classes. The FEMNIST dataset is built by partitioning the images in Extended MNIST (Cohen et al., 2017) based on the writer of the handwritten digits and characters. For data preprocessing, we changed all words in the AG and SEC datasets to lower-case and removed punctuation marks. The breakdown of the datasets is in Table 1.

Table 1: Dataset breakdown.

| **Dataset** | **Train** | **Test** | **Samples/client** | **# classes** |
|---|---|---|---|---|
| | # samples | # samples | (Average) | |
| AG | $120,000$ | $7,600$ | 43 | 4 |
| SEC | $1,021$ | 134 | 3 | 2 |
| CelebA | $155,529$ | $19,962$ | 20 | 40 (binary) |
| FEMNIST | $734,033$ | $83,818$ | 227 | 62 |

**Model Configuration.** We use the test accuracy and the test area under the curve (AUC) as evaluation metrics. Models with higher values of test accuracy and AUC are better. We use the BERT-Base (Uncased) pre-trained model (ber; Devlin et al., 2018) to extract embedded features in the AG and SEC datasets. In the CelebA and FEMNIST datasets, we use the ResNet-18 pre-trained model (img; He et al., 2016). Dimension of the extracted embedded features in the AG and SEC datasets is $r = 768$, and in the CelebA and FEMNIST datasets is $r = 512$. For text and image classification tasks, we use two fully connected layers on top of embedded features, each of which consists of $1,500$ hidden neurons and uses a ReLU activation function. The output dimension is corresponding to the number of classes, i.e., 4, 2, 40, and 62 in the AG, SEC, CelebA, and FEMNIST datasets. SGD optimizer with the learning rate is $0.01$ in the AG and SEC datasets, $0.1$ in the FEMNIST and CelebA datasets.

**Experimental setting for anonymization (Sun et al., 2021).**

In LDP-FL (Sun et al., 2021), they design a LDP mechanism to perturb the weights at the local client, then each local client applies a split and shuffle mechanism on the weights of local model and sends each weight through an anonymous mechanism to the cloud. The purpose of the shuffling mechanism is to break the linkage among the model weight updates from the same clients and to mix them among updates from other clients, making it harder for the cloud to combine more than one piece of updates to infer more information about any client. Therefore, the key idea of the shuffle mechanism in LDP-FL is to mitigate the privacy degradation by high data dimension and many training/query iterations. In other words, the client anonymity is preserved, and the privacy budget will not accumulate.

When comparing with LDP-FL, we maintain their mechanism' spirits of no privacy accumulation. In the submission, we consider there is no privacy accumulation over the training iterations. It is equivalent to the shuffling step that breaks the linkage among the model weight updates with associated clients. We also used the same randomized response mechanism in the paper, which is Eq. 2 (Sun et al., 2021), to perturb the weight.

In the revision, we added an experiment that do not consider the privacy accumulation over data dimension and training/query iterations. This completely follows the gist of LDP-FL. In addition,

---

[1] https://www.sec.gov/edgar.shtml

the weights we used for LDP-FL. without actual shuffling or splitting can be considered a lossless process, therefore the results we reported here can be counted as an upper-bound result for LDP-FL.

As shown in Table 2, we obtained the slightly higher accuracy of LDP-FL compared with $f$-RR on the AG dataset. The key component of LDP-FL that helps to reduce the privacy accumulation issue is the shuffling mechanism. However, as pointed out in (Erlingsson et al., 2019), in the real world, it is possible that the anonymizers (i.e., shuffler) can either be compromised or collude with the coordinating server to extract sensitive information. Even though there is a marginally lower trade-off between privacy loss and model utility compared with LDP-FL, the advantage of $f$-RR is that it perturbs the data only once, then used the perturbed data for training process without facing an extra privacy risk potentially caused by the compromised or colluded anonymizer.

Table 2: Results of LDP-FL without privacy accumulation.

| $\epsilon_X$ | 1 | 2 | 3 | 4 | 5 | 6 | 7 | 8 | 9 | 10 |
|---|---|---|---|---|---|---|---|---|---|---|
| LDP-FL | 78.73 | 82.21 | 83.25 | 84.19 | 84.36 | 84.22 | 84.64 | 84.43 | 84.31 | 84.72 |
| $f$-RR | 72.46 | 79.72 | 81.42 | 79.35 | 81.11 | 79.84 | 79.50 | 80.05 | 81.20 | 82.72 |
| Noiseless model | 87.59 |  |  |  |  |  |  |  |  |  |

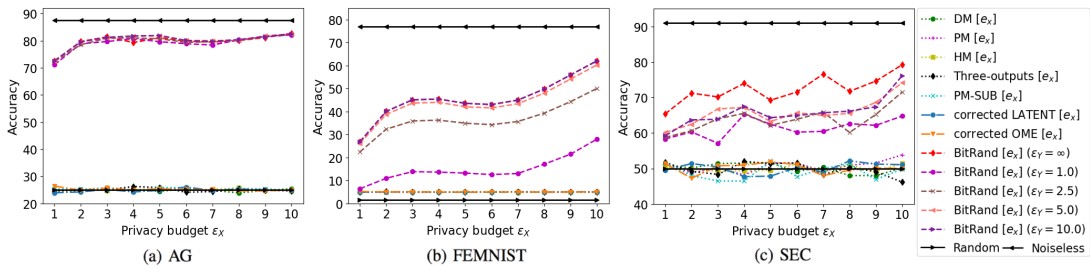

Figure 15: Accuracy of LDP algorithms applied on the embedded features $e_x$ in the AG, SEC, and FEMNIST datasets.

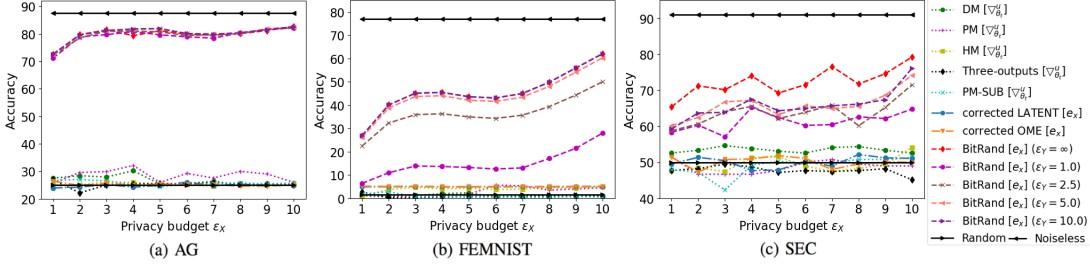

Figure 16: Accuracy of LDP algorithms applied on the gradients $\nabla_{\theta_t}^u$ in the AG, SEC, and FEMNIST datasets.

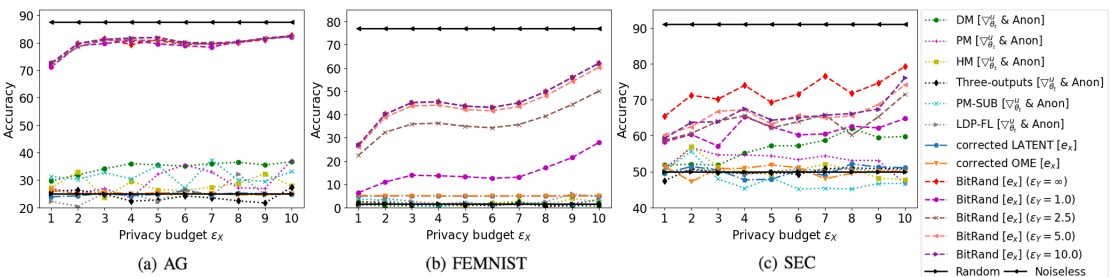

Figure 17: Accuracy of LDP algorithms applied on the gradients $\nabla_{\theta_t}^u$ with the anonymizer (Sun et al., 2021).

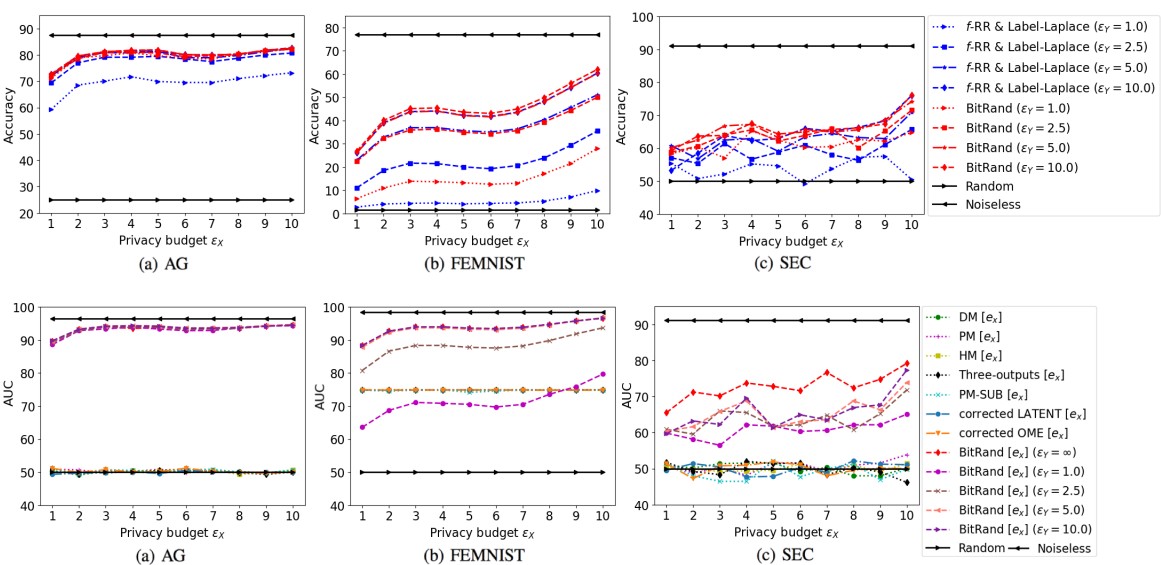

Figure 19: AUC values of LDP algorithms applied on the embedded features $e_x$ in the AG, SEC, and FEMNIST dataset.

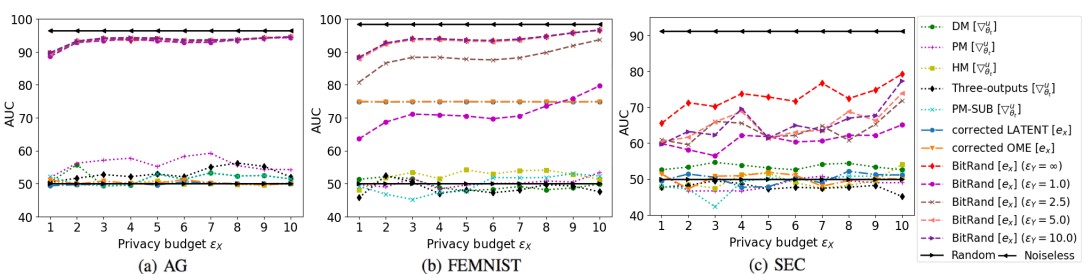

Figure 20: AUC values of LDP algorithms applied on the gradients $\nabla_{\theta_t}^u$ in the AG, SEC, and FEMNIST datasets.

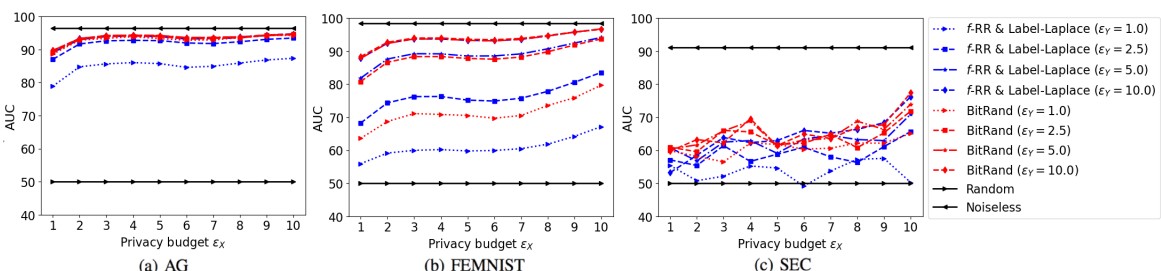

Figure 21: AUC values of each mechanism applied on labels in the AG, SEC, and FEMNIST datasets.

Table 3: AUC values of each algorithm applied on $e_x$ in the CelebA dataset. **Average** is the average of all 40 attributes.

| Algorithm [$\nabla_{\theta_t}^u$ with Anon] | Attribute | Attractive | Heavy Makeup | High Cheekbones | Male | Mouth Slightly Open | Smiling | Lipstick | **Average** |
|---|---|---|---|---|---|---|---|---|---|
| Noiseless | $\epsilon_X = \infty$ | 78.05 | 85.47 | 76.53 | 92.77 | 72.91 | 79.15 | 88.85 | 68.09 |
| DM | $\epsilon_X = 1$ | 51.96 | 50.00 | 50.9 | 50.12 | 51.54 | 51.77 | 50.63 | 50.17 |
| | $\epsilon_X = 5$ | 52.48 | 50.00 | 51.04 | 50.14 | 51.64 | 52.08 | 50.94 | 50.21 |
| | $\epsilon_X = 10$ | 52.08 | 50.00 | 50.87 | 50.20 | 51.91 | 52.03 | 50.40 | 50.19 |
| PM | $\epsilon_X = 1$ | 51.83 | 50.00 | 51.13 | 50.09 | 52.22 | 51.95 | 50.58 | 50.20 |
| | $\epsilon_X = 5$ | 51.67 | 50.00 | 50.93 | 50.39 | 52.31 | 51.63 | 50.59 | 50.19 |
| | $\epsilon_X = 10$ | 52.62 | 50.00 | 50.98 | 50.16 | 52.21 | 52.02 | 50.49 | 50.21 |
| HM | $\epsilon_X = 1$ | 52.55 | 50.00 | 50.56 | 50.13 | 51.90 | 51.53 | 50.68 | 50.18 |
| | $\epsilon_X = 5$ | 52.07 | 50.00 | 50.72 | 50.16 | 51.65 | 51.99 | 50.61 | 50.18 |
| | $\epsilon_X = 10$ | 51.86 | 50.00 | 50.56 | 50.09 | 52.26 | 52.19 | 50.63 | 50.19 |
| Three outputs | $\epsilon_X = 1$ | 52.57 | 50.00 | 50.59 | 50.19 | 51.80 | 51.83 | 50.62 | 50.19 |
| | $\epsilon_X = 5$ | 52.07 | 50.00 | 50.75 | 50.12 | 52.14 | 51.66 | 50.60 | 50.18 |
| | $\epsilon_X = 10$ | 51.94 | 50.00 | 50.54 | 50.19 | 51.44 | 52.30 | 50.38 | 50.17 |
| PM-SUB | $\epsilon_X = 1$ | 52.44 | 50.01 | 51.10 | 50.13 | 52.02 | 51.66 | 50.74 | 50.20 |
| | $\epsilon_X = 5$ | 52.49 | 50.00 | 50.87 | 50.15 | 52.18 | 51.64 | 50.51 | 50.20 |
| | $\epsilon_X = 10$ | 51.94 | 50.00 | 50.30 | 50.19 | 52.02 | 52.28 | 50.85 | 50.19 |
| $f$-RR and Label-Laplace | $\epsilon_X = 1, \epsilon_Y = 1$ | 50.00 | 50.81 | 50.00 | 50.03 | 50.00 | 50.00 | 50.00 | 50.12 |
| | $\epsilon_X = 1, \epsilon_Y = 2.5$ | 50.00 | 50.00 | 50.00 | 51.39 | 50.00 | 50.00 | 50.00 | 50.20 |
| | $\epsilon_X = 1, \epsilon_Y = 5$ | 51.92 | 50.06 | 50.31 | 51.41 | 50.00 | 50.01 | 50.00 | 50.22 |
| | $\epsilon_X = 5, \epsilon_Y = 1$ | 50.64 | 50.00 | 50.07 | 50.01 | 50.00 | 50.39 | 50.00 | 50.21 |
| | $\epsilon_X = 5, \epsilon_Y = 2.5$ | 50.00 | 50.23 | 50.00 | 49.99 | 50.00 | 50.00 | 50.00 | 50.10 |
| | $\epsilon_X = 5, \epsilon_Y = 5$ | 51.87 | 52.40 | 50.00 | 50.00 | 50.00 | 50.00 | 50.00 | 50.21 |
| | $\epsilon_X = 10, \epsilon_Y = 1$ | 51.70 | 50.00 | 50.00 | 50.00 | 50.00 | 51.03 | 50.00 | 50.12 |
| | $\epsilon_X = 10, \epsilon_Y = 2.5$ | 50.01 | 50.12 | 50.00 | 50.00 | 50.00 | 50.05 | 51.12 | 50.16 |
| | $\epsilon_X = 10, \epsilon_Y = 5$ | 50.00 | 50.00 | 50.00 | 50.13 | 50.01 | 50.00 | 50.00 | 50.21 |
| BitRand | $\epsilon_X = 1, \epsilon_Y = \infty$ | 59.8 | 60.65 | 56.7 | 64.57 | 54.56 | 56.44 | 64.93 | 51.86 |
| | $\epsilon_X = 1, \epsilon_Y = 1$ | 54.52 | 55.03 | 53.37 | 56.99 | 52.63 | 53.34 | 57.43 | 50.91 |
| | $\epsilon_X = 1, \epsilon_Y = 2.5$ | 59.08 | 58.20 | 55.36 | 62.02 | 54.29 | 55.86 | 61.91 | 51.52 |
| | $\epsilon_X = 1, \epsilon_Y = 5$ | 59.98 | 60.15 | 56.28 | 63.67 | 54.14 | 56.91 | 63.98 | 51.77 |
| | $\epsilon_X = 5, \epsilon_Y = \infty$ | 67.15 | 71.81 | 60.65 | 77.91 | 58.16 | 61.82 | 76.15 | 54.99 |
| | $\epsilon_X = 5, \epsilon_Y = 1$ | 58.04 | 59.87 | 54.24 | 62.99 | 54.05 | 55.67 | 62.35 | 51.82 |
| | $\epsilon_X = 5, \epsilon_Y = 2.5$ | 64.61 | 67.70 | 58.78 | 73.56 | 56.2 | 60.04 | 72.09 | 53.50 |
| | $\epsilon_X = 5, \epsilon_Y = 5$ | 67.51 | 71.41 | 60.94 | 77.96 | 57.82 | 61.57 | 76.11 | 54.82 |
| | $\epsilon_X = 10, \epsilon_Y = \infty$ | 75.09 | 81.96 | 70.14 | 90.22 | 65.26 | 71.76 | 86.68 | 63.31 |
| | $\epsilon_X = 10, \epsilon_Y = 1$ | 61.87 | 64.88 | 59.04 | 68.82 | 56.8 | 60.33 | 67.20 | 53.84 |
| | $\epsilon_X = 10, \epsilon_Y = 2.5$ | 71.35 | 76.69 | 66.88 | 84.48 | 62.74 | 68.33 | 81.52 | 58.16 |
| | $\epsilon_X = 10, \epsilon_Y = 5$ | 74.45 | 82.28 | 69.23 | 89.88 | 64.81 | 71.47 | 86.38 | 62.25 |

Table 4: AUC values of each algorithm applied on the gradients $\bigtriangledown_{\theta_t}^u$ with the anonymizer (Sun et al., 2021) in the CelebA dataset.

| Algorithm [$\bigtriangledown_{\theta_t}^u$ with Anon] / Attribute | Attractive | Heavy Makeup | High Cheekbones | Male | Mouth Slightly Open | Smiling | Lipstick | **Average** |
|---|---|---|---|---|---|---|---|---|
| Noiseless — $\epsilon_X = \infty$ | 78.05 | 85.47 | 76.53 | 92.77 | 72.91 | 79.15 | 88.85 | 68.09 |
| DM — $\epsilon_X = 1$ | 43.44 | 55.86 | 49.63 | 65.42 | 48.86 | 43.33 | 60.50 | 49.85 |
| DM — $\epsilon_X = 5$ | 46.94 | 44.25 | 49.00 | 49.68 | 49.97 | 51.73 | 59.24 | 51.00 |
| DM — $\epsilon_X = 10$ | 38.60 | 40.81 | 51.87 | 64.81 | 49.55 | 49.05 | 48.58 | 50.25 |
| PM — $\epsilon_X = 1$ | 47.48 | 62.64 | 53.94 | 49.60 | 50.88 | 49.07 | 52.04 | 50.81 |
| PM — $\epsilon_X = 5$ | 53.65 | 67.99 | 47.72 | 44.75 | 50.13 | 52.34 | 43.20 | 49.72 |
| PM — $\epsilon_X = 10$ | 56.85 | 45.04 | 52.46 | 48.42 | 50.15 | 49.16 | 49.81 | 50.30 |
| HM — $\epsilon_X = 1$ | 49.61 | 63.99 | 52.06 | 47.27 | 51.90 | 48.43 | 38.25 | 50.05 |
| HM — $\epsilon_X = 5$ | 53.39 | 42.84 | 47.26 | 59.54 | 53.23 | 55.21 | 67.67 | 51.54 |
| HM — $\epsilon_X = 10$ | 44.02 | 68.26 | 51.07 | 54.43 | 50.17 | 49.91 | 44.04 | 50.42 |
| Three outputs — $\epsilon_X = 1$ | 54.71 | 43.89 | 55.70 | 55.37 | 48.46 | 54.93 | 43.11 | 49.99 |
| Three outputs — $\epsilon_X = 5$ | 35.45 | 65.80 | 50.13 | 42.53 | 50.21 | 45.62 | 50.36 | 51.54 |
| Three outputs — $\epsilon_X = 10$ | 48.53 | 46.76 | 50.03 | 51.96 | 53.64 | 49.57 | 31.48 | 49.21 |
| PM-SUB — $\epsilon_X = 1$ | 41.65 | 51.61 | 50.05 | 63.01 | 50.35 | 49.02 | 51.05 | 50.29 |
| PM-SUB — $\epsilon_X = 5$ | 47.62 | 43.88 | 50.86 | 54.82 | 51.76 | 53.90 | 46.81 | 51.79 |
| PM-SUB — $\epsilon_X = 10$ | 40.82 | 74.82 | 49.86 | 55.00 | 54.19 | 52.39 | 45.04 | 51.34 |
| LDP-FL — $\epsilon_X = 1$ | 41.51 | 49.43 | 50.24 | 33.03 | 49.01 | 49.40 | 53.76 | 48.71 |
| LDP-FL — $\epsilon_X = 5$ | 50.35 | 53.27 | 51.99 | 56.76 | 49.71 | 50.02 | 56.31 | 51.74 |
| LDP-FL — $\epsilon_X = 10$ | 46.71 | 46.90 | 50.11 | 52.04 | 47.74 | 48.50 | 49.42 | 50.28 |
| $f$-RR and Label-Laplace — $\epsilon_X = 1, \epsilon_Y = 1$ | 50.00 | 50.81 | 50.00 | 50.03 | 50.00 | 50.00 | 50.00 | 50.12 |
| $f$-RR and Label-Laplace — $\epsilon_X = 1, \epsilon_Y = 2.5$ | 50.00 | 50.00 | 50.00 | 51.39 | 50.00 | 50.00 | 50.00 | 50.20 |
| $f$-RR and Label-Laplace — $\epsilon_X = 1, \epsilon_Y = 5$ | 51.92 | 50.06 | 50.31 | 51.41 | 50.00 | 50.01 | 50.00 | 50.22 |
| $f$-RR and Label-Laplace — $\epsilon_X = 5, \epsilon_Y = 1$ | 50.64 | 50.00 | 50.07 | 50.01 | 50.00 | 50.39 | 50.00 | 50.21 |
| $f$-RR and Label-Laplace — $\epsilon_X = 5, \epsilon_Y = 2.5$ | 50.00 | 50.23 | 50.00 | 49.99 | 50.00 | 50.00 | 50.00 | 50.10 |
| $f$-RR and Label-Laplace — $\epsilon_X = 5, \epsilon_Y = 5$ | 51.87 | 52.40 | 50.00 | 50.00 | 50.00 | 50.00 | 50.00 | 50.21 |
| $f$-RR and Label-Laplace — $\epsilon_X = 10, \epsilon_Y = 1$ | 51.70 | 50.00 | 50.00 | 50.00 | 50.00 | 51.03 | 50.00 | 50.12 |
| $f$-RR and Label-Laplace — $\epsilon_X = 10, \epsilon_Y = 2.5$ | 50.01 | 50.12 | 50.00 | 50.00 | 50.00 | 50.05 | 51.12 | 50.16 |
| $f$-RR and Label-Laplace — $\epsilon_X = 10, \epsilon_Y = 5$ | 50.00 | 50.00 | 50.00 | 50.13 | 50.01 | 50.00 | 50.00 | 50.21 |
| BitRand — $\epsilon_X = 1, \epsilon_Y = \infty$ | 59.8 | 60.65 | 56.7 | 64.57 | 54.56 | 56.44 | 64.93 | 51.86 |
| BitRand — $\epsilon_X = 1, \epsilon_Y = 1$ | 54.52 | 55.03 | 53.37 | 56.99 | 52.63 | 53.34 | 57.43 | 50.91 |
| BitRand — $\epsilon_X = 1, \epsilon_Y = 2.5$ | 59.08 | 58.20 | 55.36 | 62.02 | 54.29 | 55.86 | 61.91 | 51.52 |
| BitRand — $\epsilon_X = 1, \epsilon_Y = 5$ | 59.98 | 60.15 | 56.28 | 63.67 | 54.14 | 56.91 | 63.98 | 51.77 |
| BitRand — $\epsilon_X = 5, \epsilon_Y = \infty$ | 67.15 | 71.81 | 60.65 | 77.91 | 58.16 | 61.82 | 76.15 | 54.99 |
| BitRand — $\epsilon_X = 5, \epsilon_Y = 1$ | 58.04 | 59.87 | 54.24 | 62.99 | 54.05 | 55.67 | 62.35 | 51.82 |
| BitRand — $\epsilon_X = 5, \epsilon_Y = 2.5$ | 64.61 | 67.70 | 58.78 | 73.56 | 56.2 | 60.04 | 72.09 | 53.50 |
| BitRand — $\epsilon_X = 5, \epsilon_Y = 5$ | 67.51 | 71.41 | 60.94 | 77.96 | 57.82 | 61.57 | 76.11 | 54.82 |
| BitRand — $\epsilon_X = 10, \epsilon_Y = \infty$ | 75.09 | 81.96 | 70.14 | 90.22 | 65.26 | 71.76 | 86.68 | 63.31 |
| BitRand — $\epsilon_X = 10, \epsilon_Y = 1$ | 61.87 | 64.88 | 59.04 | 68.82 | 56.8 | 60.33 | 67.20 | 53.84 |
| BitRand — $\epsilon_X = 10, \epsilon_Y = 2.5$ | 71.35 | 76.69 | 66.88 | 84.48 | 62.74 | 68.33 | 81.52 | 58.16 |
| BitRand — $\epsilon_X = 10, \epsilon_Y = 5$ | 74.45 | 82.28 | 69.23 | 89.88 | 64.81 | 71.47 | 86.38 | 62.25 |

Table 5: AUC values of each algorithm applied on $\bigtriangledown_{\theta_t}^{u}$ in the CelebA dataset. **Average** is the average of all 40 attributes.

| Algorithm [$\bigtriangledown_{\theta_t}^{u}$ with Anon] Attribute | Attractive | Heavy Makeup | High Cheekbones | Male | Mouth Slightly Open | Smiling | Lipstick | **Average** |
|---|---|---|---|---|---|---|---|---|
| Noiseless | $\epsilon_X = \infty$ | 78.05 | 85.47 | 76.53 | 92.77 | 72.91 | 79.15 | 88.85 | 68.09 |
| DM | $\epsilon_X = 1$ | 57.96 | 29.57 | 46.45 | 61.43 | 50.58 | 50.9 | 47.61 | 47.96 |
| | $\epsilon_X = 5$ | 59.85 | 39.72 | 51.01 | 53.36 | 51.15 | 49.97 | 32.64 | 48.62 |
| | $\epsilon_X = 10$ | 56.33 | 44.01 | 50.05 | 55.19 | 49.8 | 52.05 | 30.79 | 50.00 |
| PM | $\epsilon_X = 1$ | 51.87 | 39.38 | 49.3 | 73.24 | 49.41 | 48.9 | 52.99 | 49.66 |
| | $\epsilon_X = 5$ | 48.42 | 40.86 | 46.84 | 72.35 | 50.55 | 49.52 | 47.68 | 50.66 |
| | $\epsilon_X = 10$ | 49.02 | 36.59 | 49.03 | 59.12 | 50.39 | 50.46 | 53.36 | 49.22 |
| HM | $\epsilon_X = 1$ | 46.31 | 33.84 | 52.46 | 42.45 | 48.33 | 50.98 | 48.23 | 50.41 |
| | $\epsilon_X = 5$ | 52.39 | 34.32 | 52.49 | 29.79 | 50.02 | 48.04 | 49.90 | 49.37 |
| | $\epsilon_X = 10$ | 53.3 | 49.65 | 52.23 | 35.33 | 50.82 | 47.69 | 47.19 | 50.33 |
| Three outputs | $\epsilon_X = 1$ | 45.91 | 63.71 | 50.16 | 61.60 | 51.35 | 45.39 | 61.21 | 50.58 |
| | $\epsilon_X = 5$ | 35.89 | 55.43 | 48.95 | 55.22 | 47.60 | 46.30 | 58.41 | 50.39 |
| | $\epsilon_X = 10$ | 35.03 | 60.51 | 49.85 | 60.69 | 47.70 | 43.97 | 54.05 | 49.88 |
| PM-SUB | $\epsilon_X = 1$ | 61.56 | 63.02 | 46.78 | 38.77 | 52.23 | 50.38 | 48.60 | 48.90 |
| | $\epsilon_X = 5$ | 58.59 | 57.14 | 48.93 | 31.43 | 50.54 | 53.93 | 51.75 | 47.93 |
| | $\epsilon_X = 10$ | 65.09 | 55.82 | 47.10 | 33.13 | 49.70 | 51.11 | 59.87 | 49.61 |
| $f$-RR and Label -Laplace | $\epsilon_X = 1, \epsilon_Y = 1$ | 50.00 | 50.81 | 50.00 | 50.03 | 50.00 | 50.00 | 50.00 | 50.12 |
| | $\epsilon_X = 1, \epsilon_Y = 2.5$ | 50.00 | 50.00 | 50.00 | 51.39 | 50.00 | 50.00 | 50.00 | 50.20 |
| | $\epsilon_X = 1, \epsilon_Y = 5$ | 51.92 | 50.06 | 50.31 | 51.41 | 50.00 | 50.01 | 50.00 | 50.22 |
| | $\epsilon_X = 5, \epsilon_Y = 1$ | 50.64 | 50.00 | 50.07 | 50.01 | 50.00 | 50.39 | 50.00 | 50.21 |
| | $\epsilon_X = 5, \epsilon_Y = 2.5$ | 50.00 | 50.23 | 50.00 | 49.99 | 50.00 | 50.00 | 50.00 | 50.10 |
| | $\epsilon_X = 5, \epsilon_Y = 5$ | 51.87 | 52.40 | 50.00 | 50.00 | 50.00 | 50.00 | 50.00 | 50.21 |
| | $\epsilon_X = 10, \epsilon_Y = 1$ | 51.70 | 50.00 | 50.00 | 50.00 | 50.00 | 51.03 | 50.00 | 50.12 |
| | $\epsilon_X = 10, \epsilon_Y = 2.5$ | 50.01 | 50.12 | 50.00 | 50.00 | 50.00 | 50.05 | 51.12 | 50.16 |
| | $\epsilon_X = 10, \epsilon_Y = 5$ | 50.00 | 50.00 | 50.00 | 50.13 | 50.01 | 50.00 | 50.00 | 50.21 |
| BitRand | $\epsilon_X = 1, \epsilon_Y = \infty$ | 59.8 | 60.65 | 56.7 | 64.57 | 54.56 | 56.44 | 64.93 | 51.86 |
| | $\epsilon_X = 1, \epsilon_Y = 1$ | 54.52 | 55.03 | 53.37 | 56.99 | 52.63 | 53.34 | 57.43 | 50.91 |
| | $\epsilon_X = 1, \epsilon_Y = 2.5$ | 59.08 | 58.20 | 55.36 | 62.02 | 54.29 | 55.86 | 61.91 | 51.52 |
| | $\epsilon_X = 1, \epsilon_Y = 5$ | 59.98 | 60.15 | 56.28 | 63.67 | 54.14 | 56.91 | 63.98 | 51.77 |
| | $\epsilon_X = 5, \epsilon_Y = \infty$ | 67.15 | 71.81 | 60.65 | 77.91 | 58.16 | 61.82 | 76.15 | 54.99 |
| | $\epsilon_X = 5, \epsilon_Y = 1$ | 58.04 | 59.87 | 54.24 | 62.99 | 54.05 | 55.67 | 62.35 | 51.82 |
| | $\epsilon_X = 5, \epsilon_Y = 2.5$ | 64.61 | 67.70 | 58.78 | 73.56 | 56.2 | 60.04 | 72.09 | 53.50 |
| | $\epsilon_X = 5, \epsilon_Y = 5$ | 67.51 | 71.41 | 60.94 | 77.96 | 57.82 | 61.57 | 76.11 | 54.82 |
| | $\epsilon_X = 10, \epsilon_Y = \infty$ | 75.09 | 81.96 | 70.14 | 90.22 | 65.26 | 71.76 | 86.68 | 63.31 |
| | $\epsilon_X = 10, \epsilon_Y = 1$ | 61.87 | 64.88 | 59.04 | 68.82 | 56.8 | 60.33 | 67.20 | 53.84 |
| | $\epsilon_X = 10, \epsilon_Y = 2.5$ | 71.35 | 76.69 | 66.88 | 84.48 | 62.74 | 68.33 | 81.52 | 58.16 |
| | $\epsilon_X = 10, \epsilon_Y = 5$ | 74.45 | 82.28 | 69.23 | 89.88 | 64.81 | 71.47 | 86.38 | 62.25 |

