# OpenReview forum: "Bit-aware Randomized Response for Local Differential Privacy in Federated Learning"
_ICLR.cc/2022/Conference — ICLR 2022 Submitted_

### Official Review · Reviewer_yGHK · 2021-11-02

**Correctness:** 3
**Technical Novelty And Significance:** 2
**Empirical Novelty And Significance:** 2
**Recommendation:** 3
**Confidence:** 4

**Main Review:**

1. Though the authors name their mechanism as $f$-RR and claim its novelty, it is indeed an asymmetric version of RAPPOR that has been studied previously in the context of frequency estimation (see [WBLJ 2019] for instance). The only difference is the specific choice of parameters.


2. This paper tries to apply frequency estimation techniques (such as RR or RAPPOR), which are usually designed for categorical data, into privatizing continuous data. Therefore the proposed mechanism can be viewed as first quantizing the data (i.e., mapping a vector $v \in \mathbb{R}^r$ to $r$ binary strings), and then applying RAPPOR on it. However, recent works have shown that such a naive two-stage approach is usually highly sub-optimal (at least in several canonical tasks such as mean estimation, see [CKO 2020]). Therefore I highly doubt the optimality of the proposed algorithm.


3. The authors claim that the proposed solution resolves the curse of privacy composition; however, from the main utility guarantee Theorem 4, I don't think the problem is solved. Theorem 4 only gives an error bound on a single coordinate, so when applying the algorithm to a $r$-dim vector, there should be an additional $r$ factor. In general, I don't think it is possible to improve the state-of-the-art LDP mechanisms (such as [BDFKR 2019]) as performance guarantees of these mechanisms (order-wisely) match the information-theoretic lower bounds. It would also be good if the authors can provide standard error guarantees for their algorithm (e.g. the $\ell_1/\ell_2$ error for mean estimation) and compare them with previous works.


4. I also doubt the correctness of the experimental results. For instance, upon a quick look at the codes, the "Duchi Mechanism" (which, I believe, should be the privUnit mechanism in [BDFKR 2019]) is indeed implemented as [DJW 2013]. Note that [DJW 2013] is a weaker version of privUnit and is only optimal when $\varepsilon = O(1)$. It would be good if the authors can include more implementation details in the appendix.



[WBLJ 2019] Locally differentially private protocols for frequency estimation

[CKO 2020] Breaking the Communication-Privacy-Accuracy Trilemma

[BDFKR 2019] Protection Against Reconstruction and Its Applications in Private Federated Learning

[DJW 2013] Minimax optimal procedures for locally private estimation

**Summary Of The Paper:**

The paper proposes an $\epsilon$-LDP mechanism, termed $f$-RR, to privatize a real number. The mechanism first maps $v$ into its binary representation (with a sign bit) and then flips each bit with probability $p_i$. The authors then extend their mechanism to privatize a $r$-dimensional vector by sequentially performing the (scalar version) algorithm for each of the $r$ coordinates. By picking proper parameters, the proposed mechanism preserves $\epsilon$-LDP. Finally, several experimental results compared with other previous methods are provided.

**Summary Of The Review:**

In general, the proposed privatization scheme is not novel, and the analysis seems not optimal. I appreciate the authors' efforts in conducting extensive experiments, but there might be some issues when comparing the proposed algorithm with previous works, which I hope the authors can clarify.

---

> ### Author Response · Authors · 2021-11-22
> **Response to Reviewer yGHK - Part 2**
>
> Q3. Resolving the curse of privacy composition. Comparison with other methods based on standard error guarantees for mean estimation.
>
> A: We agree with the reviewer that we mitigate the curse of privacy composition, rather than totally resolve the problem. We have toned it down in the revision.
>
> As requested by the reviewer, we conducted one experiment for mean estimation and compared with other mechanisms. We created a synthetic data that consists of $N=1,000$ data samples $x_i (i=[1,N])$, each of them has $d=768$ dimensions.  The mean estimation is calculated over each dimension as $f_j(D) = \frac{1}{N} \sum_{i=1}^N x_{ij}$ for $j\in [1,d]$.  Root mean square error (RMSE) is used to evaluate the error between the original vector and the randomized/estimated vector. The binary-encoding-based approaches (i.e., $f$-RR, corrected LATENT, and corrected OME) achieve a significantly small error compared with others. Among them, $f$-RR obtains the smallest error, which further shows the effectiveness of our proposed mechanism.
>
> The results are as follows:
>
>  $\epsilon$      $|1|2|3|4|5|6|7|8|9|10$
>
> DM    $|51.14| 24.13| 16.55| 12.13|  9.80|  7.92|
>   6.66| 6.29| 5.16| 4.98$
>
> PM         $|59.11| 26.86| 18.92| 13.46| 10.99| 9.40| 8.30| 6.93| 6.30| 5.61$
>
> HM $| 50.88| 24.88| 15.99| 12.03| 9.58| 7.86| 6.71| 6.02| 5.47| 4.90$
>
> Three-outputs    $|48.10| 25.31| 16.21| 12.07| 9.59| 8.18| 6.71| 6.45| 5.39| 5.02$
>
> PM-SUB         $|55.89| 27.47| 19.26| 14.41| 11.06| 8.83| 8.39| 6.78| 6.30| 5.48$
>
> corrected LATENT $| 0.205| 0.204| 0.201| 0.199| 0.197| 0.194| 0.192| 0.190| 0.188| 0.185$
>
> corrected OME    $|0.093| 0.090| 0.093| 0.097| 0.097| 0.102| 0.105| 0.111| 0.114| 0.120$
>
> $f$-RR        $|0.031| 0.029| 0.027| 0.020| 0.012| 0.007| 0.004| 0.003| 0.002| 0.001|$
>
> Q4. Correctness of the experimental results, i.e., Duchi mechanism.
>
> A: Thanks the reviewer for pointing it out. We agree that the implementation is mainly based on [DJW 2013] than [BDFKR 2019]. We will revise this reference in the revision.
>
> However, the code for [BDFKR 2019] is not publicly available. We have attempted to contact the authors of [BDFKR 2019] several times for the code; but the authors never responded to our requests. We hope that if the authors of [BDFKR 2019] (by any chance) read our rebuttal, please share the legitimate code of the significant work in [BDFKR 2019] with us and the community. We do appreciate! We are aware that this issue happens with other research teams too.
>
> From our side, we could not reproduce the results reported in [BDFKR 2019] after many attempts. Also, we notice that the privacy budget $\epsilon$ used in [BDFKR 2019] is remarkably large (e.g., $\epsilon \in [100, 500, 1,000, 5,000]$) in the applications of federated learning on MNIST, CIFAR-10, Flicrk, and Reddit datasets. That indicates a loose privacy protection as pointed out in [Nasr et al. 2021]. Compared with [BDFKR 2019], our mechanism can achieve high model utility under tight privacy budgets, i.e., $\epsilon \in [1, 10]$.
>
> [BDFKR 2019] Bhowmick, A., Duchi, J., Freudiger, J., Kapoor, G., and Rogers, R. (2018). Protection against reconstruction and its applications in private federated learning. arXiv preprint arXiv:1812.00984.
>
> [Nasr et al. 2021] Nasr, M., Song, S., Thakurta, A., Papernot, N., and Carlini, N. (2021). Adversary instantiation: Lower bounds for differentially private machine learning. IEEE SP 2021.

---

> ### Author Response · Authors · 2021-11-22
> **Response to Reviewer yGHK - Part 1**
>
> We thank the reviewer for the constructive and helpful comments. Following are our responses:
>
> Q1. Asymmetric version of RAPPOR with specific choice of parameters.
>
> A: Asymmetric version of RAPPOR (e.g., [WBLJ 2019]) designs different randomization probabilities for different inputs. The technique is well-applied in the context of frequency estimation and successfully reduce the communication cost from $O(d)$ to $O(\log n)$ ($d$ is data dimension and $n$ is the number of samples). However, simply applying the mechanism [WBLJ 2019] does not optimize the model utility and the privacy-utility trade-off when working with machine learning or deep learning models.
>
> Different from the asymmetric version of RAPPOR, our proposed $f$-RR mechanism focuses on mitigating the privacy-utility trade-off. To achieve that, besides the asymmetric nature of the randomization probabilities, our designed $f$-RR consists of two key components: 1) The bit-aware term $i\%l/l$, which indicates the location of the bit i in each embedded feature associated with the sensitivity of the bit at that location; and 2) The adjustable but bounded $\alpha$, which takes into account the correlation between privacy loss and the sensitivity of embedded features to mitigate the privacy-utility trade-off and the curse of privacy composition.
>
> The bit-aware property refers to the bits with a more substantial influence on the model utility have smaller randomization probabilities, and vice-versa, under the same privacy protection. By incorporating sensitivities of binary encoding bits into a generalized privacy loss bound, we show that increasing the dimensions of embedded features $r$, encoding bits $l$, and model outcomes $C$ marginally affect the randomization probabilities in BitRand under the same privacy budget. This dimension-elastic property is crucial to mitigate the curse of privacy composition by retaining a high value of data transmitted correctly through our randomization given large dimensions of $r$, $l$, and $C$.
>
> Besides the $f$-RR for protecting the data, we also include the label-RR for protecting the label in our proposed BitRand mechanism, that provides a complete protection for every data sample.
>
>
> Q2. This paper tries to apply frequency estimation techniques. Therefore the proposed mechanism can be viewed as first quantizing the data, and then applying RAPPOR on it. However, recent works have shown that such a naive two-stage approach is usually highly sub-optimal (at least in several canonical tasks such as mean estimation, see [CKO 2020])
>
> A: Our work is totally different from applying frequency estimation techniques or mean estimation. Our approach focuses on solving the learning task, not the mean of the distribution or the frequency estimation.
>
> However, we further provide a comparison between our approach and baselines given the mean estimation task in the below answer (for the Q3). Among all the baselines, $f$-RR obtains the smallest error, which further shows the effectiveness of our proposed mechanism.

---

> ### Comment · Reviewer_yGHK · 2021-11-29
> **Response**
>
> I thank the authors' detailed response as well as providing new experimental results. My evaluation basically remains the same.
>
> I still think the basic idea behind $f$-RR and RAPPOR are similar. In a nutshell, for a given binary vector, RAPPOR flips each bit with a certain probability such that the perturbed vector satisfies DP. The only differences compared to $f$-RR are 1) $f$-RR considers a specific binary representation and 2) the privacy budget allocated to each bit is optimized under this representation. Therefore I cannot say the proposed method is novel.
>
> Regarding the additional mean estimation results provided by the authors in the response, it is hard to believe that the $f$-RR can outperform privUnit or DJW2014 (which the authors label as DM). Both privUnit and DJW are proved to be optimal for mean estimation and there are information-theoretic lower bounds.

---

> > ### Author Response · Authors · 2021-11-29
> > **Response to Reviewer yGHK**
> >
> > We thank the reviewer for the response.
> >
> > Question: The novelty of f-RR and RAPPOR
> >
> > A: RAPPOR and our $f$-RR follow the same spirit of randomized responses in LDP in which flipping each bit with a certain probability such that the perturbed vector satisfies LDP. However, as pointed out by the reviewer that there are key differences between our $f$-RR and RAPPOR.
> >
> > Our approach optimized the privacy budget allocated to each bit compared with RAPPOR. This optimization is non-trivial, not straightforward, and is a clear step forward compared with baseline approaches. That highlights the novelty of our approach. More importantly, our approach achieves notably better results compared with baseline approaches in both theoretical and empirical results.
> >
> > Question: it is hard to believe that the f-RR can outperform privUnit or DJW2014 (which the authors label as DM)
> >
> > A: As long as the correctness and integrity of our approach are valid, our results are solid and convincing! We also included our implementations in our submission and revision for reproducing the results.
> >
> > Although information-theoretic lower bounds in privUnit or DJW2014 are important, it is worth noting that there is no theoretical proof that there will definitely not exist any better lower bounds or any better mechanisms compared with the lower bounds provided in privUnit or DJW2014. That is the reason why the community is still actively investigating in this direction for better solutions.

---

### Official Review · Reviewer_S5g9 · 2021-11-02

**Correctness:** 3
**Technical Novelty And Significance:** 3
**Empirical Novelty And Significance:** 3
**Recommendation:** 5
**Confidence:** 3

**Main Review:**


Pros:

I think the idea seems natural and as far as I see the LDP mechanism that takes into account the signifigance of the digit in the noise variance of the randomiser has not been considered before (good discussion of related literature).

The paper is well written and all the math that I checked seems correct.


Cons:

I find the paper a bit difficult to read, everything seems to be discussed on a very low level. That is of course not a deficit per se, but a bit more higher level view on the problem would help, I think.

Example: It is totally possible I missed something, but it is unclear to me, why do you only consider the LDP mechanism for the embedding and then observe the error in the decoded representation (that mathcal(E) representing the decoder) ? I mean, if the gist of the paper is that bit-aware LDP randomiser, why not to also have utility comparisons for that (more) directly ?

I think the presentation could be improved. Examples:
In Section 5, Figure 3: This figure is not referred to anywhere, nor are the labels of the figure. You mean that you measure the expected error of the decoded features so that f-RR is replaced by Gaussian mechanism etc.?
In Figure 3, what is the sensitivity for the Gaussian mechanism, for example? sqrt(r) ?
And for the Laplace mechanism? How do you use the Gaussian mechanism for the binary r-vector?

Other:

-The figures are quite small and thus a bit hard to read.
-Would it be clearer if Fig. 7 had logarithmic y-axis?
-When talking about bit-wise DP, I think it would be good to mention also the paper
Mironov, Ilya. On significance of the least significant bits for differential privacy. In: Proceedings of the 2012 ACM conference on Computer and communications security. 2012. p. 650-661.

EDIT: I think the paper has its merits and the bit-wise mechanism is an interesting approach, however I think the paper is not yet mature for publication in ICLR. Some of my concerns regarding the presentation (e.g. why is only the privatisation of the embedded representation studied) seem to be shared also by other reviewers. Thus, I have decided to keep my score.

**Summary Of The Paper:**

The importance of rigorously taking care of privacy leakage in a DP algorithm on the bit level is well-know, see e.g.

Mironov, Ilya. On significance of the least significant bits for differential privacy. In: Proceedings of the 2012 ACM conference on Computer and communications security. 2012. p. 650-661.

This paper considers locally differentially private (LDP) mechanisms for binary representations of floating point numbers. The technique seems like a natural thing to do: the individual bits are flipped with a probability that scales with their importance.


**Summary Of The Review:**

An interesting LDP randomiser that takes into account the significance of bits in binary encoding of floating point numbers (and vectors). However, I think the presentation should be improved to meet the bar of ICLR. Currently, I think this paper will not be easily understandable to anyone else than those working close to this i.e. practical algorithms for LDP and federated learning.

---

> ### Author Response · Authors · 2021-11-22
> **Response to Reviewer S5g9**
>
> We thank the reviewer for the constructive and helpful comments. Following are our responses:
>
> Q1. Why only consider the LDP mechanism for the embedding and then observe the error in the decoded representation? Why not to have utility comparisons directly?
>
> A: There are two key reasons why we performed privacy and utility analysis w.r.t. the encoding/decoding $\mathcal{E}$ instead of $f$-RR, as discussed next. In our threat model, the attacker aims to infer the original value of the randomized embedded features (extracted by applying gradients attack [Yin et al., 2021; Zhao et al., 2020; Zhu et al., 2019]). Therefore, we have to guarantee that the randomized responses applied on the binary encoding vector has to provide privacy protection at the embedded feature space.
> Second, different from RAPPOR, in which every bit is independent to each other and has the same sensitivity (sensitivity equals $1$), the sensitivity of bits in our $f$-RR are different given the embedded feature domain (Lemma 1). Therefore, solely conducting privacy analysis at the bit level of $f$-RR (i.e., sensitivity equals 1) is insufficient to provide protection at the embedded feature space. Consequently, performing the privacy analysis given the encoding/decoding $\mathcal{E}$ is necessary to provide $\epsilon_X$-LDP for the embedded features.
>
> [Yin et al., 2021] Yin, H., Mallya, A., Vahdat, A., Alvarez, J. M., Kautz, J., & Molchanov, P. (2021). See through Gradients: Image Batch Recovery via GradInversion. CVPR (pp. 16337-16346).
>
> [Zhao et al., 2020] Zhao, B., Mopuri, K. R., & Bilen, H. (2020). idlg: Improved deep leakage from gradients. arXiv preprint arXiv:2001.02610.
>
> [Zhu et al., 2019] Zhu, L., & Han, S. (2020). Deep leakage from gradients. In Federated learning (pp. 17-31). NeurIPS 2019.
>
> Q2. Sensitivity of the Gaussian and Laplace mechanism.
>
> A: The Gaussian and Laplace mechanisms naturally apply an addition operation, which add noise into the data or embedded features. Therefore, in our analysis of expected error bound comparison for an embedded feature (Figure 3), we add noise into the embedded feature following the Gaussian and Laplace mechanisms. The sensitivity captures the magnitude by which an embedded feature can change in the worst case. In our experiment and analysis, we use $l=10$ bits in which $1$ sign bit, $5$ bits for the integer, and $4$ bits for the fraction part. Therefore, the maximum the embedded feature can be change, i.e., the sensitivity, is  $2\sum_{i=-4}^4 2^i$. Note that, we multiply $\sum_{i=-4}^4 2^i$ by $2$ since when we flip the sign bit, it significantly changes the value of the embedded feature from $-a$ to $a$ in which $a = \sum_{i=-4}^4 2^i$.
>
> Q3. Figure 3 reference and figures representation.
>
> A: We refer to Figure 3 and its analysis in the second paragraph of page 6 (Section 5). Thanks the reviewer for the thorough suggestion. We have increased the size of the figures and change Fig.7 to logarithm y-axis.
>
> Q4. Reference to [Mironov 2012] paper.
>
> A: The [Mironov 2012] paper focuses on addressing the floating-point arithmetic in implementation of DP applications.
> The inconsistency between mathematical abstraction of Laplace mechanism with sampling ``uniform" floating-point numbers can be exploited to carry out privacy attacks. Floating-point arithmetic is a leaky abstraction, which is ubiquitous in computer systems and is difficult to argue about formally and hard to get right in applications, including all the RR mechanisms.
>
> However, our work does not focus on solving the floating-point arithmetic in programming languages. We focus on optimizing the trade-off between privacy protection and model utility when working with machine learning or deep learning models. Implementation is programming language-agnostic in our mechanism.
>
> We have clarified the difference between our work with [Mironov 2012] in our revision.
>
> [Mironov 2012] Mironov, I. On significance of the least significant bits for differential privacy. In the 2012 ACM conference on Computer and communications security (pp. 650-661).

---

> > ### Comment · Reviewer_S5g9 · 2021-11-23
> > **Response**
> >
> > I thank the authors for addressing my concerns. I understand that it is crucial to analyse the privacy of the embedded representations in case the threat model assumes that the adversary has access to those, however I think that the proposed bit-sensitive mechanism has components that would be beneficial is other settings as well (independent of encoding/decoding setting) and I think it could be compared to other methods without the embedding as well, that is what I meant. However, for this specific setting of adversary having access to the encoding I think it makes sense to carry out the analysis this way.

---

> > > ### Author Response · Authors · 2021-11-23
> > > **We are thankful and happy to address the reviewer comments!**
> > >
> > > We are glad and happy to address all the constructive comments from the reviewer. We do agree that our mechanism can be applied to other settings, making the broader impacts of our works beyond the embedded features. The natural extension of our work would be extensively evaluating our mechanism in a variety of privacy-preserving applications. We reserve this for future work since we would like to keep our paper concentrated on optimizing the fundamental trade-off between privacy and model utility in federated learning.
> > >
> > > The findings in our paper are exciting and significant, departing from existing works! We hope that the reviewer enjoys reading our work and look forwards to further discussion to make our work even stronger and more meaningful.

---

> ### Author Response · Authors · 2021-11-30
> **Only the privatization of the embedded representation.**
>
> We thank the reviewer for the reviewer's EDIT. Regarding the privatization of the embedded representation, we have comprehensively addressed it in our rebuttal, which is confirmed to be sufficient by the reviewer Sanq.
>
> Also, in your response, the reviewer clearly stated that carrying out the privacy analysis on the embedding is reasonable!

---

### Official Review · Reviewer_Sanq · 2021-11-02

**Correctness:** 3
**Technical Novelty And Significance:** 3
**Empirical Novelty And Significance:** Not applicable
**Recommendation:** 6
**Confidence:** 3

**Main Review:**

+ The paper provides a simple but useful trick of tweaking the bit randomization probabilities based on the bit-index that results in "dimension-elastic" privacy/utility gains.
+ The paper is in general well written and easy to follow.
+ The comparison with prior research is very comprehensive along with strong experimental evaluation.

Comments.
1. I am a bit confused as to why is the privacy analysis is performed w.rt the encoding/decoding $\mathcal{E}$ instead of just $f-RR$.As  mentioned, the sensitivity of $f-RR$ is lower - 1 and since $\mathcal{E}$ is applied to the output of f-RR,  by post-processing guarantee, LDP would still hold. Is the reason for choosing to analysis for $\mathcal{E}(f-RR)$ the fact that the gradient computations require the decoding process and hence, this results in better utility by directly analysing w.r.t to $\mathcal{E}$?
2. Could the authors provide some details about motivation/purpose of the embedding step? Is it a smarter approach for dimensionality reduction or the embedded features carry some intrinsic meaning? Also, how integral is the embedding step to the proposed scheme - would the privacy/utility gains hold true if Alg.1 is applied directly on the input $x$?
3. How is the proposed different label-RR scheme different from the RR techniques showcased by Ghazi et al. 2021? Also , why is the evaluation baseline  Label-Laplace instead of Ghazi et al.?
4. Why is the sign bit valued at 2^{m+1} and not 2^m (since the MSB integer bit is 2^{m-1})? Also, is there any specific reason why the sign bit is assigned the highest weightage?
5. Why is $\epsilon_i$ intractable (just below Eq. 6)? Is it not a user defined privacy parameter whose value is an input to the algo?
6.  Is not $|\mathcal{E}(f-RR(v_a,i)-\mathcal{E}(f-RR(v_{a|i},i)|$ randomized (since f-RR is)? How does this imply $|\mathcal{E}(f-RR(v_a,i)-\mathcal{E}(f-RR(v_{a|i},i)| < |f-RR(v_a,i)-f-RR(v_{a|i},i)|$?
7. I could not understand why $P(f-RR(v_{a}(i)=v_z(i)))/P(f-RR(v_{a|i}(i))=v_z(i)) )\geq 1$ is the worst case?
8. I could not understand how is the anonymization work by Sun et a.l applied in the evaluation - exactly what is being compared against?
9. Anonymization (shufflers) can be also implemented by cryptographic techniques such as mixed nets. Additionally, most of the work on shuffling DP provide amplification results that can work with using the same randomization mechanism for all. So I would suggest toning down the relevant sentences in Page 2.

**Summary Of The Paper:**

The paper proposes a randomized-response algorithm that takes into the bit indices and prioritises higher order bits. As a result, the utility is higher than other competitive algorithms. Additionally, the analysis allows the bit randomization probabilities not to be affected too much by the dimension of the data.

**Summary Of The Review:**

Overall, the paper provides an effective technique for improved utility for RR. However, I have a few confusions regarding the solution setting (see comments above).

---

> ### Author Response · Authors · 2021-11-22
> **Response to Reviewer Sanq - Part 4**
>
>
> Q7. Why $\frac{P(f\text{-RR}(v_a(i)) = v_z(i) )}{P(f\text{-RR}(v_{a|i}(i)) = v_z(i) )} \ge 1$ is the worst case?
>
> A: Since $P(f\text{-RR}(v_a(i)) = v_z(i) )$ and $P(f\text{-RR}(v_{a|i}(i)) = v_z(i) )$ are probabilities, there are two cases: (1) $\frac{P(f\text{-RR}(v_a(i)) = v_z(i) )}{P(f\text{-RR}(v_{a|i}(i)) = v_z(i) )} \ge 1$ and (2) $0 \le \frac{P(f\text{-RR}(v_a(i)) = v_z(i) )}{P(f\text{-RR}(v_{a|i}(i)) = v_z(i) )} \le 1$.
>
> The first case is what we presented in our submission.  Let us consider the second case. When $0 \le \frac{P(f\text{-RR}(v_a(i)) = v_z(i) )}{P(f\text{-RR}(v_{a|i}(i)) = v_z(i) )} \le 1$, Eq.7 becomes:
>
> $ \frac{P(f\text{-RR}(v_a(i)) = v_z(i) )}{P(f\text{-RR}(v_{a|i}(i)) = v_z(i))}  \leq \frac{P(f\text{-RR}(v_{a|i}(i)) = v_z(i))}{  P(f\text{-RR}(v_a(i)) = v_z(i) )} $
>
>  $ \le \Big( \frac{P(f\text{-RR}(v_{a|i}(i)) = v_z(i))}{  P(f\text{-RR}(v_a(i)) = v_z(i) )}  \Big)^{\frac{\Delta_i}{|\mathcal{E}(f\text{-RR}(v_a, i)) - \mathcal{E}(f\text{-RR}(v_{a|i}, i))|}} \leq \exp(\epsilon_i)$
>
> Therefore, Eq.7 (in the submission) (equivalent to the above Eq.) still holds in this case, which enables us to quantify a generalized privacy loss bound of a RR mechanism (Theorem 1).
> From the above Eq., Eq.18 (in the submission) becomes:
>
> $ \frac{P( f\text{-RR}(v_x )
> = v_z)}{P(f\text{-RR}(\widetilde{v}_x ) = v_z )} $
>
> $ \le \prod_{i = 0}^{rl - 1}\Big(  \frac{P( f\text{-RR}(v_x(i))=v_z(i) )}{P(f\text{-RR}(v_{x|i}(i))=v_z(i) )} \Big)^{ \frac{\Delta_i}{\mathbb{E} |\mathcal{E}( f\text{-RR}(v_x,i ) )- \mathcal{E}( f\text{-RR} (v_{x|i},i)  )| }} $
>
> $ \le \prod_{i = 0}^{rl - 1}\Big(  \frac{ P(f\text{-RR}(v_{x|i}(i))=v_z(i) )  }{  P( f\text{-RR}(v_x(i))=v_z(i) ) } \Big)^{ \frac{\Delta_i}{\mathbb{E} |\mathcal{E}( f\text{-RR}(v_x,i ) )- \mathcal{E}( f\text{-RR} (v_{x|i},i)  )| }} $
>
> $ = \prod_{i = 0}^{rl - 1} \Big( \frac{ P(f\text{-RR}(v_x(i))=0 |v_x (i)=1 )  P(f\text{-RR}(v_x(i))=1 |v_x (i)=0 ) }{ P(f\text{-RR}(v_x(i))=1 |v_x (i)=1 )  P(f\text{-RR}(v_x(i))=0 |v_x (i)=0 ) } \Big)^{\frac{ 1 }{p_{Xi}^2 + q_{Xi}^2  }} $
>
> $ = \prod_{i = 0}^{l - 1} ( \alpha^2 \exp (2\epsilon_X \frac{i}{l}) )^{\frac{ r }{p_{Xi}^2 + (1-p_{Xi})^2  }}$
>
> Then, following all the same steps in Appendix E, we obtain the same results with the first case. Therefore, the bound in Theorem 2 holds for both cases.
>
> Thanks the reviewer for pointing this out. For the sake of completeness, we have revised our writing to clearly show these two cases, which lead to the same result that we have in our submission.
>
> Q8. Is not $|\mathcal{E}(f-RR(v_a,i)) - \mathcal{E}(f-RR(v_{a|i},i))|$ randomized (since f-RR is)? How does this imply $|\mathcal{E}(f-RR(v_a,i)) - \mathcal{E}(f-RR(v_{a|i},i))| < |f-RR(v_a,i) - f-RR(v_{a|i},i)|$?
>
> A: Yes, $|\mathcal{E}(f-RR(v_a,i)) - \mathcal{E}(f-RR(v_{a|i},i))|$ is not randomized. We use $|\mathcal{E}(f-RR(v_a,i)) - \mathcal{E}(f-RR(v_{a|i},i))|$ in computing the sensitivity and privacy loss bound and what we imply is $|\mathcal{E}(f-RR(v_a,i)) - \mathcal{E}(f-RR(v_{a|i},i))| \le |\mathcal{E}(v_a) - \mathcal{E}(v_{a|i})|$, not $|\mathcal{E}(f-RR(v_a,i)) - \mathcal{E}(f-RR(v_{a|i},i))| < |f-RR(v_a,i) - f-RR(v_{a|i},i)|$.

---

> > ### Comment · Reviewer_Sanq · 2021-11-30
> > **Thank you for your rebuttal**
> >
> > Thank you for the detailed  and comprehensive response to my questions. All of queries have been sufficiently answered.

---

> > > ### Author Response · Authors · 2021-11-30
> > > **We are happy to answer your questions.**
> > >
> > > Thanks for your response. We are happy to answer all your questions.

---

> ### Author Response · Authors · 2021-11-22
> **Response to Reviewer Sanq - Part 3**
>
> Q6. Anonymization work by Sun et al. and cryptographic techniques such as mixed nets.
>
> 1) Anonymization work by Sun et al.
>
> A: In Sun et al., they design a LDP mechanism to perturb the weights at the local client, then each local client applies a split and shuffle mechanism on the weights of local model and sends each weight through an anonymous mechanism to the cloud. The purpose of the shuffling mechanism is to break the linkage among the model weight updates from the same clients and to mix them among updates from other clients, making it harder for the cloud to combine more than one piece of updates to infer more information about any client. Therefore, the key idea of the shuffle mechanism in Sun et al. is to mitigate the privacy degradation by high data dimension and many training/query iterations. In other words, the client anonymity is
> preserved, and the privacy budget will not accumulate.
>
> When comparing with Sun et al., we maintain their mechanism' spirits of no privacy accumulation.
> In the submission, we consider there is no privacy accumulation over the training iterations. It is equivalent to the shuffling step that breaks the linkage among the model weight updates with associated clients. We also used the same randomized response mechanism in the paper, which is Eq.2 (Sun et al.), to perturb the weight.
>
> In the revision, we added an experiment  that do not consider the privacy accumulation over data dimension and training/query iterations. This completely follows the gist of Sun et al. In addition, the weights we used for Sun et al. without actual shuffling or splitting can be considered a lossless process, therefore the results we reported here can be counted as an upper-bound result for Sun et al.
>
> We obtained the slightly higher accuracy of Sun et al. compared with $f$-RR on the AG dataset. The key component of Sun et al. that helps to reduce the privacy accumulation issue is the shuffling mechanism. However, as pointed out in (Erlingsson et al., 2019), in the real world, it is possible that the anonymizers (i.e., shuffler) can either be compromised or collude with the coordinating server to extract sensitive information.
> Even though there is a marginally lower trade-off between privacy loss and model utility compared with Sun et al.,  the advantage of $f$-RR is that it perturbs the data only once, then used the perturbed data for training process without facing an extra privacy risk potentially caused by the compromised or colluded anonymizer.
>
> The results for $\epsilon \in [1,10]$ are as follows:
>
> $\epsilon$      $|1|2|3|4|5|6|7|8|9|10$
>
> Sun et al.     $|78.73| 82.21| 83.25| 84.19| 84.36| 84.22| 84.64| 84.43| 84.31| 84.72$
>
> $f$-RR         $|72.46| 79.72| 81.42| 79.35| 81.11| 79.84| 79.50| 80.05| 81.20| 82.72$
>
> Noiseless model $| 87.59$
>
>
> 2) Cryptographic techniques such as mixed nets.
>
> Mixed nets are typically a protocol or a mechanism that tries to break the link between the source of the request and the destination by using a chain of proxy servers (i.e., mixes). The mixed nets take in data or messages from multiple clients/senders, shuffle them, and send them in the random orders to the next destination.
> Adversaries can provide long term correlation attacks and track the sender and receiver of the packets. The protected information can be leaked if the entire path is revealed or the intermediate proxy servers are compromised. Therefore, mixed nets may fall into the same privacy risk as discussed in the shuffler in which the proxy servers can be compromised or collude with the server to extract the information. In addition, mixed nets introduce an notably large overhead in communication and computation cost.
> Different from these techniques, our proposed $f$-RR does not require any trusted server.
>  Thanks to the post-processing property of DP, applying $f$-RR into the inputs/embedded features and using the randomized embedded features for further calculation (e.g., gradients or model updates) is efficient and still LDP privacy preservation.

---

> ### Author Response · Authors · 2021-11-22
> **Response to Reviewer Sanq - Part 2**
>
> Q4. Clarification on the sensitivity of the sign bit.
>
> A: As showing in Appendix C (Proof of Lemma 1), the sensitivity of a bit i captures the magnitude by which the bit i can change the decoding function in the worst case. Therefore, if $i$ is the sign  bit ($i=0$), we have:
>
> $\Delta_i =\max_{v_a} \mathcal{E}(f-RR(v_{a}, 0)) -  \mathcal{E}(f-RR(v_{a|i}, 0)) $
>
> $= \max_{v_a}  \| (2 b_0-1)  \sum_{i=1}^{l-1} b_i  2^{m-i} -(2b^\prime_0-1) \sum_{i=1}^{l-1} b_i  2^{m-i}  \|_1 $
>
> $= \max_{v_a}  \| (2 b_0-2 b^\prime_0)  \sum_{i=1}^{l-1} b_i  2^{m-i} \|_1$
>
> $ \le  \max_{v_a}  ( | 2 b_0-2 b^\prime_0|  \| \sum_{i=1}^{l-1} b_i  2^{m-i} \|_1 )$.
>
> The sensitivity of bit $i$, which is $\Delta_i$, is maximized when $b_0$ and $b^\prime_0$ are different and all $\forall i \in [1, l-1]: b_i =1$. Then, $\Delta_i \le \max_{v_{a|0}} 2 \sum_{i=1}^{l-1} 2^{m-i} $. Since $2^1 + \ldots + 2^{l-2} = 2^{l-1} -2$ [Induction Proofs], we have: $2\sum_{i=1}^{l-1} 2^{m-i} = 2^{m+1-(l-1)} (2^{l-1}-1)  < 2^{m+1}$.
> As a result, $\Delta_i = 2^{m+1}$.
>
> [Induction Proofs] https://www.purplemath.com/modules/inductn3.htm
>
> Q5. Why is $\epsilon_i$ intractable (just below Eq. 6)? Is it not a user defined privacy parameter whose value is an input to the algo?
>
> A: In our mechanism, $\epsilon_X$ is a user-predefined privacy parameter, but $\epsilon_i$ for each bit is not. Instead of using a user-predefined $\epsilon_i$ which leads to a strong composition in privacy loss accumulation, we bound the privacy loss by identifying $\alpha$ (Theorem 2) to quantify the total privacy loss across all the bits, which is bounded by the user-predefined $\epsilon$. Therefore, $\epsilon_i$ is intractable.

---

> ### Author Response · Authors · 2021-11-22
> **Response to Reviewer Sanq - Part 1**
>
>
> We thank the reviewer for the constructive and helpful comments. Following are our responses:
>
> Q1. Why the privacy analysis is performed w.r.t. the encoding/decoding $\mathcal{E}$ instead of $f$-RR?
>
> A: There are two key reasons why we performed privacy analysis w.r.t. the encoding/decoding $\mathcal{E}$ instead of $f$-RR, as discussed next. In our threat model, the attacker aims to infer the original value of the randomized embedded features (extracted by applying gradients attack [Yin et al., 2021; Zhao et al., 2020; Zhu et al., 2019]). Therefore, we have to guarantee that the randomized responses applied on the binary encoding vector has to provide privacy protection at the embedded feature space, i.e., the results from the decoding $\mathcal{E}\big(f\text{-RR}(\cdot)\big)$.
> Second, different from RAPPOR, in which every bit is independent to each other and has the same sensitivity (sensitivity equals $1$), the sensitivity of bits in our $f$-RR are different given the embedded feature domain (Lemma 1). Therefore, solely conducting privacy analysis at the bit level of $f$-RR (i.e., sensitivity equals 1) is insufficient to provide protection at the embedded feature space. Consequently, performing the privacy analysis given the encoding/decoding $\mathcal{E}$ is necessary to provide $\epsilon_X$-LDP for the embedded features.
>
> [Yin et al., 2021] Yin, H., Mallya, A., Vahdat, A., Alvarez, J. M., Kautz, J., & Molchanov, P. (2021). See through Gradients: Image Batch Recovery via GradInversion. CVPR (pp. 16337-16346).
>
> [Zhao et al., 2020] Zhao, B., Mopuri, K. R., and Bilen, H. (2020). idlg: Improved deep leakage from gradients. arXiv preprint arXiv:2001.02610.
>
> [Zhu et al., 2019] Zhu, L., and Han, S. (2020). Deep leakage from gradients. In Federated learning (pp. 17-31). NeurIPS 2019.
>
>
> Q2. Motivation of the embedding step.
> A: Embedding is a necessary step to reduce the input dimension while enriching the information captured from the input data in the embedding space. Embedding step has been widely used and shown its effectiveness in many machine learning domains and systems, especially in natural language modeling where the input representation of text data, such as sentences, usually is sparse given the large dimension of vocabulary. Embedding features can be learned and reused across models. This is also true given image data.
>
> Directly applying our mechanism on $x$ does not affect the privacy gains hold true in Alg. 1. However, with the effectiveness of the embedding step as aforementioned, without the embedding step, the model utility could be affected.
>
>
> Q3. Comparison to label-RR showcased by Ghazi et al. 2021. Why is the evaluation baseline Label-Laplace instead of Ghazi et al.?
>
> Although our label-RR is inspired by the randomizing probability (Eq.1, Ghazi et al. 2021) in the LabelDP showcased by Ghazi et al. 2021, there are two major differences between our label-RR and LabelDP discussed next.
>
> \textbf{(1)} We combine $f$-RR and label-RR to completely protect a data sample. As pointed out in [Busa-Fekete et al. 2021], only protecting the label as in the LabelDP offers a weaker privacy protection than it appears, as the features are sufficiently predictive of the label, obscuring the label is not enough, as a classifier can still be trained on such noisy data; hence, a user experiences privacy loss due to both the public release of the features and the private release of the label. \textbf{(2)} The RRWithPrior algorithm that is used to guarantee LabelDP requiring  publicly available priors and the multi-stage training  (LP-MST)  illustrating RRWithPrior algorithm  cannot be straightforwardly applied to federated learning. In LP-MST algorithm, the dataset is partitioned into subsets, then based on the prior probability to randomize the label and add the data with that randomized label to the training data. These steps are presently applied on centralized training and it has not been show how to be effectively applied in federated learning.
>
> Using the upper bound of $\beta$ (Theorem 3) results in the same randomizing probabilities between label-RR and LabelDP. For instance, in Eq.3, $p_Y = \frac{\exp(\beta)}{ 1+ \exp(\beta)} = \frac{\exp(\epsilon_Y - \ln (C-1))}{ 1+ \exp(\epsilon_Y - \ln (C-1))} = \frac{\exp(\epsilon_Y)}{C-1+\exp(\epsilon_Y)}$ and $q_Y = \frac{1}{ (1+\exp(\beta)) (C-1) } = \frac{1}{ C-1 + \exp(\epsilon_Y)}$, which is equivalent to Eq.1 [Ghazi et al.]. Therefore, we did not include LabelDP [Ghazi et al.] in comparison. We have clarified why no comparison to LabelDP is in experiments in our revision.
>
> We included Label-Laplace as baseline since this is one of the well-known techniques to protect privacy for labels.
>
> [Busa-Fekete et al. 2021] Busa-Fekete, R. I., Syed, U., and Vassilvitskii, S. (2021, October). On the Pitfalls of Label Differential Privacy. In NeurIPS 2021 Workshop LatinX in AI.

---

### Official Review · Reviewer_UqE3 · 2021-11-03

**Correctness:** 3
**Technical Novelty And Significance:** 2
**Empirical Novelty And Significance:** 2
**Recommendation:** 6
**Confidence:** 3

**Main Review:**

The idea of this paper is to introduce a bit-aware randomization technique. That is a randomized response algorithm that flips a bit based on its position in the binary representation of the scalar (or naturally extend to a vector). I found the paper a bit hard to read because of clutter and too many notional changes. I have some clarifying questions from the authors:

1. What is the notation $i$%$\ell$ used in the definition of the bit-aware term? It needs to be defined more explicitly.

2. Is the decoding function $\mathcal E$ the averaging of the gradient? I would rather have the author clearly state that in the first 8 pages than point to the lines in the algorithm. I do not understand why the authors call it a decoding function to begin with. Decoding is a specific technical term in coding theory.

3. What is $f^\theta$? There is no such notation in the Algorithm. Is it a function taken from a set of functions parameterized by the model?

4. After equation (6), why is the ratio of the probabilities the worst case?

5. How does equation (6) happen? Why can you decompose the probability like that? If we change the binary vector by adding $1$, more than one coordinate gets affected; hence, the probability of seeing a bit in different coordinates is dependent. Am I missing something here? The same form of independence is claimed in the guarantee of Theorem 7.

6. Am I correct in understanding that $\epsilon_X$ is the privacy budget in the embedded space?

7. Coming to the proof of Theorem 2, we have $\mathbb E[ \cdot]$ in equation (13). This does not make sense in any practical setting. This pretty much means that the privacy bound is in the average case, which is weird!! From what I see, the end result would be some kind of average-case DP, which has its own pitfalls! Can the author clarify this?

8. What is the notion of neighboring dataset? It is very essential to know that before I can make a reasonable comparison with the previous work.

9. Randomized response does not give an unbiased estimate. On the other hand, to do SGD, we need unbiased estimate of the gradient while ensuring that the variance is bounded. So, I feel there should be some form of bias correction, which is not in this paper.

**Summary Of The Paper:**

The paper tries to address the question whether we can have different noise scale for different bits for a high dimensional vector and still preserves privacy. They propose a method to do that.

**Summary Of The Review:**

There are a lot of clarifying question that I do not understand in this paper.

The authors have answered most of the clarification questions I had during the author's response phase. I still feel the paper is cluttered and not ready for ICLR and at maximum is a borderline accept mainly due to the strength of the problem studied.

---

> ### Author Response · Authors · 2021-11-22
> **Response to Reviewer UqE3 - Part 3**
>
>
> Q3. Clarification on average-case DP.
>
> A: In our work, we consider the worst case is the case that all the bits of two neighboring vectors can be different. The expectation $\mathbb{E}|\mathcal{E}(f\text{-RR}(v_a, i)) - \mathcal{E}(f\text{-RR}(v_{a|i}, i))|$ in Eq.13 is used to
> quantify the difference of every two extreme vectors at bit $i$. Then for the whole vector, it is the sum over the expectation $\mathbb{E}(\cdot)$ of all bits $i$, as follows: $\sum_{a \in [1,r]} |\mathcal{E}(f\text{-RR}(v_a, i)) - \mathcal{E}(f\text{-RR}(v_{a|i}, i))| = r \times \mathbb{E}|\mathcal{E}(f\text{-RR}(v_a, i)) - \mathcal{E}(f\text{-RR}(v_{a|i}, i))|$.
> Therefore, the expectation in  Eq.13 does not imply the average-case scenario.
>
>
> The sum over the expectation $\mathbb{E}(\cdot)$ of all bits $i$, i.e., $\sum_{a \in [1,r]} \sum_{i \in [0, l- 1]} |\mathcal{E}(f\text{-RR}(v_a, i)) - \mathcal{E}(f-RR(v_{a|i}, i))| = r \times \sum_{i \in [0, l- 1]} \mathbb{E}|\mathcal{E}(f-RR(v_a, i)) - \mathcal{E}(f-RR(v_{a|i}, i))|$, is used to bound the privacy loss as follows:
>
> $\frac{P( f-RR(v_x ) = v_z)}{P(f-RR(\widetilde{v}_x ) = v_z )} $
>
> $ \le \prod_{a \in [1,r]} \prod_{i \in [0, l-1]} \big(\frac{P(f-RR(v_a(i)) = v_z(i))}{P(f-RR(v_{a|i}( i)) = v_z(i))} \big)^{\frac{\Delta_i}{\mathbb{E}|\mathcal{E}(f-RR(v_a, i)) - \mathcal{E}(f-RR(v_{a|i}, i))|}}$
>
>  $\leq \exp( \sum_{a \in [1, r]} \sum_{i \in [0, l-1]} \frac{\epsilon_i |\mathcal{E}(f-RR(v_a, i)) - \mathcal{E}(f-RR(v_{a|i}, i))|}{\Delta_i}) $
>
>  $= \exp(\sum_{i \in [0, l-1]} \frac{\epsilon_i \big( r \times \mathbb{E} |\mathcal{E}(f-RR(v_a, i)) - \mathcal{E}(f-RR(v_{a|i}, i))|\big)}{\Delta_i}) $
>
>  $\leq \exp(r \sum_{i \in [0, l-1]} \epsilon_i) $
>
> As a result, our mechanism quantifies the privacy loss given the worse case, that is, two extreme vectors can be different in any bits.
>
> Q4. Notion of neighboring dataset.
>
> A: We do not need a definition of neighboring dataset in our work, especially in LDP preservation. Instead, we consider neighbouring vectors in which the two vectors differ from all the bits (the worst case). This is similar to lines of work on randomized responses to preserve LDP, such as RAPPOR [Erlingsson et al., 2014], OLH, BLH, SUE, and OUE [Wang et al., 2017], LATENT [Arachchige et al., 2019], and OME [Lyu et al., 2020].
>
> Q5. Gradient estimation in SGD.
>
> A: Here, we randomize the binary encoded vectors of the embedded features. The randomized binary encoded vectors preserve LDP. Then, we decode the randomized binary encoded vectors to get their real values, which are treated as LDP-preserving embedded features. Following the post-processing property of DP, the gradients computed on the LDP-preserving embedded features preserve LDP. Therefore, it is not necessary to consider the variance of the gradient.
>
> [Dwork et al., 2014] Dwork, C., & Roth, A. (2014). The algorithmic foundations of differential privacy. Found. Trends Theor. Comput. Sci., 9(3-4), 211-407.
>
> [Erlingsson et al., 2014] Erlingsson, Ú., Pihur, V., & Korolova, A. (2014, November). Rappor: Randomized aggregatable privacy-preserving ordinal response. In Proceedings of the 2014 ACM SIGSAC conference on computer and communications security (pp. 1054-1067).
>
> [Wang et al., 2017] Wang, T., Blocki, J., Li, N., & Jha, S. (2017). Locally differentially private protocols for frequency estimation. In 26th {USENIX} Security Symposium ({USENIX} Security 17) (pp. 729-745).
>
> [Arachchige et al., 2019] Arachchige, P. C. M., Bertok, P., Khalil, I., Liu, D., Camtepe, S., & Atiquzzaman, M. (2019). Local differential privacy for deep learning. IEEE Internet of Things Journal, 7(7), 5827-5842.
>
> [Lyu et al., 2020] Lyu, L., Li, Y., He, X., & Xiao, T. (2020, July). Towards differentially private text representations. In Proceedings of the 43rd International ACM SIGIR Conference on Research and Development in Information Retrieval (pp. 1813-1816).

---

> ### Author Response · Authors · 2021-11-22
> **Response to Reviewer UqE3 - Part 2**
>
> Q2. Decomposition and the worst case of Equation (6)
>
> 1) Probability decomposition in Eq.6
>
> A: When applying $f$-RR to a binary encoding vector $v_x$, we apply $f$-RR independently on each bit of $v_x$. Therefore, the randomizing probability of one bit will not affect the randomization probability of another bit. These are independent randomization events across all the bits. To generalize privacy loss bound for all the bits in $v_x$ (Theorem 1) and due to the independent randomization events of $f$-RR, in Eq.6, we focus on investigating the privacy loss on protecting only a single bit $i$ while keeping all the other bits the same (no privacy loss).
>
> Therefore, we can decompose the probability $\frac{P(f\text{-RR}(v_a, i) = v_z)}{P(f\text{-RR}(v_{a|i}, i) = v_z)}$ into a product of each bit, as $\frac{P(f\text{-RR}(v_a, i) = v_z)}{P(f\text{-RR}(v_{a|i}, i) = v_z)} =  \prod_{j \in [0,l-1]} \frac{P(v_a(j) = v_z(j))}{P(v_{a|i}( j) = v_z(j))}= \frac{P(f\text{-RR}(v_a(i)) = v_z(i))}{P(f\text{-RR}(v_{a|i}( i)) = v_z(i))} \times \prod_{j \neq i, j \in [0,l-1]} \frac{P(v_a(j) = v_z(j))}{P(v_{a|i}( j) = v_z(j))}$. Since all the other bits differing from $i$ keep the same, $\forall j \neq i, j \in [0,l-1]: \frac{P(v_a(j) = v_z(j))}{P(v_{a|i}( j) = v_z(j))} = 1$. As a result, we have $\frac{P(f\text{-RR}(v_a, i) = v_z)}{P(f\text{-RR}(v_{a|i}, i) = v_z)} = \frac{P(f\text{-RR}(v_a(i)) = v_z(i))}{P(f\text{-RR}(v_{a|i}( i)) = v_z(i))}$. Based upon this, from our explanation before Eq.6, we have $\frac{P(f\text{-RR}(v_a(i)) = v_z(i))}{P(f\text{-RR}(v_{a|i}( i)) = v_z(i))} \le \exp(\frac{\epsilon_i |\mathcal{E}(f\text{-RR}(v_a, i)) - \mathcal{E}(f\text{-RR}(v_{a|i}, i))|}{\Delta_i})$.
>
> Here, we do not apply the adding operation to achieve local DP, instead we randomize each bit independently. Therefore, the probability of seeing a bit in different coordinates is independent, which is similar to previous work in randomized responses to provide LDP protection, such as RAPPOR [Erlingsson et al., 2014], OLH, BLH, SUE, and OUE [Wang et al., 2017], LATENT [Arachchige et al., 2019], and OME [Lyu et al., 2020].
>
> Could the reviewer please clarify what Theorem 7 is refered to? Since we only have 6 Theorems in total.
>
> 2) The worst case of the ratio $\frac{P(f\text{-RR}(v_a(i)) = v_z(i) )}{P(f\text{-RR}(v_{a|i}(i)) = v_z(i) )}$.
>
> A: Since $P(f\text{-RR}(v_a(i)) = v_z(i) )$ and $P(f\text{-RR}(v_{a|i}(i)) = v_z(i) )$ are probabilities, there are two cases: (1) $\frac{P(f\text{-RR}(v_a(i)) = v_z(i) )}{P(f\text{-RR}(v_{a|i}(i)) = v_z(i) )} \ge 1$ and  (2) $0 \le \frac{P(f\text{-RR}(v_a(i)) = v_z(i) )}{P(f\text{-RR}(v_{a|i}(i)) = v_z(i) )} \le 1$.
>
> The first case is what we presented in our submission.  Let us consider the second case. When $0 \le \frac{P(f\text{-RR}(v_a(i)) = v_z(i) )}{P(f\text{-RR}(v_{a|i}(i)) = v_z(i) )} \le 1$, Eq.7 becomes:
>
> $\frac{P(f\text{-RR}(v_a(i)) = v_z(i) )}{P(f\text{-RR}(v_{a|i}(i)) = v_z(i))}  \leq \frac{P(f\text{-RR}(v_{a|i}(i)) = v_z(i))}{  P(f\text{-RR}(v_a(i)) = v_z(i) )} $
>
>  $ \le \Big( \frac{P(f\text{-RR}(v_{a|i}(i)) = v_z(i))}{  P(f\text{-RR}(v_a(i)) = v_z(i) )}  \Big)^{\frac{\Delta_i}{|\mathcal{E}(f\text{-RR}(v_a, i)) - \mathcal{E}(f\text{-RR}(v_{a|i}, i))|}} \leq \exp(\epsilon_i)$
>
> Therefore, Eq.7 (in the submission) (equivalent to the above Eq) still holds in this case, which enables us to quantify a generalized privacy loss bound of a RR mechanism (Theorem 1).
> From the above Eq., Eq.18 (in the submission) becomes:
>
> $ \frac{P( f\text{-RR}(v_x )
> = v_z)}{P(f\text{-RR}(\widetilde{v}_x ) = v_z )} $
>
>  $\le \prod_{i = 0}^{rl - 1}\Big(  \frac{P( f\text{-RR}(v_x(i))=v_z(i) )}{P(f\text{-RR}(v_{x|i}(i))=v_z(i) )} \Big)^{ \frac{\Delta_i}{\mathbb{E} |\mathcal{E}( f\text{-RR}(v_x,i ) )- \mathcal{E}( f\text{-RR} (v_{x|i},i)  )| }} $
>
> $ \le \prod_{i = 0}^{rl - 1}\Big(  \frac{ P(f\text{-RR}(v_{x|i}(i))=v_z(i) )  }{  P( f\text{-RR}(v_x(i))=v_z(i) ) } \Big)^{ \frac{\Delta_i}{\mathbb{E} |\mathcal{E}( f\text{-RR}(v_x,i ) )- \mathcal{E}( f\text{-RR} (v_{x|i},i)  )| }} $
>
> $ = \prod_{i = 0}^{rl - 1} \Big( \frac{ P(f\text{-RR}(v_x(i))=0 |v_x (i)=1 )  P(f\text{-RR}(v_x(i))=1 |v_x (i)=0 ) }{ P(f\text{-RR}(v_x(i))=1 |v_x (i)=1 )  P(f\text{-RR}(v_x(i))=0 |v_x (i)=0 ) } \Big)^{\frac{ 1 }{p_{Xi}^2 + q_{Xi}^2  }} $
>
> $ = \prod_{i = 0}^{l - 1}  \Big( \alpha^2 \exp (2\epsilon_X \frac{i}{l}) \Big)^{\frac{ r }{p_{Xi}^2 + (1-p_{Xi})^2  }}$
>
> Then, following all the same steps in Appendix E, we obtain the same results with the first case. Therefore, the bound in Theorem 2 holds for both cases.
>
> Thanks the reviewer for pointing this out. For the sake of completeness, we have revised our writing to clearly show these two cases, which lead to the same result that we have in our submission.

---

> > ### Comment · Reviewer_UqE3 · 2021-11-30
> > **Re: Response to Reviewer UqE3 - Part 2**
> >
> > Thanks to the authors for the clarification on this. I apologize for being so late in my response.
> > These remarks answers all of my questions. I still feel that the paper can do a lot better with a good exposition and just concentrating on one aspects of the paper. Rest of the things can go in the appendix. Unfortunately, I have to agree with another reviewer (Reviewer S5g9) that the paper is not ready for ICLR as in its present form. The paper does study an important and the authors do a good job in answering the questions. I think a well written paper for this problem would have an impact, but not well written might have a good chance of being lost in the crowd of papers studying similar problems. I cannot champion the paper, but I am more than ready to update my score to think that the paper has borderline chance to be accepted.

---

> > > ### Author Response · Authors · 2021-11-30
> > > **Thanks for the Reviewer's Feedback**
> > >
> > > We are thankful for the reviewer's feedback and do appreciate for recognizing the significance of our work. We will reorganize our paper to highlight the core contribution of our works better.

---

> ### Author Response · Authors · 2021-11-22
> **Response to Reviewer UqE3 - Part 1**
>
> We thank the reviewer for constructive and helpful comments. Following are our responses:
>
> Q1. Clarification on: 1) the notation $i\%l$, 2) decoding function $\mathcal{E}$, 3) $f^{\theta}$, and 4) privacy budget $\epsilon_X$
>
> A: We have revised our writing to add the following explanations to the revision.
>
> 1) In the definition of the bit-aware term, we use $i\%l$ to indicate the location of the bit $i$ in each embedded feature. The location is associated with the sensitivity of the bit at that location. There are $r$ embedded features, each of which is encoded using $l$ bits. Each bit in $l$ bits has different sensitivity (Lemma 1), but they share the same sensitivity at the same location across the binary encoding vectors of different embedded features. Therefore, $i\%l$ is used to reflect the exact location of bit $i$ in its $l$-bit binary encoded vector among $rl$ concatenated binary bits.
>
> 2) The decoding function $\mathcal{E}$ is not the averaging of the gradient. It is to converse a binary encoded vector back to its real value, which is used as an embedded feature.
>
> 3) $f^{\theta}$ is the model that $N$ clients jointly train by minimizing the loss function on their local training data. It is used to map the embedded feature $e_x$ to a vector of scores $y_x$. $\theta$ is the model's parameter updated during training.
>
> 4) $\epsilon_X$ is the privacy budget in the embedded space.

---

### Author Response · Authors · 2021-11-23
**We are thankful for constructive reviews!**

Dear Reviewers,

We thank all the reviewers for your constructive comments on our work. We have addressed all the questions and concerns in our rebuttal and revised our paper to reflect them.

We look forwards to further discussion to make our paper even stronger and more meaningful. If there is any further question or suggestion, please feel free to let us know.

Best regards,

The authors

---

### Decision · Program_Chairs · 2022-01-20

**Decision:**

Reject

**Comment:**

The paper proposes BitRand, a bit-aware randomized response algorithm, to preserve local differential privacy in federated learning. The main idea is to take into account the bit indices and prioritise higher order bits focussed towards achieving a utility which is higher than other algorithms which are oblivious to the floating point bit representation. Additionally, the analysis allows the bit randomization probabilities not to be affected too much by the dimension of the data.

Overall, the paper lay right at the borderline of acceptance. The paper's core idea, their development and experiments were all liked by the reviewers. The main issue the reviewers brought up was the writing and structuring of the paper and the presentation of the overall results. Most reviewers agreed that the paper presentation was not ready to match the bar for ICLR. There are multiple suggestions that reviewers have made - through their questions and direct comments and addressing those and rewriting the paper to highlight these aspects better will significantly improve the paper.